# Silicon uptake and isotope fractionation dynamics by crop species

Daniel A. Frick[1], Rainer Remus[2], Michael Sommer[2,3], Jürgen Augustin[2], Danuta Kaczorek[2], Friedhelm von Blanckenburg[1,4]

[1]GFZ German Research Centre for Geosciences, Potsdam, 14473, Germany.
[2]Leibniz Centre for Agricultural Landscape Research (ZALF), Müncheberg, 15374, Germany.
[3]Institute of Environmental Science and Geography, University of Potsdam, Potsdam, 14476, Germany
[4]Institute of Geological Science, Freie Universität Berlin, Berlin, 12249, Germany.

*Correspondence to*: Daniel A. Frick (dfrick@gfz-potsdam.de)

**Abstract.** That silicon is an important element in global biogeochemical cycles is widely recognized. Recently, its relevance for global crop production has gained increasing attention in light of possible deficits in plant-available Si in soil. Silicon is beneficial for plant growth and is taken up in considerable amounts by crops like rice or wheat. However, plants differ in the way they take up silicic acid from soil solution, with some species rejecting silicic acid while others actively incorporate it. Yet because the processes governing Si uptake and regulation are not fully understood these classifications are subject to intense debate. To gain a new perspective on the processes involved, we investigated the dependence of silicon stable isotope fractionation on silicon uptake strategy, transpiration, water use, and Si transfer efficiency. Crop plants with rejective (tomato, *Solanum lycopersicum* and mustard, *Sinapis alba*) and active (spring wheat, *Triticum aestivum*) Si uptake were hydroponically grown for 6 weeks. Using inductively coupled plasma mass spectrometry, the silicon concentration and isotopic composition of the nutrient solution, the roots, and the shoots were determined. We found that measured Si uptake does not correlate with the amount of transpired water and is thus distinct from Si incorporation expected for unspecific passive uptake. We interpret this lack of correlation to indicate a highly selective Si uptake mechanism. All three species preferentially incorporated light $^{28}$Si, with a fractionation factor $1000 \cdot \ln(\alpha)$ of -0.33 ‰ (tomato), -0.55 ‰ (mustard) and -0.43 ‰ (wheat) between growth medium and bulk plant. Thus, even though the rates of active and passive Si root uptake differ, the physico-chemical processes governing Si uptake and stable isotope fractionation do not. We suggest that isotope fractionation during root uptake is governed by a diffusion process. In contrast, the transport of silicic acid from the roots to the shoots depends on the amount of silicon previously precipitated in the roots and the presence of active transporters in the root endodermis, facilitating Si transport into the shoots. Plants with significant biogenic silica precipitation in roots (mustard, and wheat), preferentially transport silicon depleted in $^{28}$Si into their shoots. If biogenic silica is not precipitated in the roots, Si transport is dominated by a diffusion process and hence light silicon $^{28}$Si is preferentially transported into the tomato shoots. This stable Si isotope fingerprinting of the processes that transfer biogenic silica between the roots and shoots has the potential to track Si availability and recycling in soils and to provide a monitor for efficient use of plant-available Si in agricultural production.

## 1 Introduction

Silicon (Si) is the second-most abundant element in the Earth's crust and occurs in a wide variety of silicate minerals. Weathering of these minerals mobilises Si and represents the starting point of Si biogeochemical cycling in terrestrial ecosystems – a sometimes complex web of Si transfers and transformations. One crucial but poorly understood aspect of terrestrial Si biogeochemistry is biological cycling (Carey and Fulweiler, 2012; Derry et al., 2005; Sommer et al., 2006, 2013). Si has well documented biological roles (Cooke et al., 2016; Frew et al., 2018; Katz, 2019), and may be recycled multiple times through higher plants before being lost from an ecosystem. Today, agricultural land use exerts an increasing influence on the Si cycle, and in the future widespread deficits in plant-available Si in soils might develop (Carey and Fulweiler, 2016). Such shortages would endanger future food production. Strategies for addressing this potential problem requires, among other things, a better knowledge of Si uptake dynamics. Thus, approaches are needed that identify the processes governing Si uptake and regulation thereof. Here we propose and validate geochemical tools to trace plant Si uptake, to improve our ability to address questions not only on weathering, ecosystem nutrition strategies, and geo-pedo-biosphere interactions but also plant physiological processes.

Despite having a disputed biochemical role, Si is considered beneficial for plant growth, including crops: Si increases abiotic stress mediation (aluminium and heavy metal toxicity, salinity), biotic stress resistance (defence against herbivores), and improves the plants' structural stability (Cooke et al., 2016; Coskun et al., 2019b; Epstein, 1994, 1999, 2001; Exley and Guerriero, 2019; Frew et al., 2018; Katz, 2019; Ma, 2004; Richmond and Sussman, 2003). Higher plant species form a continuous spectrum in the extent to which Si is incorporated. Traditionally, higher plants were grouped into three categories: active, passive and rejective, according to the amount of Si taken up (Marschner and Marschner, 2012). Active species (e.g. rice, and wheat) take up Si with a higher silicon / water ratio than that in the soil solution, thus enriching Si relative to transpired water. Passive uptake species (most dicotyledons) neither enrich nor deplete the Si relative to the transpired water. Rejective species (e.g. tomato, mustard, and soybean) strongly discriminate against Si during uptake (Epstein, 1999; Hodson et al., 2005; Ma et al., 2001; Takahashi et al., 1990). However, whether the terminology "active" or "passive" is justified is subject to an intense debate that revolves around the evidence for involvement of an active, metabolically controlled process in some plant species (Coskun et al., 2019a; Exley, 2015; Exley et al., 2020).

Progress in this debate depends on identifying the transporters and mechanism that regulates Si uptake. In this regard genome sequencing has disclosed the transporters responsible for Si uptake (Ma & Yamaji, 2006; Ma *et al.*, 2006, 2007; Mitani *et al.*, 2009, see also Ma & Yamaji, 2015; YAN *et al.*, 2018 for an overview). In rice, a cooperative system of Si-permeable channels at both the root exodermis and endodermis (called Lsi1, Low Silicon 1 transporter, a thermodynamically passive transporter from the family of aquaporin-like proteins) incorporates Si, whereas a metabolically active efflux transporter (Lsi2, a putative anion-channel transporter) loads Si into the xylem (Broadley et al., 2012). The research on the identification of molecular

pathways and mechanisms supplements and extends the phenomenological classification of the Si uptake, in particular where genomic data is available that disclose functional Si transporters (Coskun et al., 2019b). Even this approach, however, does not seem to be sufficient to describe the real complexity of Si uptake. Recent empirical studies demonstrated the simultaneous operation of passive uptake mechanisms and actively facilitated Si uptake through Si uptake transporter (Sun et al., 2016b; YAN et al., 2018). Yet other researchers have suggested that the low permeability of Lsi1 does not permit the transfer of silicic acid at all (Exley et al., 2020). Thus, it remains debated what contribution active and passive Si transporters make during Si uptake by the different plant species.

Conventional approaches employed in the study of uptake, translocation, and accumulation of Si in living organisms include either radioactive tracers (e.g. $^{31}$Si, $^{32}$Si) or homologue elements (e.g. Germanium and the radionuclide $^{68}$Ge). Both techniques impose limitations on growth experiments, either due to safety concerns arising from radioactivity or due to physiological differences between the homologue element Ge and Si (Exley et al., 2020; Takahashi et al., 1990). As a homologue element, Ge is taken up in the same form as Si, $Ge(OH)_4^0$. In the absence of Si, plants seem to incorporate $Ge(OH)_4$ at a higher rate than in its presence (Takahashi et al., 1990). Several studies have shown that plants fractionate Si relative to Ge, resulting in a lowered Ge/Si ratio in the phytoliths formed (Blecker et al., 2007; Cornelis et al., 2010; Derry et al., 2005; Opfergelt et al., 2010). There is also evidence that Ge interacts differently with organic molecules than Si (Pokrovski and Schott, 1998; Sparks et al., 2011; Wiche et al., 2018). In some cases, Ge also appears to be toxic to organisms (Marron et al., 2016). Thus, Ge or Ge/Si ratios are problematic tracers of plant Si uptake and translocation processes.

Si stable isotope ratios provide a powerful alternative approach. Each physico-chemical transport process (e.g. absorption, uptake, diffusion, and precipitation) may be accompanied by a shift in an element's stable isotope ratios - so-called mass-dependent isotope fractionation (Poitrasson, 2017). This isotope fractionation either entails an equilibrium isotope effect, where the isotopes are partitioned between compounds according to bond strength, or a kinetic isotope effect, where the isotope fractionation depends on the relative rate constants of reactions involving the different isotopologues. For stable Si isotope fractionation in aqueous media, both equilibrium effects (He et al., 2016; Stamm et al., 2019) and kinetic effects (Geilert et al., 2014; Oelze et al., 2015; Poitrasson, 2017; Roerdink et al., 2015) have been observed. In plant growth studies Si isotope ratio measurements, when combined with establishing the Si mass balance, isotope fractionation factors, and plant physiological properties allows the exploration of Si pathways in higher plants.

Previous studies on stable Si fractionation in higher plants focused on accumulator plants, namely rice (Ding et al., 2008a; Köster et al., 2009; Sun et al., 2008, 2016b, 2016a), banana (Delvigne et al., 2009; Opfergelt et al., 2006, 2010), bamboo (Ding et al., 2008b) and cucumber (Sun et al., 2016b) and most of these studies show the preferential incorporation of lighter Si isotopes. Importantly, in most of these studies, Si concentrations in the growth media were held constant by frequently replenishing the nutrient solution. This imparts the disadvantage that the dynamics (temporal evolution) of the Si isotope

fractionation during uptake cannot be derived from the isotope shift recorded by the nutrient solution over the course of the experiment, nor does the provision of constant Si amounts allow additional constraints to be placed on Si uptake mechanisms employed by plants.

In this study we elucidated the mechanisms of Si uptake using crop species that differ significantly in their Si uptake capacity, the presence of specific Si transporters and their transpiration rate. To do so, we combined the measurement of physiological plant performance ratios with observations of the shifts in the Si isotope ratios due to mass dependent isotope fractionation. Three crops - tomato, mustard, and wheat - were grown in a hydroponic system under the same environmental conditions, with nutrients being supplied only once, during the onset of the experiment, allowing direct quantification of the dynamics of isotopic fractionation from the temporal evolution of the nutrient solutions' isotopic composition. With the combination of the physiological plant performance ratios and isotope chemical parameters we developed new insights to the mechanisms underlying the different Si uptake and translocation strategies.

## 2 Materials and methods

### 2.1 Nutrient solution

The nutrient solution was prepared from technical grade salts following the recipe after Schilling *et al.*, 1982; and Mühling & Sattelmacher, 1995. Silicon was added in the form of sodium silicate trihydrate ($Na_2O_7Si_3 \cdot 3H_2O$) to an initial Si starting concentration of 49.5 μg·g$^{-1}$ (1.76 mM). Detail composition can be found in supplementary methods S1. Ultrapure water (resistivity 18.2 MΩ·cm) was used to prepare the nutrient solutions and to weekly restock water transpired by the plants.

### 2.2 Plant species

Three species were chosen based on their silicon uptake characteristics, the ability to grow in hydroponic environments, and previous knowledge about their Si transporter. Tomato (*Solanum lycopersicum* cultivar MICRO TOM) and mustard (*Sinapis alba*) are both rejective of Si, while spring wheat (*Triticum aestivum* cultivar SW KADRILJ) actively takes up Si (Hodson et al., 2005; Takahashi et al., 1990). The two Si excluder species differ in the presence of the NOD26-like-instrinsic proteins (orthologues of Lsi1, homologous gene sequence of low-silicon rice 1) which are associated with the transport of Si. In the family of Brassicaceae (mustard) these are absent (Sonah et al., 2017), whereas for tomato the Lsi1 homologue seems to be present but inactive (Deshmukh et al., 2016, 2015). Conversely, the alleged active Si efflux transporter (Lsi2-like) are present in the family of Brassicacea (Sonah et al., 2017), but not in tomato (Sun et al., 2020). An ongoing controversy surrounds the significance of the Lsi1 homologue in tomato. Whereas Deshmukh et al., 2015 used Si uptake studies to infer the transporter to be non-functional, Sun et al., 2020 observed the contrary using Ge as homologue element. Sun and co-workers concluded that the low Si uptake is caused by the lack of a functional Si efflux transporter Lsi2 at the root endodermis.

## 2.3 Plant germination and growth conditions

Plant seeds were germinated in Petri dishes with half-strength nutrient solution used for the later growth experiment that contained no added sodium silicate trihydrate. After cotyledons formed, seedlings were transferred into a foam disk and grown for a further two weeks in the same half-strength nutrient solution. Four plants each were then transferred into one experimental container that was filled with fresh nutrient solution including sodium silicate trihydrate, and each species was replicated in three containers. Plants were germinated and grown in a growth chamber under controlled climate conditions. Each week the pots were weighed without the lid and the plants, and the mass of transpired water was replenished with ultrapure water (18.2 MΩ·cm). The weight difference to the previous week is considered to quantify the mass of water transpired by the plants. The pots were closed with a fixed and completely sealed lid, and thus evaporation is considered to be very small and, in any case, identical between the plant species and triplicates. The temperature in the growth chamber during the day and night was maintained at 18 °C for 14 h and at 15 °C for 10 h, respectively, and the daylight intensity at the top of the container was adjusted to 350 µE·m$^{-2}$·s$^{-1}$) at the start of the experiment. The relative humidity was maintained at approximately 65 %. Details of the plant germination and growth conditions are provided in supplementary methods S2.

## 2.4 Sampling

The nutrient solutions were sampled at the start of the experiment and then every seven days until harvesting. For sampling, 40 mL were taken after replenishing water loss via transpiration loss and mixing of the solution. All sampled nutrient solutions were stored until analysis in precleaned PP vials in darkness at 4 °C. The 280 mL sample taken over the course of 6 weeks corresponds to 3.5 % of the initial nutrient solution. After 6 weeks the plants were harvested, and shoots (stem and leaves) were separated from the roots. The roots were immersed multiple times in ultrapure water to remove potential extracellular Si deposits and attached nutrients. The plant parts were dried at 104 °C to constant weight.

## 2.5 Determination of concentrations and isotope ratios

The chemical compositions of the growth solution and the digested plant samples (see section 2.5.2 for the digestion procedure) were measured using an axial inductively coupled plasma optical emission spectroscopy (ICP-OES, Varian 720-ES, instrument settings are reported in Table S1). Samples and standard were analysed following a procedure by Schuessler *et al.*, 2016. Briefly, the samples and standards were doped with an excess of CsNO$_3$ (1 mg g$^{-1}$) to reduce matrix effects in the ICP source that are likely to be caused from the high nitrogen content of the samples and quantified applying an external calibration. The relative analytical uncertainties are estimated to be below 10% and agreed with the nominal concentration of the starting solutions.

### 2.5.1 Nutrient solution purification

The high nutrient content and the organic acids in the nutrient solution potentially impair the chromatographic purification of Si. Thus the nutrient solution was digested following the "Sample preparation of water samples" by Steinhoefel et al., 2017 without employing an additional step for the removal of dissolved organic carbon. Briefly, based on the concentration measured, an aliquot of each nutrient solution containing approximately 1000 µg Si was dried down in silver crucibles on a hotplate at 80-95 °C. The crucibles were then filled with 400 mg NaOH (Merck pellets, p.a. grade, previously checked for low Si blank levels) and ultrapure water to the initial fill level and dried down. This step ensured that Si attached to the crucible walls was also immersed in NaOH. A blank containing ultrapure water and NaOH was processed in parallel to the samples to check for contamination of Si and other elements introduced in the procedure.

### 2.5.2 Plant samples digestion

The oven-dried samples were homogenised by milling the plant parts in a tungsten carbide planetary ball mill (Pulversiette 7, Fritsch). 50-800 mg of plant material, depending on the Si concentration determined in an exploratory subset of the samples, was weighed into Ag crucibles and combusted overnight (2h at 200 °C, 4h at 600 °C, then cooled to room temperature) in a furnace (LVT 5/11/P330, Nabertherm). A blank (empty crucible) was processed together with the samples. After cooling 400 mg NaOH (TraceSELECT, Sigma-Aldrich, checked for low Si blank levels) was added.

### 2.5.3 Fusion and chromatography

The crucibles containing the sample (nutrient solution or plant material) and NaOH were placed in a furnace at 750 °C for 15 min to perform the fusion. The fusion cake was dissolved in ultrapure water (for 24h, followed by 30 min ultrasonic bath), the solution was decanted into precleaned PP flask. The remains of the fusion cake were fully dissolved in 0.03 M HCl (for 3h), and both solutions were combined and the pH was adjusted to 1.5. The Si concentration was determined by ICP-OES and approximately 60 µg Si (present in the form of silicic acid) was chromatographically separated using cation exchange resin (following a procedure outlined by Georg *et al.*, 2006; Zambardi & Poitrasson, 2011; Schuessler & von Blanckenburg, 2014). The Si yield of the fusion procedure and the column chemistry was determined in a 1:10-fold dilution by ICP-OES. Si blanks of the fusion and column separation procedure were in general below 1 µg Si, equivalent to less than 1 % of the total Si processed. See Methods S3 for more details.

### 2.5.4 Silicon isotope ratio measurements

The purified solutions were acidified to 0.1 M HCl and diluted to a concentration of 0.6 µg·g$^{-1}$. Sample and standard were both doped with 0.6 µg·g$^{-1}$ Mg and the $^{25}$Mg/$^{24}$Mg ratio used as a monitor of mass bias drift and to ensure stable measurement conditions during the analysis (Oelze et al., 2016). The solutions were introduced using an ESI ApexHF desolvator and a PFA nebuliser (measured uptake 140 µL min$^{-1}$) into the MC-ICP-MS (Neptune, equipped with the Neptune Plus Jet Interface,

Thermo Fisher Scientific; instrument settings are given in Table S1). Measurements were made in dynamic mode (magnet jump) alternating between Si and Mg isotopes, each for 30 cycles with 4 s integration time. ERM-CD281 and BHVO-2 were analysed together with the nutrient and plant samples to ensure complete fusion, dissolution, and chromatographic separation. ERM-CD281 resulted in $\delta^{30}Si$ = -0.34 ± 0.20 ‰, 2s, n=13 and BHVO-2 in $\delta^{30}Si$ = -0.29 ± 0.09 ‰, 2s, n=40, in line with literature values (Jochum et al., 2005 for BHVO-2 and Delvigne et al., 2019 for ERM-CD281). The results of reference materials are reported in the supplementary information in Table S2, and the results of growth solutions and plants in Table S3 and Table S4. All $\delta^{29/28}Si$ and $\delta^{30/28}Si$ are reported in delta notation relative to NBS28 (NIST SRM8546) unless stated otherwise (Coplen et al., 2002; Poitrasson, 2017). An isotopic difference between two compartments is expressed as $\Delta^{30}Si$, calculated following Eq. (1):

$$\Delta^{30}Si_{a-b} = \delta^{30}Si_a - \delta^{30}Si_b \qquad (1)$$

where $\delta^{30}Si_a$ is the Si isotopic composition of the compartment a and $\delta^{30}Si_b$ the composition of compartment b. The silicon isotopic composition of a bulk plant is calculated from the mass weighted Si isotopic composition of separate plant parts and expressed as $\delta^{30}Si_{plant}$:

$$\delta^{30}Si_{plant} = \frac{\delta^{30}Si_{root} \cdot M_{root} + \delta^{30}Si_{shoot} \cdot M_{shoot}}{M_{root} + M_{shoot}} \qquad (2)$$

where the subscripts plant, root and shoot refer to the bulk plant, and roots and shoots, respectively, and M is the mass of silicon incorporated into the roots or shoots of the plant.

## 2.6 Plant performance ratios, elemental and isotopic budgets

### 2.6.1 Plant performance ratios

We define the plant transpiration as the amount of water taken up by the plants via the roots. Transpiration was measured weekly by weighing the remaining growth solution with the lids and plants removed. The difference in mass from the previous week is considered to be the mass of water transpired by the plants. The gravimetrically determined transpiration does not account for the amount of water present in the plants at harvest nor any possible guttation (Joachimsmeier et al., 2012). At the end of the experiment, the following plant performance ratios were calculated:

1. Water use efficiency: total dried phytomass (g) divided by the amount of transpired water (L), calculated separately for each pot.

2. Si uptake efficiency: total Si mass (mg) in plants divided by the amount of transpired water (L), calculated separately for each pot.

3. Si transfer efficiency: Si mass (mg) in plant shoots divided by the amount of transpired water (L), calculated separately for each pot.

We also calculated an "expected Si uptake" defined to represent exactly the mass of Si contained in the water utilised. This value was calculated from on the amount of transpired water and the nutrient solution Si concentration determined in the week prior:

$$Expected\ Si\ Uptake = \sum_{Week=1}^{Week=6}[Si]_{week\ i-1} \cdot m_{transpired\ water,week\ i} \tag{3}$$

where $[Si]_{week\ i-1}$ is the Si concentration in the nutrient solution the week prior, and $m_{transpired\ water,\ week\ i}$ the mass of water transpired during past week. The plant Si uptake characteristics can be classified based on the ratio between the measured (based on the biomass and the Si concentration measured therein) and the expected Si uptake. A ratio of greater than 1 indicates an active uptake mechanism, a ratio much smaller than 1 a rejective strategy, and a ratio of 1 indicates passive uptake.

### 2.6.2 Element budgets

The digested plant samples and nutrient solutions were analysed prior to the column purification by ICP-OES, and the concentrations of major elements (Ca, Fe, K, Mg, P, S and Si) and the retrieval was determined using Eq. (4):

$$Retrieval^X = \frac{M^X_{Solution,end} + M^X_{Plants}}{M^X_{Solution,start}}\ in\ [\%] \tag{4}$$

where $M_{solution,\ end}$ is the mass of the element X in the solution at the end of the experiments, $M_{Plants}$ is the mass of the element X in the plants, and $M_{Solution,\ start}$ the mass of the element X in the solution at the beginning of the experiment.

### 2.6.2 Silicon isotope budget

A simple test of whether incomplete recovery of Si or analytical artefacts in the Si isotope composition measurements are affecting the results is offered by an isotope budget. The concept is that the summed Si isotope composition of the remaining growth solution at the end of the experiment and the Si taken up by plants should be identical to the Si isotope composition of the initial growth solution. The Si total isotope composition at harvest is estimated using Eq. (5):

$$\delta_{Total} = \frac{M^{Si}_{solution}\delta^{30}Si_{solution} + M^{Si}_{plants}\delta^{30}Si_{plants}}{M^{Si}_{solution} + M^{Si}_{plants}} \tag{5}$$

where $M^{Si}_{solution}$ and $M^{Si}_{plants}$ are the Si amounts in the remaining nutrient solution and the plant parts at harvest, respectively, and $\delta^{30}Si_{solution}$ and $\delta^{30}Si_{plants}$ the Si isotope composition of the remaining nutrient solution and plants parts at the end of the experiment, respectively.

### 2.7 SEM-EDX analysis of mustard root phytoliths

To explore the form of silica in mustard roots, phytoliths were extracted and visualised using SEM-EDX. One gram of dried mustard roots was taken for analysis. Removal of organic matter was conducted by igniting the samples in a muffle furnace at 500°C for 5h. The residue was subjected to additional oxidation using 30% $H_2O_2$ for 0.5h. Ca oxalates were dissolved by 80°C in HCl (10Vol.%) for 10 min. The residue was washed with water, and dried at 105°C. SEM-EDX analysis was performed with a ZEISS EVO MA10 (HV, LV, LaB6 cathode) equipped with a Bruker QUANTAX EDS system including a liquid

nitrogen free XFlash R 5010 Detector (energy resolution of 123 eV for MNKa at 100,000cps). The SEM operated at 20keV, with an average working distance of 10.5 mm. Software: Esprit 2.1.1., incl Qmap.

## 3 Results

### 3.1 Plant dry mass and transpiration

Substantial differences are apparent in the growth rate between and within all three plant species. During the six-week period mustard formed the greatest amount of dry biomass, with an average of 7 g per plant (range: 0.7 - 16.6 g). Spring wheat produced on average 4 g (range: 1.9 - 5.6 g), and tomato produced the lowest amount of biomass per plant with an average of 3 g (range: 0.2 – 8.7 g, see Table 1 and Table S4 for the individual results). No dependence of replicated growth experiments on pot placement or proximity to the venting system was apparent. The amount of water transpired by the plants during the growth period is correlated with the biomass formed ($r_{Spearman\ Rank} = 0.95$, p-value <0.001). In contrast, no differences between plant species were observed in terms of the shoot-root ratios (5.4 – 6.5 g·g$^{-1}$, Table 2).

### 3.2 Dynamics of water, Si and other nutritive elements uptake

The three plant species revealed quite different transpiration dynamics during the 6 weeks of plant growth. After a lag phase of two weeks, differences in transpiration between mustard and the other two species became apparent. Figure 1a shows the cumulative transpiration for the three replicate growth experiments and species (see Table S6 for the individual transpired water amounts). Mustard showed the highest, wheat intermediate and tomatoes the lowest cumulative transpiration. The water use efficiency (see 2.6.1) of tomato was significantly higher (3.8 g·L$^{-1}$) than that of the other two plant species (2.4 - 2.6 g·L$^{-1}$, Table 2).

Based on the temporal evolution of Si concentrations in the nutrient solutions (Figure 1b) spring wheat exhibited the highest total Si uptake, mustard an intermediate amount, and tomato the lowest total Si uptake and the Si contents of bulk plants reflect this sequence (Table 1): spring wheat as an Si accumulator took up the most Si (448 mg), followed by mustard (150 mg). Tomato took up the least amount (95 mg). Considering only roots, the highest Si concentrations and Si amounts were found in mustard, while spring wheat and tomato were significantly lower. In contrast, considering only plant shoots, the highest Si mass were found in wheat while Si concentrations in mustard and tomato were similar, but more than an order of magnitude lower (Table 1). Spring wheat also showed a much higher Si uptake efficiency than the other two plant species, which resemble each other (Table 2 and Figure 1). The same trend holds for the Si mass ratio between roots and shoots (Table 2). Moreover, wheat shows a much higher efficiency of Si transport into the shoot per mass of transpired water than the other two plant species. In contrast to the Si uptake efficiency, the Si mass ratio between root and shoot for mustard was lower than for tomato (Table 2). For the calculation of Si uptake rates, we assume there is no back diffusion or efflux of Si out of the plant roots.

Such a process has not been reported in the literature and would be driven against the concentration difference between the root and the nutrient solution Si concentration and against the water flow direction (Raven, 2001).

The expected Si uptake (see 2.6.1 and Eq. 3 for a definition) traces the passive uptake of Si contained in the water utilised by the plants. The dynamics throughout the experiment is shown in Figure 1c (closed symbols) together with the ratio of measured and expected Si uptake (open symbols) at the end of the experiment. The measured and expected Si uptake ratios for all three species deviate significantly from 1 (see Table 2). The means of the measured and expected Si uptake for mustard (57.2[a] ± 1.3 mg vs 457.9[b] ± 16.4 mg), wheat (337.0[b] ± 67.9 mg vs 177.3[a] ± 40.7 mg) and tomato (15.5[a] ± 4.9 mg vs 141.1[b] ± 27.0 mg) are significantly different (denoted [a/b], based on t-test at 5% significance level,). This indicates that Si uptake or transport in the three plant species investigated under the given environmental conditions differs from unspecific passive uptake or unspecific passive transport within the plants.

After 6 weeks of growth, some nutrients were fully consumed, and the first mustard plants showed signs of deficiency in the form of chlorosis in young and old leaves. Mustard, forming the largest biomass, had also the largest demand for Ca (mean ~644 mg per container), Mg (~140 mg), P (~205 mg) and S (~209 mg). Fig. S1 in the supplement shows the temporal evolution of the other nutrient concentrations.

**3.3 Element and Si isotope budgets**

The biomass amounts, concentrations, and isotope compositions used to calculate element and Si isotope budgets are reported in Table S4. The element retrievals are shown in Table 3. All three species showed less than complete retrieval, with variable deficits between elements. For Si the retrieval amounted to between 83% (mustard) and 90% (wheat). For the other nutrients (Ca, Fe, K, Mg, P and S, see Table 3) the retrievals were between 70% and 110%. Sulphur in mustard was an exception, with a retrieval of only 50%, which we attribute to the loss of volatile S species during drying and charring, leading to the low retrieval (Blanck et al., 1938). The results for the Si isotope budget are shown in Table 4. Within uncertainty, there is no significant difference between the isotopic composition of the starting solution and the weighted average isotopic composition of the different compartments at the end of the experiment. Thus, we conclude that all significant pathways that fractionate Si isotopes are accounted for.

**3.4 Dynamics of isotope fractionation between the nutrient solution and plants**

The average initial $\delta^{30}Si$ composition of the nutrient solution is -0.21 ± 0.07 ‰ (2 s, relative to NBS28; individual results are reported in Table S3). The temporal evolution of the nutrient solution and the individual Si isotopic composition of the roots, shoots and the entire plants are shown in Figure 2 (reported as $\Delta^{30}Si$ relative to the nutrient solution). All three plant species preferentially incorporated the lighter silicon isotope ($^{28}Si$), leaving the nutrient solution enriched in heavier silicon ($^{30}Si$). After an initial lag phase for all three species, in which the nutrient solutions' Si isotope composition does not vary, its isotopic

composition becomes increasingly enriched in $^{30}$Si. Tomato and mustard, as rejective Si taxa, took up only about 10% of the Si predicted by water transpiration rates over the course of the experiment (Fig. 1; Table 2), such that the enrichment of the nutrient solution in $^{30}$Si was relatively small ($^{Tomato}\Delta^{30}Si_{Solution:End-Start}$=+0.13 ‰, $^{Mustard}\Delta^{30}Si_{Solution:End-Start}$=+0.19 ‰, calculated using Eq. (1)). As an Si accumulator, wheat incorporated almost all available Si within six weeks. The remaining Si is strongly enriched in $^{30}$Si ($^{Wheat}\Delta^{30}Si_{Solution:End-Start}$=+0.83 ‰). In week six one growth solution was so strongly depleted in Si that Si isotope ratios could not be determined.

Tomato plants incorporate light Si, where the bulk plant Si isotope composition, expressed as $^{Tomato}\Delta^{30}Si_{plants}$ averaged -0.27 ±0.06 ‰ ($^{Species}\Delta^{30}Si_{parts}$ are relative to the nutrient solution at the beginning, calculated using Eq. (2), and uncertainties are 95% CI). The Si present in the roots is isotopically indistinguishable from the nutrient solution ($^{Tomato}\Delta^{30}Si_{roots}$ = 0.01 ± 0.16 ‰), whereas the tomato shoots contain lighter Si ($^{Tomato}\Delta^{30}Si_{shoots}$ = -0.36 ±0.12 ‰). In contrast, mustard roots are lighter in their Si isotope composition ($^{Mustard}\Delta^{30}Si_{roots}$ = -0.77 ± 0.15 ‰) than the above-ground parts ($^{Mustard}\Delta^{30}Si_{shoots}$ = -0.05 ± 0.11 ‰). Nevertheless, mustard plants incorporated overall light Si ($^{Mustard}\Delta^{30}Si_{plants}$ = -0.45 ± 0.09 ‰). Since wheat consumed almost all available Si no significant fractionation between the plant and solution was observable ($^{Wheat}\Delta^{30}Si_{plants}$ = -0.07 ± 0.26 ‰). Most of the Si was deposited in the shoots, with an isotopic composition close to the composition of the starting solution ($^{Wheat}\Delta^{30}Si_{shoots}$ = -0.06 ± 0.26 ‰). The roots, however, preferentially stored light Si ($^{Wheat}\Delta^{30}Si_{roots}$ = -1.04 ± 0.34 ‰), similar to the mustard roots.

Our experimental setup allows us to determine the Si isotope fractionation factors into bulk plants directly from the temporal evolution of the Si isotope composition of the nutrient solution. This approach differs from previous studies of Si isotope fractionation by plants, in which the Si pool in the nutrient solution was frequently replenished (Ding et al., 2008a; Sun et al., 2008, 2016b). Evaluating the temporal evolution of wheat nutrient solution (Figure 3) and assuming no back-diffusion, a Rayleigh like fractionation can be fitted using Eq. (6) (Mariotti et al., 1981):

$$\frac{R}{R_0} = f_{solution}^{\alpha-1} \tag{6}$$

where $f_{solution}$ is the fraction of Si in the remaining solution, $R_0$ the initial $^{30}Si/^{28}Si$ isotope ratio, R the $^{30}Si/^{28}Si$ isotope ratio of the product, and $\alpha$ the fractionation factor. A best fit to the data, minimising the root-mean-square-deviation, results in $\alpha_{Plant-solution}$ for tomato of 0.99970 ($1000\cdot\ln(\alpha)$ = -0.33 ‰), for mustard an $\alpha_{Plant-solution}$ of 0.99945 ($1000\cdot\ln(\alpha)$ = -0.55 ‰), and for wheat an $\alpha_{Plant-solution}$ of 0.99957 ($1000\cdot\ln(\alpha)$ = -0.43 ‰), respectively (Figure 3). We use a Monte Carlo approach to estimate uncertainty on $\alpha_{Plant-solution}$, by calculating $\alpha_{Plant-solution}$ on 500 permutations of the dataset in which values for $\delta^{30}Si$ and Si concentration were randomly drawn from a normal distribution with means and standard deviations provided by the measurement (Table 5). Within uncertainty, there is no significant difference in the bulk fractionation factor between active and rejective uptake species. The best fit through all results, across the three plant species from this study, results in a fractionation factor $1000\cdot\ln(\alpha)$ of -0.41 ± 0.09 ‰ (1 s) at an initial Si concentration of 49.5 µg·g$^{-1}$ (ca. 1.76 mM).

If we assume the uptake of Si to be governed by diffusion through cell membranes and Si permeable transporters (Ma et al., 2006, 2007; Ma and Yamaji, 2015; Mitani et al., 2009; Zangi and Filella, 2012) and the diffusion of Si is non-quantitative, the lighter isotopes will be enriched in the target compartment (Sun et al., 2008; Weiss et al., 2004). To a first approximation, the difference between the diffusion coefficient of isotopologues $^{28}Si(OH)_4$ and $^{30}Si(OH)_4$ sets the theoretical upper limit of observable isotopic fractionation in a system dominated by diffusion. The diffusion coefficient ratio approximated by Eq. (7) corresponds to the fractionation factor in an idealised system consisting of pure water and silicic acid only (Mills and Harris, 1976; Richter et al., 2006).

$$\frac{D_{^{28}Si(OH)_4}}{D_{^{30}Si(OH)_4}} = \sqrt{\frac{\left(\frac{m_{^{30}Si(OH)_4} \times m_{H_2O}}{m_{^{30}Si(OH)_4} + m_{H_2O}}\right)}{\left(\frac{m_{^{28}Si(OH)_4} \times m_{H_2O}}{m_{^{28}Si(OH)_4} + m_{H_2O}}\right)}} \tag{7}$$

where D is the diffusion coefficient of a given Si molecule, and $m_{H_2O}, m_{^{28}Si(OH)_4}, m_{^{30}Si(OH)_4}$ are the molecular masses of the solvent (assuming pure water), $^{28}Si(OH)_4$ and $^{30}Si(OH)_4$, respectively. For $^{28}Si(OH)_4$ and $^{30}Si(OH)_4$ in pure water this results in a ratio of 0.99839 (1000·ln($\alpha$) = -1.61 ‰). The observed $\alpha_{Plant}$ is about four times smaller with 1000·ln($\alpha$) of -0.33 to -0.55‰. The theoretical diffusion coefficient exceeding the measured coefficient has been observed in other systems (e.g. O'Leary, 1984).

### 3.5 SEM-EDX analysis of mustard root phytoliths

Phytolith extraction revealed that considerable amounts of Si in the mustard roots are stored as phytoliths. The phytoliths observed were of elongated shape and consisted mainly of $SiO_2$ with some minor fraction carbon (~16 %), potassium (~4 %) and iron (~1 %) (see Fig. S2). The mechanisms of precipitation of the silicic acid in the mustard root remains unclear. The finding offers however an explanation for the isotopic difference between mustard, wheat, and tomato roots, since precipitation favours the incorporation of light $^{28}Si$. Si in mustard roots precipitates as biogenic silica, a process observed previously in wheat roots too (Hodson and Sangster, 1989), whereas tomato does not form root phytoliths.

## 4 Discussion

### 4.1 Reliability of the combined element and isotope ratio approach

In contrast to previous studies, we added a finite nutrient amount to growth solutions and replenished only the transpired water. The combination of plant physiological ratios (water use efficiency, element budgets and biomass production) with stable isotope ratio measurements allows us to explore the temporal evolution of Si uptake and translocation. Several aspects of our data attest to the reliability of our approach and results. Concerning Si uptake dynamics, Si recovery rates of >80% (see Table 3) corroborate the reliability of our results. The same is observed for the isotope budgets. There is no significant difference between the isotopic composition of the starting solution and the weighted average of the isotopic compositions of the different

compartments at the end (see Table 4). This implies all significant pathways that fractionate Si isotopes have been accounted for. The Si retrieval rate between 83 and 90% is likely not caused by a single systematic analytical uncertainty or unaccounted sink of Si, but rather a combination of container wall absorption (up to 0.1%), root washing procedure (up to 1%), the weekly sampling (up to 3.5%) and analytical uncertainties (up to 10%). As the initial concentration of Si at the onset of the experiment (49.5 µg/g) was slightly above the solubility limits of amorphous silica at 15-18 °C (44.2 – 47.1 µg/g), a fraction of the silicon could also have been lost to polymerisation and precipitation. Guttation (Joachimsmeier et al., 2012; Yamaji et al., 2008) and litter fall were not observed during the experiment. Even if guttation were present, no Si would be lost since under the experimental conditions the fluid would evaporate leaving silica on the shoots. Thus, silicic acid excreted by guttation is counted towards the Si amounts in the shoots.

## 4.2 Si uptake strategies

The ratio between measured Si uptake and the expected Si amount that would have entered the plant in a purely passive uptake mechanism (see 2.6.1, plant performance ratios) shows that wheat accumulates Si and mustard and tomato both reject Si (Figure 1 and Table 2). The accumulation of Si in wheat can be explained by the cooperation of an influx transporter (Lsi1-like) into the roots and the presumed presence of an efflux transporter (Lsi2-like) from the roots into the xylem. As closely related cereals have such transporters, we expect them to be present in wheat too (Ma and Yamaji, 2015). In rice, mutants with either defective Lsi1 or Lsi2 transporter lead to significantly lower Si accumulation (Köster et al., 2009). The direct comparison between both mutants revealed that Lsi1 carries a larger share of Si incorporation, thus a defective Lsi2 can partially be compensated (Köster et al., 2009). Our results show clear evidence that active, metabolism-driven processes or mechanisms must have been involved for wheat. The 2-fold excess of the expected amount of Si taken up cannot be explained by a passive mechanisms (e.g. Exley, 2015).

Our experiments show a striking similarity in Si uptake characteristics between mustard and tomato. Considering the differences in ontogenesis between the plant species, this may be a fortuitous coincidence. In particular, the relatively low temperatures may have inhibited the growth of the more thermophilic tomato, while the conditions were closer to optimal for mustard and summer wheat. Tomatoes have the genetic capacity to accumulate Si, since an orthologue of Lsi1 is present in the genes. An insertion in the amino acid sequence however, lead to a loss of the Si uptake functionality (Deshmukh et al., 2016, 2015), and thus tomato like mustard, rejects Si.

With our experimental approach we also detect significant differences between the crop species in Si transfer from the root to the shoot (Table 2). Wheat, which probably has a metabolically active efflux transporter (Lsi2-like) at the root-xylem interface, has the highest Si transfer efficiency per water mass ($49.3 \pm 8.4$ mg shoot Si·L$^{-1}$). The transfer efficiency for tomato is significantly higher than mustard ($3.5 \pm 0.4$, and $2.4 \pm 0.3$ mg shoot Si·L$^{-1}$, respectively), which is not readily explainable by differences in root Si efflux pathways since tomato does not contain the active efflux transporter orthologue Lsi2 while mustard

does (Ma & Yamaji, 2015; Sonah *et al.*, 2017). The remarkably high Si concentration and amounts in mustard roots, and thus the lower Si transfer efficiency of mustard can be explained by phytolith formation (see Fig. S2). A similar immobilization of silica in roots has already been observed in wheat (Hodson and Sangster, 1989) and other grasses (Paolicchi et al., 2019). Other possible reasons for this phenomenon will be discussed based on the results on Si isotope fractionation.

### 4.3 Dynamics of Si isotope fractionation during uptake

The plant performance parameters disclose two distinctly different Si uptake mechanisms: an active strategy in wheat, and a rejective strategy in tomato and mustard. Despite these different Si uptake mechanisms, we find preferential uptake of light Si isotopes observed in all three species with the average $1000 \cdot \ln(\alpha)$ of $-0.41 \pm 0.09$ ‰ (1 s). We can only speculate on the reasons for the plants preferring $^{28}Si$ over $^{30}Si$. Si is taken up (actively facilitated) through Si permeable channels (orthologues of Lsi1 in rice, maize and barley) and passively with the water flow. Nowhere along these pathways does a change in the coordination

sphere of silicic acid occur (Ma et al., 2006, 2007; Mitani et al., 2009) which could lead to the preferential incorporation of the heavy Si isotope in the fraction taken up. Thus we speculate that both pathways favour the light isotopologue because of its greater diffusion coefficient (Sun et al., 2008; Weiss et al., 2004), a process for which a predicted maximum isotope fractionation of $-1.6$ ‰ (based on Eq. (7)) is expected. While the processes of active and rejective Si uptake differ in the amounts of Si (per time, and root mass) taken up into the plants, we speculate that the physico-chemical processes governing

Si uptake, which induce the stable isotope fractionation, are identical at a given initial concentration in the nutrient solution.

Our new Si fractionation factors (tomato $-0.33$ ‰, and mustard $-0.55$ ‰) are the first to be reported for non-Si accumulator plants and together with wheat ($-0.43$ ‰) are similar to those measured in other Si accumulator species. These include rice: $-0.30$ ‰ (Sun et al., 2008), $-1.02 \pm 0.33$ ‰* (* indicates results recalculated from $^{29/28}Si$ to $^{30/28}Si$, Ding et al., 2005) and

$-0.79 \pm 0.07$ (Sun et al., 2016a); banana: $-0.77 \pm 0.21$ ‰* (Opfergelt et al., 2006) and $-0.68$ ‰* (Delvigne et al., 2009); and corn and wheat: $-1.00 \pm 0.31$ ‰* (Ziegler et al., 2005). The only positive fractionations for Si isotopes reported are by Y. Sun and co-workers (Sun et al., 2016b) for rice ($+0.38$ and $-0.32$ ‰) and cucumber ($+0.27$ and $+0.20$ ‰). Previous experiments with the same rice species by L. Sun *et al.* however yielded a fractionation factor of $-0.30$ ‰ (Sun et al., 2008). The authors speculate that an active uptake mechanism preferentially incorporates heavy Si isotopes – a hypothesis that is not supported

by our results, or that the different fractionation factors "could also be also be affected by the silicon isotopic composition fluctuations in different batches of nutrient solutions caused by the frequent replacement" (Sun et al., 2016b). Excluding these positive fractionation factors the range found for all published bulk plant Si isotope fractionation factors ($-0.32$ to $-1.02$ ‰) is larger than that determined in our study ($-0.33$ to $-0.55$ ‰). These differences can arise from differences in species, chosen experimental conditions such as concentration of nutrient solution or temperature in the experiments.

## 4.4 Silicon fractionation between the roots and shoots

The presence or absence of the efflux (Lsi2-like metabolically active) transporter allows to explore its influence on isotope fractionation in the root and during further transport. (1) If Lsi2 has a similar functionality as Lsi1, a preference for the light $^{28}$Si as caused by diffusion should emerge which would be indistinguishable from the passive diffusion in the absence of Lsi2. (2) Alternatively, the presence of Lsi2 could also induce equilibrium isotope fractionation during a change in the speciation of silicic acid, causing the preferential transport of either $^{28}$Si or $^{30}$Si. (3) The third possibility are indirect effects in the roots such as precipitation of silicic acid in the roots which enrich the remaining silicic acid which is transported into the shoots in heavy $^{30}$Si.

The three crop species show large differences in their root Si isotopic composition. Mustard and spring wheat preferentially store light $^{28}$Si in their roots ($^{Mustard}\Delta^{30}Si_{roots}$ -0.77 $\pm$ 0.15 ‰, $^{Wheat}\Delta^{30}Si_{roots}$ -1.04 $\pm$ 0.34 ‰, relative to the nutrient solution) whereas tomato does not show a preference for either the lighter or heavier silicon isotopes ($^{Tomato}\Delta^{30}Si_{roots}$ -0.01 $\pm$ 0.16 ‰). The further transport of Si from the roots into the xylem seems not be driven by a diffusion process through Lsi2. Thus, hypothesis (1), that Lsi2 has a similar functionality as Lsi1 and transports Si in a diffusive process, is not likely. For mustard and wheat orthologues of Lsi2 have been shown to be involved in the Si transport (Deshmukh et al., 2016; Sonah et al., 2017). The current understanding of the molecular functionality of Lsi2, however, does provide not sufficient evidence for an equilibrium process where preferential transport of $^{30}$Si over $^{28}$Si into the xylem would be expected (hypothesis 2).

The isotopic difference between the Si in the shoots and in the roots ($^{30}\Delta_{Root-Shoot}$) for mustard and wheat amounts to -0.72 and -0.98 ‰, respectively, and can be explained by Si precipitation in the roots. Indeed, we observed mustard root phytoliths (Fig. S2). Mineral deposition in wheat roots has also been observed by Hodson & Sangster, (1989), supporting hypothesis (3). Precipitation of biogenic silica in the root would enrich the residual mobile silicon pool in heavy $^{30}$Si, which is then transported into the shoots. Köster *et al.*, 2009, showed that rice mutants with a defective Lsi2 lead to an additional (compared to non-mutants) preferential transport of heavy $^{30}$Si into the straw. This could be explained by an oversaturation in the roots due to the missing efflux transporter (Lsi2), leading to additional biogenic silica precipitation in the roots. The positive $^{30}\Delta_{Root-Shoot}$ of +0.37 ‰ for tomato, where Lsi2 is absent, indicate that the pool of Si in the roots was depleted in $^{28}$Si by a preferential diffusion process of the lighter isotope.

Within the shoots, Si is not homogenously distributed. Several researchers have observed an enrichment of $^{30}$Si along the transpiration stream (Ding et al., 2005; Hodson et al., 2008; Sun et al., 2016b), compatible with a Rayleigh-like fractionation within the shoots. A possible explanation for this observation is the formation of phytoliths. Early in the transpiration stream, the kinetically controlled condensation of silicic acid leads to the preferential incorporation of $^{28}$Si into phytoliths (e.g. Frick

*et al.*, 2019), whereas the remaining silicic acid in the fluid is enriched in $^{30}$Si and further transported along the transpiration stream.

## 5 Conclusion

The amount of Si uptake into crop plants and the distribution of Si within them is species-specific, and the relative contributions from different uptake strategies varies. For all three species analysed here, the measured uptake deviates from that expected if Si was simply taken up passively with transpired water. Instead, the 2-fold excess in uptake observed for wheat suggests involvement of an active, metabolism-driven mechanism.

Regardless of uptake strategy (active or rejective) all three crop species studied preferentially incorporate light silicon ($^{28}$Si) with fractionation factors $1000 \cdot \ln(\alpha)$ for tomato (-0.33 ‰), mustard (-0.55 ‰) and wheat (-0.43 ‰) being indistinguishable within uncertainty. This similarity indicates that the physico-chemical processes governing Si uptake, whether active or passive, or with Lsi1-like transporters present or absent, are identical. The incorporation and fractionation of stable Si isotopes at the root cortex is likely governed by the preferential diffusion of the lighter homologue of silicic acid. In contrast, at the root

endodermis, for species with the Lsi2-like transporter (wheat and mustard), the further transport of silicic acid from the roots into the xylem and shoots is not controlled by the preferential diffusion of light $^{28}$Si. Rather the precipitation of $^{28}$Si-enriched biogenic silica in the roots governs the isotope composition of remaining Si transported into and deposited within the shoots. For plant species that do not precipitate biogenic silica in the roots, further transport is governed by diffusion, in which $^{28}$Si is preferentially transported into the shoots.

The results presented here improve our understanding of Si uptake dynamics. By future integration of these stable isotope-based methods with biochemical and molecular genetic methods a more comprehensive model of Si uptake and regulation in plant species could be obtained. For a mechanistic understanding of isotope fractionation during transport of silicic acid and precipitation of biogenic silica, the bio-molecular processes involved in the dehydration of silicic acid and its conversion into

amorphous silica are required (He et al., 2015; Leng et al., 2009). To this end, isotope-spiking during plant growth and ripening may prove valuable, both in elucidating the fluxes of silicic acid between different pools and sources, and in fingerprinting the biochemical processes involved via their associated stable isotope fractionation.

The merits of such advanced understanding of Si biochemistry are potentially large. The continuous removal of Si-rich crop

residues from croplands with increasing agricultural activity may eventually result in Si deficits in soils (Carey and Fulweiler, 2016). Such shortages have the potential to reduce crop yields, and thus global food security (Cooke et al., 2016; Epstein, 1999). Developing the ability to track Si availability in soils and its recycling from plants may help in tackling this upcoming problem and developing strategies for more efficient use of plant-available Si in agricultural production.

**Author Contribution**

All authors designed the study, D. A. F. and R. R. have grown the plants, D. A. F. has analysed the samples and evaluated the data, prepared the figures. D. K. has prepared and imaged the phytoliths. All authors have contributed to the discussion, interpretation and writing of this manuscript.

**Competing interests**

The authors declare that they have no conflict of interest.

**Acknowledgments**

D.A.F. thanks the Swiss National Science Foundation for an Early Postdoc Mobility fellowship (P2EZP2_168836), the GFZ German Research Centre for Geosciences, Helmholtz Laboratory for the Geochemistry of the Earth Surface (HELGES) and the Leibniz Centre for Agricultural Landscape Research for providing excellent (laboratory) infrastructure. The authors would like to thank Christian Buhtz for taking care of the plants during the hydroponic experiments, and C. Delvigne, M. Hodson and two anonymous reviewers for their reviews and improvements during the discussion phase on Biogeosciences. D.A.F. would like to thank Josefine Buhk and Jutta Schlegel for their support in HELGES.

**Data availability**

All data used in this study are available in the supplementary, containing the tables S1– S6.

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

**Figures**

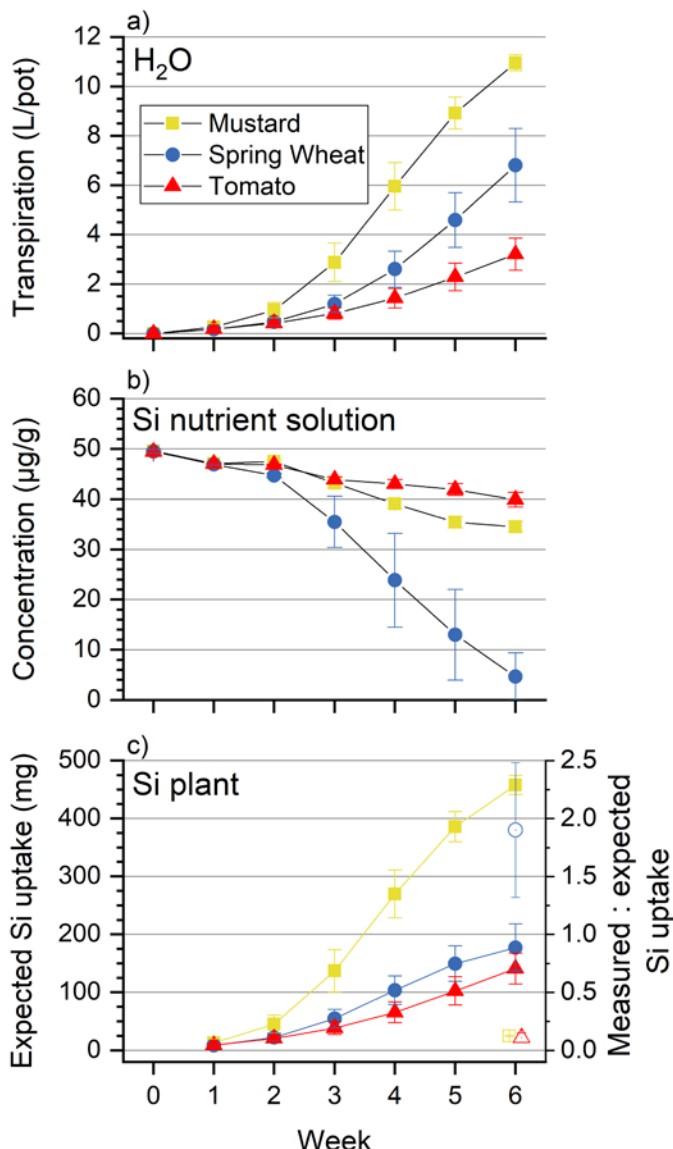

Figure 1: Cumulative transpiration (panel a), Si concentration in the nutrient solution (in µg/g, panel b) and the expected Si uptake through transpiration of tomato, mustard and spring wheat during 6 weeks (panel c). Shown is the mean ± standard deviation from 3 pots with 4 plants each. In panel c) a ratio of measured and expected Si uptake (open symbols) of greater than 1 indicates an active uptake mechanism, a ratio much smaller than 1 a rejective strategy.

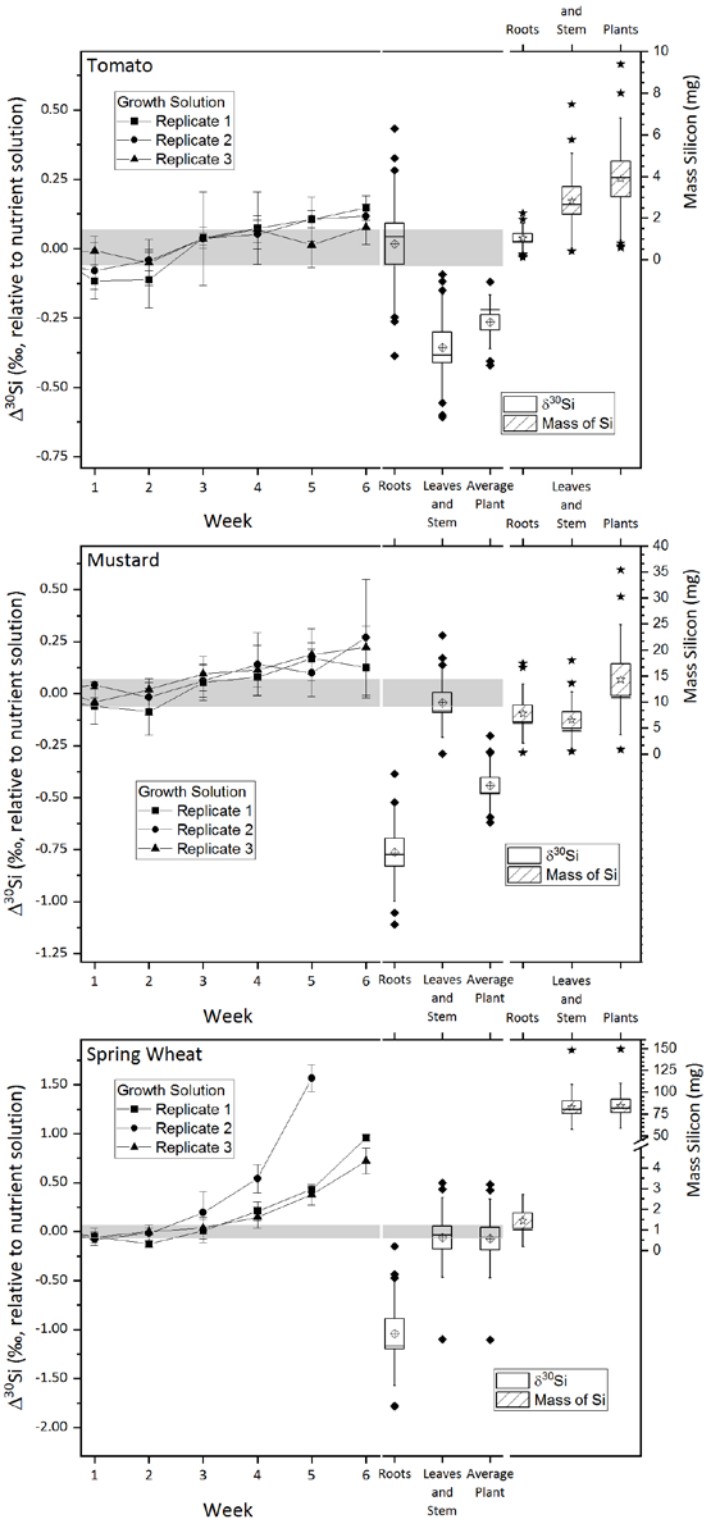

**Figure 2:** Silicon isotope composition (left) and mass of silicon taken up (right) during the growth of tomato, mustard, and wheat. The left y-axis shows the $\delta^{30}Si$ in ‰ relative to the nutrient solution, the right y-axis the mass of silicon incorporated by the plants in mg incorporated. The line connects $\delta^{30}Si$ from the weekly sampled nutrient solution (week 1 to 6). The box plots denote $\delta^{30}Si$ (left) and plant organ Mg mass (right), per species 12 roots and 12 leaves and stem samples were analysed, plant averages were weighted by organ mass (calculated using Eq. (2). Uncertainty bars are based on 2 standard uncertainties, grey area denotes the silicon isotopic composition of the starting solution ± two standard deviations. All box sizes denote one standard uncertainty, whisker indicate one standard deviation, horizontal line in the box shows the median, empty diamond/stars in the box indicate the mean and filed diamonds/stars show outliers, outside of one standard deviation.

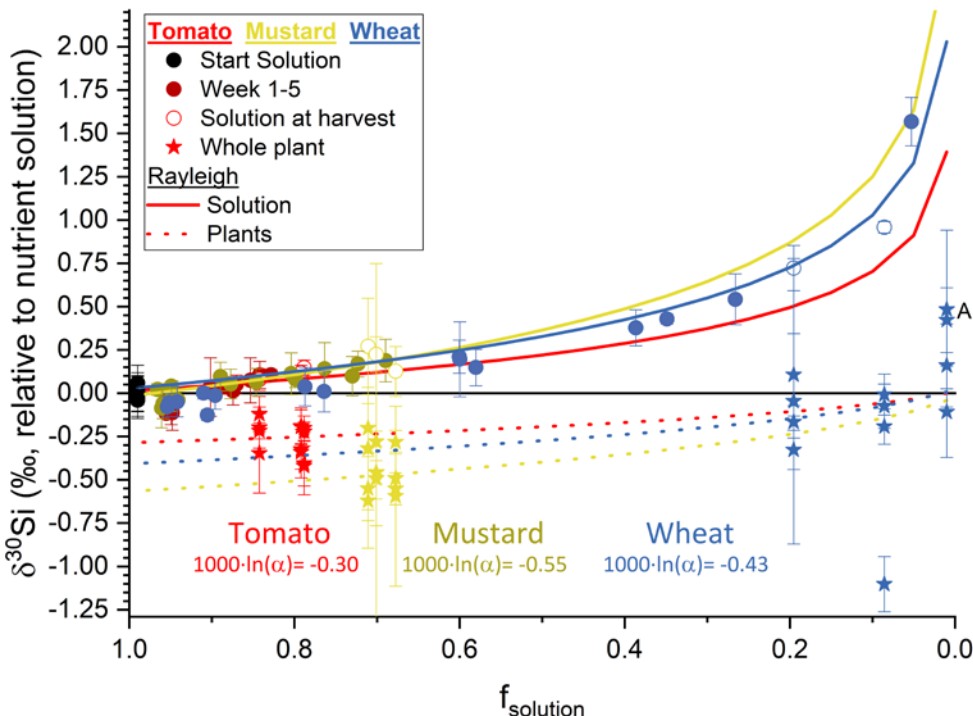

**Figure 3: The silicon isotope composition (expressed in δ³⁰Si ‰ relative to nutrient solution) versus the amount of silicon taken up by the plants (expressed as dimensionless f_solution) (circles represents the nutrient solution, tomato in red, mustard in yellow and wheat in blue, starting solutions in black). Red, yellow and blue solid lines represent the best fit through a Rayleigh-like fractionation for the remaining solution, the dotted line the accumulated silicon isotope composition in the plants derived thereof. Stars are the mass-weighted average isotopic composition of the individual plants at the respective f_solution of the container at harvest. Plant samples denoted with A have no corresponding solution value, since the concentration of silicon was below the amount required for an isotope ratio determination. Uncertainty bars are based on two standard deviations.**

**Tables**

| Parameter | | Plant species | | |
|---|---|---|---|---|
| | | Mustard | Wheat | Tomato |
| Dry matter [g pot⁻¹] | Root | $3.9 \pm 1.1$ | $2.6 \pm 0.6$ | $1.7 \pm 0.2$ |
| | Shoot | $25.0 \pm 4.2$ | $13.7 \pm 2.0$ | $10.3 \pm 1.5$ |
| | Total plant | $29.0 \pm 5.2$ | $16.3 \pm 2.5$ | $12.0 \pm 1.7$ |
| Plant Si content [mg Si g⁻¹ dry matter] | Root | $8.6 \pm 4.3$ | $2.5 \pm 2.8$ | $3.5 \pm 1.8$ |
| | Shoot | $1.0 \pm 0.3$ | $24.2 \pm 6.3$ | $1.4 \pm 0.7$ |
| | Total plant | $2.0 \pm 0.4$ | $20.9 \pm 4.0$ | $1.3 \pm 0.2$ |
| Plant Si uptake [mg Si pot⁻¹] | Root | $31.1 \pm 4.8$ | $5.8 \pm 3.1$ | $4.1 \pm 1.3$ |
| | Shoot | $26.1 \pm 3.8$ | $331.3 \pm 70.1$ | $11.4 \pm 3.6$ |
| | Total plant | $57.2 \pm 1.3$ | $337.0 \pm 67.9$ | $15.5 \pm 4.9$ |
| Transpiration [L pot⁻¹] | Pot | $11.0 \pm 0.3$ | $6.8 \pm 1.5$ | $3.2 \pm 0.6$ |

**Table 1: Dry matter, plant Si content, plant Si uptake and water transpiration of mustard, wheat and tomato after 6 weeks (hydroponic culture; mean ± standard deviation based on 3 pots with 4 plants each).**

| Quotient | Plant species | | |
|---|---|---|---|
| | Mustard | Wheat | Tomato |
| Dry mass ratio [g shoot g⁻¹ root] | $6.5 \pm 0.7$ | $5.4 \pm 0.9$ | $5.9 \pm 0.2$ |
| Si mass ratio [mg Si in shoot mg⁻¹ Si in root] | $0.9 \pm 0.2$ | $72.7 \pm 47.8$ | $2.7 \pm 0.2$ |
| Water use efficiency [g L⁻¹] | $2.6 \pm 0.5$ | $2.4 \pm 0.2$ | $3.8 \pm 0.3$ |
| Si uptake efficiency [mg plant Si L⁻¹] | $5.2 \pm 0.3$ | $50.3 \pm 8.8$ | $4.8 \pm 0.6$ |
| Si transfer efficiency [mg shoot Si L⁻¹] | $2.4 \pm 0.3$ | $49.3 \pm 8.4$ | $3.5 \pm 0.4$ |
| Uptake classification (measured / expected Si uptake) | $0.12 \pm 0.01$ | $1.9 \pm 0.6$ | $0.11 \pm 0.04$ |

**Table 2: Ecophysiological performance ratios for mustard, wheat and tomato (means ± standard deviation based on 3 pots with 4 plants each). The uptake classification is based on the ratio of measured and expected Si uptake. A ratio of greater than 1 indicates an active uptake mechanism, a ratio much smaller than 1 a rejective strategy and a ratio of 1 is passive uptake.**

| | [mg] | Mustard | | | Wheat | | | Tomato | | |
|---|---|---|---|---|---|---|---|---|---|---|
| | | Pot 1 | Pot 4 | Pot 7 | Pot 2 | Pot 5 | Pot 8 | Pot 3 | Pot 6 | Pot 9 |
| **Si** | $m_{Start}$ | 418 | 421 | 399 | 425 | 416 | 411 | 418 | 415 | 414 |
| | $m_{End}$ | 283 | 299 | 280 | 36 | 2 | 80 | 329 | 329 | 349 |
| | $m_{Plants}$ | 58 | 56 | 58 | 299 | 415 | 297 | 20 | 15 | 11 |
| | Retrieval | 82% | 84% | 85% | 79% | 100% | 92% | 84% | 83% | 87% |
| **Ca** | $m_{Start}$ | 544 | 543 | 524 | 548 | 542 | 541 | 549 | 542 | 543 |
| | $m_{End}$ | 3 | 0 | 0 | 382 | 376 | 423 | 139 | 182 | 264 |
| | $m_{Plants}$ | 393 | 394 | 352 | 108 | 119 | 87 | 304 | 241 | 222 |
| | Retrieval | 73% | 73% | 67% | 89% | 91% | 94% | 81% | 78% | 90% |
| **Fe** | $m_{Start}$ | 39 | 39 | 38 | 39 | 40 | 39 | 39 | 39 | 39 |
| | $m_{End}$ | 26 | 29 | 28 | 27 | 25 | 28 | 24 | 24 | 28 |
| | $m_{Plants}$ | 4 | 4 | 3 | 6 | 4 | 3 | 5 | 5 | 2 |
| | Retrieval | 76% | 85% | 82% | 85% | 73% | 80% | 73% | 75% | 78% |
| **K** | $m_{Start}$ | 1787 | 1813 | 1742 | 1817 | 1801 | 1801 | 1803 | 1809 | 1801 |
| | $m_{End}$ | 657 | 424 | 174 | 539 | 505 | 787 | 941 | 1044 | 1213 |
| | $m_{Plants}$ | 1085 | 1218 | 1500 | 1556 | 1449 | 979 | 872 | 727 | 673 |
| | Retrieval | 98% | 91% | 96% | 115% | 109% | 98% | 101% | 98% | 105% |
| **Mg** | $m_{Start}$ | 121 | 121 | 116 | 122 | 120 | 119 | 122 | 121 | 120 |
| | $m_{End}$ | 7 | 1 | 0 | 63 | 59 | 67 | 35 | 41 | 55 |
| | $m_{Plants}$ | 82 | 95 | 73 | 30 | 26 | 27 | 52 | 55 | 33 |
| | Retrieval | 74% | 79% | 63% | 76% | 70% | 80% | 72% | 79% | 74% |
| **P** | $m_{Start}$ | 173 | 176 | 171 | 177 | 175 | 176 | 176 | 177 | 177 |
| | $m_{End}$ | 5 | 2 | 1 | 0 | 0 | 11 | 5 | 20 | 52 |
| | $m_{Plants}$ | 121 | 134 | 115 | 137 | 142 | 144 | 117 | 123 | 82 |
| | Retrieval | 73% | 77% | 68% | 77% | 81% | 88% | 69% | 81% | 76% |
| **S** | $m_{Start}$ | 180 | 183 | 174 | 182 | 182 | 182 | 183 | 182 | 182 |
| | $m_{End}$ | 4 | 3 | 6 | 97 | 101 | 119 | 81 | 89 | 113 |
| | $m_{Plants}$ | 95 | 88 | 73 | 61 | 57 | 33 | 60 | 55 | 38 |
| | Retrieval | 55% | 50% | 45% | 87% | 87% | 84% | 77% | 79% | 83% |

Table 3: Major element budget for mustard, tomato and wheat. $m_{Plants}$ is calculated based on the concentration of the element in the plant digest and the dry mass, the $m_{Start}$ $m_{End}$ are the element masses in mg based on the amount of nutrient solution and the element concentration at the start and the end of the experiment. Retrieval is the ratio between $m_{Start}$ and the sum of $m_{Plants}$ and $m_{End}$. The initial amount of the elements in the seeds, taken up during germination and the amount of element discharged in the wash water are not considered.

|  | $\delta^{30}$Si | 2 s | $\delta^{30}$Si | 2 s | $\delta^{30}$Si | 2 s |
|---|---|---|---|---|---|---|
| **Mustard** | | | | | | |
| | Pot 1 | | Pot 4 | | Pot 7 | |
| **Start** | -0.23 | 0.12 | -0.19 | 0.06 | -0.15 | 0.06 |
| **End** | -0.20 | 0.30 | -0.04 | 0.38 | -0.09 | 0.26 |
| **Wheat** | | | | | | |
| | Pot 2 | | Pot 5 | | Pot 9 | |
| **Start** | -0.18 | 0.03 | -0.18 | 0.13 | -0.24 | 0.07 |
| **End** | -0.39 | 0.30 | 0.05 | 0.23 | -0.12 | 0.27 |
| **Tomato** | | | | | | |
| | Pot 3 | | Pot 6 | | Pot 9 | |
| **Start** | -0.20 | 0.08 | -0.25 | 0.10 | -0.23 | 0.02 |
| **End** | -0.09 | 0.19 | -0.11 | 0.31 | -0.14 | 0.31 |

**Table 4: Silicon isotope budget (calculated using Eq. (5)) for mustard, wheat and tomato at the start of the experiment (based on the isotopic composition of the nutrient solution) and the end (based on the plants and nutrient solution isotopic composition).**

| best fit | Mustard | Tomato | Wheat | All data |
|---|---|---|---|---|
| **$1000 \cdot \ln(\alpha)$ [‰]** | -0.55 ± 0.40 | -0.33 ± 0.32 | -0.43 ± 0.09 | -0.43 ± 0.09 |

**Table 5: $^{30}$Si/$^{28}$Si isotope fractionation factor $1000 \cdot \ln(\alpha)$ numerically approximated by reducing root-mean-square-deviation ('best fit') using Eq. (6) and uncertainties (1 s) from Monte Carlo method with n=500 seeded individual data sets.**