# Peer review of "Silicon uptake and isotope fractionation dynamics by crop species"

_Biogeosciences, 2020_

## Referee Comment (RC1) · Anonymous Referee #2 · 20 Apr 2020

[referee-annotated manuscript omitted]

---

## Referee Comment (RC2) · Martin Hodson (Referee) · 23 Apr 2020

This paper represents an interesting investigation into Si isotope fractionation in three contrasting crop plants. I am not aware that anyone has taken this approach before. I have seen that another referee has concentrated on the methodology, and I will not go over these points again. Rather I will look mostly at the interpretation of the results, and give some suggestions for improvements in the discussion.

Major Points

Line 12 and elsewhere. I am not sure that I would use "a variety of strategies (rejective,

passive and active)." As we have come to understand Si uptake by plants it has become obvious that the different species form a spectrum. You mentioned Hodson et al. (2005) and the spectrum is very evident there. I would just say that you took species that take up Si to different extents.

Line 22. Not always at the endodermis (rice)- some species have much more dispersed transporters in the root.

Line 24 and elsewhere. The finding of significant biogenic silica deposition in the roots of mustard is novel. As far as I am aware it is the first time in a non-woody dicot. The only dicot mentioned in the recent review of silicification of roots by Lux et al. (2020) is beech. I don't think you can really just say "unpublished observations". We need to know more about this- is it endodermal deposition? A picture would help.

Line 95 onwards: Sun et al. (2019) found that while there are two Si transporter homologues present in tomato (SILsi1, a homologue of the rice LSi1 influx transporter; and SILsi2, a homologue of the rice LSi2 efflux transporter), only SILsi1 is active. They suggest that the absence of active SILsi2 explains the low levels of Si accumulation in this species.

Line 321 onwards. As already stated phytoliths in the mustard root is a novel finding, and "data not shown" is not really good enough.

Line 340 and elsewhere: I really don't like reviewers that try to increase their citations by recommending their own papers! However, there are some cases where this is justified. I am very surprised that you did not mention the work of Hodson et al. (2008) on Si isotopes in wheat. Our plants were grown in soil to maturity, and so it was a different setup. But one thing is very clear: there is significant fractionation within the wheat shoot. This does not invalidate your results, but it should be noted (our culm d30Si is negative, but leaf sheaths and blades are positive leading to a positive value overall). The second point is that we also found that the lighter isotopes were deposited first. We could not measure Si in the roots because of soil contamination, but we said

"It is apparent that there are two main routes for Si transport within the wheat plant, and that heavier isotopes increase towards the end of both routes: (1) culm » leaf sheath » leaf blade; and (2) culm » rachis » inflorescence bracts. A similar pattern was reported by Ding et al. (2005) working on rice. They considered that the process of Rayleigh fractionation explained the accumulation of heavy isotopes in the upper parts of the plant. Essentially, this would involve the lighter 28Si isotope being more reactive, and thus more likely to be deposited in phytoliths. Thus, in wheat, proportionately more 28Si isotope would be deposited in the culm phytoliths, and a greater proportion of 30Si and 29Si would continue in the transpiration stream to the leaf sheath. In the sheath the same fractionation occurs, leading to an even greater concentration of heavier isotopes in the leaf blade." This is exactly the same process that you postulate is happening in the wheat roots before Si flows on to the shoots. So our work confirms your ideas in section 4.4.

Minor correction Line 43 Yan

References

Ding TP, Ma GR, Shui MX, Wan DF, Li RH. 2005. Silicon isotope study on rice plants from the Zhejiang province, China. Chem. Geol. 218: 41–50.

Hodson MJ, White PJ, Mead A, Broadley MR. 2005. Phylogenetic variation in the silicon composition of plants. Ann. Bot. 96: 1027–1046.

Hodson MJ, Parker AG, Leng MJ, Sloane HJ. 2008. Silicon, oxygen and carbon isotope composition of wheat (Triticum aestivum L.) phytoliths: implications for palaeoecology and archaeology. J. Quaternary Sci. 23: 331–339.

Lux A, Lukačová Z, Vaculík M, Švubová R, Kohanová J, Soukup M, Martinka M, Bokor B. 2020. Silicification of root tissues. Plants 2020, 9, 111. https://doi.org/10.3390/plants9010111

Sun H, Duan Y, Mitani-Uneo N, Che J, Jia J, Liu J, Guo J, Ma JF, Gong H. 2019.

[Figure]

Tomato roots have a functional silicon influx transporter but not a functional silicon efflux transporter. New Phytol. doi:10.1111/pce.13679

---

## Author Comment (AC1) · 23 Apr 2020

Dear Anonymous Referee #2

Thanks for taking your time to review our manuscript.

Regarding your immediate question how we assessed the water uptake/transpiration: The pots were weighted weekly without the lid and plants, using a balance. The weight difference to the previous week is reported as volume taken up by the plants, assuming a density of 1 g/mL. We replenished the pots by filling up with ultra-pure water to the weight from the previous week. The pots were closed with a lid, and we thus neglect

evaporation. The term transpiration is thus referred to the water taken up, which is either lost by transpiration and guttation or stored in the biomass. Based on previous reports (e.g. Joachimsmeier et al., 2012) the amount of fluid lost through guttation, was considered negligible during the course of the experiment.

Best regards, Daniel A. Frick

Reference: Joachimsmeier, I., Pistorius, J., Heimbach, U., Schenke, D., Kirchner, W. and Zwerger, P.: Frequency and intensity of guttation events in different crops in Germany, in 11th International Symposium of the ICP-BR Bee Protection Group, Wageningen (The Netherlands), November 2-4, 2011, vol. 437, pp. 87–90., 2012.

―――――――――――――――――

---

## Author Comment (AC2) · 6 May 2020

**Dear Anonymous Referee #2**

Thanks for taking your time to review our manuscript. We have considered your suggestion and have further clarified the materials and methods section.

In detail we provide our answer to your questions and suggestions:

**Line 89: What is this? A somewhat unconventional unit. Do you mean &
Line 90: Do you mean 49.5 mg/L? Is this the concentration of Si or the salt?**

µg/g is a SI unit for concentration. Our measurements are based on weighing the solutions, thus we report the concentration as 'per g' and not as 'per mL'. For the convenience of the reader we expanded the sentence and provide the concentration in mM and specified that the concentration refers to Si:

"Silicon was added in the form of $NaSiO_4$ to an initial Si starting concentration of 49.5 $\mu g \cdot g^{-1}$ (1.76 mM). Ultrapure water (resistivity 18.2 $MOhm \cdot cm$) was used to prepare the nutrient solutions and to weekly restock water taken up by the plants. Detail composition can be found in supplementary methods S1."

**Line 94: What does this mean? How can they 'reject' silicic acid? &**
**Line 94: Active uptake of silicic acid? Where is the evidence that this occurs?**

Active, passive and rejective Si uptake is a concept which has been proposed by several groups: see e.g. (Hodson et al., 2005; Takahashi et al., 1990) and also the review by M. Hodson for this manuscript: https://www.biogeosciences-discuss.net/bg-2020-66/bg-2020-66-RC2.pdf). The classification is based on the amount of silicon is taken up in relation to the water uptake and is also explained in Line 171ff. As also M. Hodson remarked in his review, the uptake of Si is not a strict classification, but a spectrum which allows to qualitatively describe the Si uptake.

**Line 99: Really, so silicic acid does not follow water into either mustard or tomato? Do you have evidence to support this?**

This is not what has been stated in the text. We justify the selection of the plant species and provide information which additional transporter channels / proteins are present in the investigated plants.

**Line 102: Added Si, but how much Si was present in these solutions?**

The amount of Si introduced by the other nutrient salts and the water was not resolvable using the ICP-OES, thus we considered these negligible. We have changed the sentence to:

"Plant seeds were germinated in Petri dishes with half-strength nutrient solution used for the later growth experiment that contained no added $NaSiO_4$. "

**Line 104: What about the significant increase in sodium content, did you have a control for this?**

We did not counterbalance or remove the Na which has been introduced by the addition of $NaSiO_4$.

**Line 105: How did you measure the volume of transpired water?**

The pots were weighted weekly without the lid and plants, using a balance. The weight difference to the previous week is reported as volume taken up by the plants, assuming a density of 1 g/mL. We replenished the pots by filling up with ultra-pure water to the weight from the previous week. The pots were closed with a lid, and we thus neglect evaporation. The term transpiration is thus referred to the water taken up, which is either lost by transpiration and guttation or stored in the biomass. Based on previous reports (e.g. Joachimsmeier et al., 2012) the amount of fluid lost through guttation, was considered negligible during the course of the experiment. We have added this information:
"Each week the pots were weighted without the lid and the plants, and the mass of transpired water was replenished with ultrapure water (18.2 $MOhm \cdot cm$). The weight difference to the previous week is considered the mass of water taken up the plants. The pots were closed with a lid, and we thus neglect evaporation."

**Line 112: What about other forms of water loss such as guttation?**

See question before. We considered water loss through guttation negligible. We have rephrased the sentence:
"For sampling, 40 mL were taken after replenishing water loss due to plant uptake and mixing of the solution."

**Line 115: What kind of extracellular Si deposits? Do you simply mean that you**

**washed off the nutrient solution?**

Thanks for bringing this to our attention, we have clarified the sentence:
"The roots were immersed multiple times in ultrapure water to remove potential extracellular Si deposits and adhered nutrients."

**Line 118: how?**

We have added a link to chapter 2.5.2 where the digestion procedure is explained.

**Line 123: Why are all essential details of methods in Supplementary files, they need to be here in M&M.**

We have expanded the section and explained how we have performed the concentration measurements by ICP-OES:
"Samples and standard were analysed following a procedure by Schuessler et al., 2016, briefly the samples and standards were doped with an excess of $CsNO_3(1mg \cdot g^{-1})$ to reduce matrix effects that are likely to be caused from the high nitrogen content of the samples and quantified applying an external calibration. The relative analytical uncertainties are estimated to be below 10% and agreed with the nominal concentration of the starting solutions."

**Line 125: What do you mean? How do you know that the aliquot contains this amount of Si? Where are the methods?**

The concentration is known from the measurement by ICP-OES, we have clarified this part:
"Based on the concentration measured, an aliquot of each nutrient solution containing approximately 1000 μg Si was dried down in silver crucibles on a hotplate at 80-95 °C."

**Line 131: estimates based upon what?**

The concentration was estimated by analysing an exploratory experiments, we have clarified this:

"50-800 mg of plant material, depending on the Si concentration determined in an exploratory study, was weighed into Ag crucibles and combusted overnight (2h at 200 °C, 4h at 600 °C, then cooled to room temperature) in a furnace (LVT 5/11/P330, Nabertherm)."

**Line 133: what does this mean?**

We removed this information since the results were not presented in this study.

**Line 134: what is the Si content of this salt?**

We have specified what the Si content of NaOH was:
"After cooling 400 mg NaOH (TraceSELECT, Sigma-Aldrich, checked for low Si blank levels) added."

**Line 137: Does plant silica dissolve under these conditions?**

The high temperature fusion of silicates, silicon, and bio silica (e.g. diatoms, phytoliths) using NaOH has been proven to be quantitative. The silicate is transformed in this fusion into its silicic form which can be dissolved in water.

**Line 138: How? You convert Si to a cation? You need to fully explain these methods.**

The dissolution procedure of silicates, silicon and bio silica is state of the art in geosciences. Si is present in $SiO_2$ as $Si^{4+}$, counterbalanced by 2 $O^{2-}$. Therefore, we do not need to convert Si into a cation. The NaOH accelerates the dissolution of the oxide, and after the addition of water silicon is present as silicic acid ($H_4SiO_4$ and depending on the pH also in the form of $H_3SiO_4^-$ (see e.g. Stamm et al., 2019, their Fig. 1 for an aqueous Si species in equilibrium diagram).

We hesitate to include the entire Supplementary Method S3 into the main text, but if both reviewers ask, we will gladly do this.

**Line 140: Again, the methods should be here and not in Supplementary files.**

We have clarified that in the supplementary files a step-by-step procedure can be found. We hesitate to include the entire Supplementary Method S3 into the main text, but if both reviewers ask, we will gladly do this.

**Line 161: It would seem that all measurements rely upon accurate measurements of water intake. Where have you written about how you measured the amount of transpired water? Why do you assume that all water uptake is reflected by this transpired volume? Again, what about processes like gutation. Even if your measurements of transpiration are accurate, they do not represent water uptake into the plant.**

See response to your question on Line 105.

We hope that these answers clarify your immediate questions and invite you to continue the review of our manuscript.

Best regards,
Daniel A. Frick

**Literature:**

Hodson, M. J., White, P. J., Mead, A. and Broadley, M. R.: Phylogenetic variation in the silicon composition of plants, Ann. Bot., 96(6), 1027–1046, doi:10.1093/aob/mci255, 2005.

Joachimsmeier, I., Pistorius, J., Heimbach, U., Schenke, D., Kirchner, W. and Zwerger, P.: Frequency and intensity of guttation events in different crops in Germany, in 11th International Symposium of the ICP-BR Bee Protection Group, Wageningen (The Netherlands), November 2-4, 2011, vol. 437, pp. 87–90., 2012.

Schuessler, J. A., Kämpf, H., Koch, U. and Alawi, M.: Earthquake impact on iron isotope signatures recorded in mineral spring water, J. Geophys. Res. Solid Earth, 121(12), 8548–8568, doi:10.1002/2016JB013408, 2016.

Stamm, F. M., Zambardi, T., Chmeleff, J., Schott, J., von Blanckenburg, F. and Oelkers, E. H.: The experimental determination of equilibrium Si isotope fractionation factors among H4SiO4o, H3SiO4− and amorphous silica (SiO2Åů0.32 H2O) at 25 and 75 °C using the three-isotope method, Geochim.  Cosmochim.  Acta, 255, 49–68, doi:10.1016/j.gca.2019.03.035, 2019.

Takahashi, E., Ma, J. F. and Miyake, Y.: The possibility of silicon as an essential element for higher plants, Comments Agric. Food Chem., 2(2), 99–102, 1990.

---

## Referee Comment (RC3) · Camille Delvigne (Referee) · 7 May 2020

The manuscript " Silicon isotope fractionation and uptake dynamics of three crop plants: laboratory studies with transient silicon concentrations" by Daniel Frick et al. brings out two important points: (a) Si isotope fractionations during plant uptake are similar no matter the Si is taken up actively or passively with water flux; (b) contrasted Si isotopes fractionations at the root-shoot interface reveal different plant Si accumulation strategies. Until now, this could only be speculated from data in the literature and for once it is clearly demonstrated. This conclusion is of great interest for the community and I'm convinced that this study will be really helpful for a large number

of studies. Also, I would like to thank the authors for the high quality of their study at all steps. The experiment is well-designed and fit-for-purpose, the dataset is of high quality, the manuscript is very well written and easy to follow. It's a pleasure to read this work that is perfectly adapted to Biogeosciences. Overall, there is very little to suggest in terms of improvements but here are some minor comments.

Title: I'm not sure that the term "transient" is the best one. To me, it's not appropriate but I'm not a native speaker. I would prefer something like "exhaustible" or "finite". Also, I would have loved a title less technical to attract more readers but it's a safe choice.

Material and methods: I agree with Reviewer 2 that it would be useful to add details on how transpiration was measured. L89: Have you checked the Si solubility limit at $15°C$? No sign of polymerization? Section 2.5.1 and 2.5.3. Why don't you analyse Si isotopes of nutrient solutions directly after a cationic purification? The content of anions is too high? As salts are not detailed in section 2.1 it's not obvious what could compromise the analysis. It's worth mentioning what you feared with these samples. I guess you did not choose the easy way for a reason. L126: It might be useful to rephrase this sentence that is a bit confusing. I had to read the sentence a few times to understand that the important thing is the amount of NaOH/$\mu$g Si and not the molarity of the solution. It's worth explaining why you add a solution and not a powder directly as for solid samples. I guess it's to recover Si left on the crucible sides. Have you tried this protocol with dissolved references like a solution of BHVO-2? There are so many different protocols for solution with a complex matrix that a quality check is useful. Alternatively, it is worth mentioning that your protocol is equivalent to the one of Steinhoefel et al 2017 (excluding the destruction of DOC) as you both use 1mmol of NaOH / 100$\mu$g Si (if my calculations are correct).

L 150: It would be useful to add some references (e.g., Savage et al., 2014 for BHVO-2 and Delvigne et al., 2019 for ERM-CD281) Camille Delvigne, Abel Guihou, Jan A. Schuessler, Paul Savage, Sebastian Fischer, Jade E. Hatton, Kate R. Hendry, Germain Bayon, Emmanuel Ponzevera, Bastian Georg, Alisson Akerman, Oleg Pokrovsky,

Frank Poitrasson, Jean-Dominique Meunier, and Isabelle Basile-Doelsch (2019). An inter-comparison exercise of the Si isotope composition of soils and plant reference materials. Geophysical Research Abstracts, Vol. 21, EGU2019-18488, 2019.

L338: It is hard to find its way with all these data as you mix 30/28 and 29/28. It would be less confusing for the reader if you stick only to 30/28 fractionation factors and just specify when it is recalculated from 29/28. Also, it may be useful to remind here your own 30/28 fractionation factors to directly see that your data are within the literature range. It's also worth mentioning that all species in your list are Si accumulators.

L384-392: The link with the previous section is a bit poor. It's too bad to end the discussion with a weak paragraph. . .

L 394: It would be more careful with the "species-specific" term as your study demonstrates that your fractionation factors are rather similar despite your 3 plants have very different Si strategies. This might be confusing and sounds contradictory.

---

## Referee Comment (RC4) · Anonymous Referee #2 · 7 May 2020

HYPOTHESIS AND THEORY
published: 09 October 2015
doi: 10.3389/fpls.2015.00853

[Figure]

**A possible mechanism of biological silicification in plants**

Christopher Exley*

The Birchall Centre, Lennard-Jones Laboratories, Keele University, Stoke-on-Trent, UK

Plants are significant exponents of biological silicification. While not all plants are generally considered as biosilicifiers the extent to which all plants deposit biogenic silica is largely unknown. There are plants which are known as silica accumulators though even in these plants the extent and degree to which their tissues are silicified is neither appreciated nor understood. An elucidation of the mechanism of silicification in biota is complicated by a lack of known bio-organic chemistry of silicic acid, the starting point in this process. Herein I argue the case that biological silicification is an entirely passive process. It is passive from the point of view that its underlying mechanisms and processes do not require us to invoke any as yet undiscovered silicon biochemistry. It is also passive in that although silicification confers clear biological/ecological advantages under certain conditions, it is actually non-essential in all plants and potentially, at least, toxic in some.

Keywords: plant silica, silicic acid, biological silicification, bioinorganic chemistry

OPEN ACCESS

Edited by:
Marta Wilton Vasconcelos,
Universidade Católica Portuguesa,
Portugal

Reviewed by:
Marta R. M. Lima,
University of California, Davis, USA
Rivka Elbaum,
Hebrew University of Jerusalem, Israel

*Correspondence:
Christopher Exley
c.exley@keele.ac.uk

Specialty section:
This article was submitted to
Plant Nutrition,
a section of the journal
Frontiers in Plant Science

Received: 17 April 2015
Accepted: 28 September 2015
Published: 09 October 2015

Citation:
Exley C (2015) A possible mechanism
of biological silicification in plants.
Front. Plant Sci. 6:853.
doi: 10.3389/fpls.2015.00853

**SILICON IS TAKEN UP BY PLANTS AS SILICIC ACID**

The only form of silicon in soil waters which is available for entry or uptake into a plant is silicic acid, $Si(OH)_4$ (Exley, 1998). Its molecular structure is a single atom of silicon surrounded in a tetrahedral configuration by four hydroxyl (or silanol) groups. The pka for $Si(OH)_4$ is *ca.* 9.6 which means that it is neutral under almost all possible soil water milieus and that it does not lose its first proton from any of the hydroxyl groups at pH below 10. Therefore $Si(OH)_4$ is a very weak acid and additionally where conditions do allow for its effective deprotonation (pH > 10) any purported monosilicate species are expected to be unstable and to immediately form disilicate anions (Exley and Sjöberg, 2014). The disilicate anion, $Si_2O_2(OH)_6{}^{2-}{}_{(aq)}$, which is a more stable form of silicate in aqueous media, is a precursor to 'cement chemistry' and through interactions with essential metal cations including calcium and magnesium would prove to be extremely toxic to plant roots should it be present in soil solutions or at root surfaces, so it too can be ruled out as a biologically available form of soil water silicon.

When considering the mechanism of biological silicification in plants it is imperative to recognize that there are simply no known circumstances where the form of silicon entering the root, or moving throughout the plant, is expected to be 'silicate,' $SiO(OH)_3{}^-{}_{(aq)}$, and so all mechanisms which suggest a role for this form of silicon in silicon uptake and movement in plants should be re-evaluated. There are no known ligands for monosilicate anions in any transporters or channels that have previously been implicated in the uptake and movement of silicon. If the movement of silicon across a membrane does not involve any binding of silicon or even other silicon-specific interactions then this movement of silicon must be entirely passive. The seminal works by Ma et al. (2006, 2007) and Yamaji et al. (2008) on the movement of silicon in plants have inadvertently confused this subject by their suggestions that they have identified silicon

transporters. These important pieces of research have actually identified channels or pores through which silicon as $Si(OH)_4$ can pass by an unidentified but almost certainly passive mechanism. It is important to emphasize that the subject of biochemistry differentiates quite clearly between channels and transporters and it is commonly and widely accepted in biochemical nomenclature that transporters mediate (or catalyze) the movement of a solute by physically binding to the solute in facilitating its movement across a membrane. The binding of $Si(OH)_4$, or any other form of silicon, has neither been demonstrated nor is known to occur in its movement across biological membranes. There are no known silicon transporters, as opposed to channels, in plants and the question then arises as to whether there are silicon-specific channels involved in the transmembrane passage of silicon? For example, experiments in which purported silicon transporters were expressed in oocytes and different rates of silicon movement through these channels were demonstrated depending upon the ambient conditions does not necessarily change such channels into transporters (Ma et al., 2006, 2007). It is still $Si(OH)_4$ which moves across the membrane and there are other factors, not specifically or necessarily related to silicon, which have influenced its passive movement into oocytes and similarly its uptake and movement into and throughout plants. We shall visit these special circumstances throughout this essay. It is important to recognize at this point that previous research interpreted gradients of total silicon across biological membranes as evidence for active transport of silicon (Ma et al., 2006, 2007; Yamaji et al., 2008). Herein these observations are not discounted but re-interpreted with an emphasis on how they might be explained by passive movement of $Si(OH)_4$.

**SILICIC ACID FOLLOWS WATER INTO PLANTS**

Occam's razor tells us quite categorically that the biologically available form of silicon in soil waters is $Si(OH)_4$. Since this is a small neutral molecule with no known biochemical interactions with organic ligands and only a very limited inorganic chemistry there is a strong likelihood that the entry of $Si(OH)_4$ into plant roots follows water (Raven, 1993; Epstein, 1994). This suggests two immediate pathways for the entry of $Si(OH)_4$, an apoplastic or extracellular route, in which $Si(OH)_4$ passes between cells, and a symplastic or intercellular route which will involve cell to cell movement of $Si(OH)_4$. While the former may be the major pathway for the entry of $Si(OH)_4$ into xylem it is also likely that both this process and the further movement of $Si(OH)_4$ throughout xylem and other conducting tissue will involve some degree of movement of $Si(OH)_4$ across biological membranes. If as is suggested herein that $Si(OH)_4$ follows water then it will also use water channels or aquaporins as gateways across membranes. Indeed the so-called silicon transporters identified by Ma et al. (2002) are aquaporins with no specific selectivity for $Si(OH)_4$ (Mitani et al., 2008). Aquaporins are a diverse set of water channels; note they are not transporters, with as many as 30 or more different forms in many plants (Li et al., 2014). These different forms, through their structural arrangements in membranes, present a continuum of selectivity toward the transmembrane passage of dissolved (physical and chemical) solutes (Ludewig and Dynowski, 2009) and will not only influence the ease with which $Si(OH)_4$ crosses membranes but potentially help to establish concentration gradients of $Si(OH)_4$ between connecting compartments. The flow of water into and throughout a plant is primarily driven by hydrostatic pressure and so the entry of $Si(OH)_4$ into a plant and its movement toward the extremities of a plant will likewise be under the influence of water flow and not simply osmosis (Knipfer and Fricke, 2011). While aquaporins do not offer significant resistance to the movement of water into and throughout a plant they are likely to be more resistive in the case of the much larger $Si(OH)_4$ molecule (Ludewig and Dynowski, 2009) with some channels allowing easier passage of this molecule than others. This selective permeability or resistance of aquaporins (and potentially other solvent/solute channels) toward the movement of $Si(OH)_4$ presents a mechanism to describe its concentration (relative to soil water), and potentially its super-saturation (>2 mM), within specific compartments.

**PLANTS ARE PERMEABLE TO SILICIC ACID**

The concentration of $Si(OH)_4$ in soil water will be significantly below its solubility of *ca*. 2 mM. As a small neutral solute, $Si(OH)_4$ will be carried by water across the relatively porous root cell wall and into the root and plant (Raats, 2007). Water flow, driven by hydrostatic forces, will enable the further movement of $Si(OH)_4$ throughout the plant potentially culminating in its elimination from the plant through guttation (Yamaji et al., 2008). It is significant that the knockdown of a gene for an aquaporin involved in the movement of silicon into shoots resulted in the enhanced elimination of $SiOH)_4$ from rice by guttation (Yamaji et al., 2008). Apoplastic water flow will ensure a continuous supply of $Si(OH)_4$ to the plant, primarily via xylem, while osmosis and symplastic water flow will deliver $Si(OH)_4$ to all regions and tissues of the plant. The movement of water into and throughout a plant involves its passage across membranes and these barriers offer varying degrees of resistance to water flow (Knipfer et al., 2011). Resistance is lowered by membrane channels known as aquaporins which are permeable to water. They are also permeable to the passive movement of solutes of sufficiently small size including $Si(OH)_4$. The permeability of aquaporins to different solutes depends upon the relative sizes of the solutes and the channel pores and, for non-apoplastic flow, additionally the maintenance of a concentration gradient of some sort across the membrane. Following such routes $Si(OH)_4$ moves freely throughout the plant from root to shoot. Theoretically where water goes, $Si(OH)_4$ has the possibility of going there too. The absence of biological silicification cannot be construed to infer the absence of $Si(OH)_4$ only the absence of conditions which would allow for its auto condensation and precipitation as biogenic silica. The uptake and movement of $Si(OH)_4$ throughout the plant does not require silicon transporters *per se* only membrane channels to allow for both the hydrostatic (hydraulic) and

osmotic movement of $Si(OH)_4$ between adjacent compartments. Wherever and by which pathways water moves throughout the plant the neutral solute $Si(OH)_4$ has the potential to follow without the need to invoke unknown or novel inorganic or organic chemistry of $Si(OH)_4$.

**A PREREQUISITE FOR PLANT SILICIFICATION**

Water is both the vehicle (solvent) and the delivery system for the distribution of $Si(OH)_4$ throughout a plant. Guttation is conceivably the only mechanism for the exit of $Si(OH)_4$ (as $Si(OH)_4$) from a plant. Silicon enters a plant as an under-saturated solution of $Si(OH)_4$ and yet it is found as amorphous hydrated silica within a plant. Biogenic silica cannot be formed spontaneously unless the concentration of $Si(OH)_4$ in a plant exceeds its solubility limit of *ca.* 2 mM. I have previously defined biological silicification as:

> *Biosilicification: the movement of silicic acid from environments in which its concentration does not exceed its solubility (<2 mM) to intracellular or systemic compartments in which it is accumulated for subsequent deposition as amorphous hydrated silica (Exley, 2009b).*

Thus the auto condensation of $Si(OH)_4$ in a plant requires either a mechanism to concentrate $Si(OH)_4$ above its solubility limit or a process whereby the barrier to its auto condensation can be lowered to enable the formation of biogenic silica at under-saturated concentrations of $Si(OH)_4$. Arguably both of these mechanisms may be involved in silicification in plants.

Xylem should be considered as the major conduit for the movement of $Si(OH)_4$ from root to shoot. Measurements of molybdate-reactive silicon have consistently demonstrated xylem exudates to be super-saturated with $Si(OH)_4$ (Hartley and Jones, 1972; Casey et al., 2003; Liang et al., 2005, 2006; Mitani et al., 2005). Some caution is required in interpreting these values as the reduced molybdosilicic acid complex only obeys Beer's law at concentrations up to *ca.* 0.2 mM and so measurements of up to 18 mM $Si(OH)_4$ in xylem exudates would have required significant pre-dilution of samples (Coradin et al., 2004). However, complementary (Fernández Honaine et al., 2013) Si NMR studies have confirmed that, whatever the precise concentration, silicon in xylem exudates is $Si(OH)_4$ and not biogenic silica or complexes of silicon (Casey et al., 2003; Mitani et al., 2005). Such super-saturated concentrations of $Si(OH)_4$ in xylem sap should be considered as anomalous as they are expected to be thermodynamically unstable (Exley and Sjöberg, 2014). However, xylem fluid *in situ* in a living plant is not a static system, it is a non-equilibrium system, and its dynamic nature combined with the relatively slow kinetics of the auto condensation of $Si(OH)_4$ under *in vivo* conditions does provide for an explanation of its super-saturation *in planta*. This concept is supported by the experimental observation that when xylem sap which contained a super-saturated concentration of $Si(OH)_4$ was removed from a plant, thereby creating a static as opposed to non-equilibrium system, the *ex planta* concentration of $Si(OH)_4$

rapidly fell toward its solubility maximum of *ca.* 2 mM (Mitani et al., 2005). This demonstrated that once the xylem fluid was outside of the plant it was not possible to maintain it as a super-saturated solution of $Si(OH)_4$.

The super-saturated levels of $Si(OH)_4$ in xylem sap of some plants have often been used as evidence for silicon transporters and the active uptake of $Si(OH)_4$ from soil water (Liang et al., 2005, 2006). If the entry of $Si(OH)_4$ into a plant depended only upon the establishment of an osmotic gradient, as is the case when oocytes, for example, are used to measure $Si(OH)_4$ uptake in model systems, then super-saturation of $Si(OH)_4$ on one side of a biological membrane would support, if not confirm, the active uptake of $Si(OH)_4$. However, since the movement of $Si(OH)_4$ into root and subsequently xylem follows water, and is not dependent upon an osmotic gradient, the concentration of $Si(OH)_4$ in xylem will actually reflect differences between rates of movement of solute $[Si(OH)_4]$ and solvent (water) into, within and out of xylem tissue. For example, consider an under-saturated solution of $Si(OH)_4$ (e.g., 0.5 mM) being pumped across a membrane which allowed the passage of water at rate X and the passage of the much larger $Si(OH)_4$ molecule at a rate of X/10. This would result in a concentration of $Si(OH)_4$ of *ca.* 5 mM in the environment immediately preceding the $Si(OH)_4$-selective membrane. The combination of hydraulic force and a membrane which resists the unrestricted passage of $Si(OH)_4$ will result in a soil water that was under-saturated with respect to the solubility of $Si(OH)_4$ becoming super-saturated within a plant compartment, for example, xylem tissue. While $Si(OH)_4$ is a small molecule it is substantially larger than water and the resistance offered to its movement *in planta* will subsequently be higher than it is to the flow of water. The 'resistors' in this circuitry will include the wide variety of plant aquaporins which together with other pores and channels will contribute significantly to the concentration of $Si(OH)_4$ within membrane-limited compartments of, for example, xylem tissue. For any given plant species, and hence any given combination of resistors including plant aquaporins, or $Si(OH)_4$ resistors, the concentration of $Si(OH)_4$ in xylem will be constant for any specific soil water $Si(OH)_4$ level.

The first step toward a plant being classified as a silica accumulator must be the establishment of a super-saturated concentration of $Si(OH)_4$ in xylem. The extent to which this is achieved will depend upon the resistance-free entry of $Si(OH)4$ into xylem in combination with $Si(OH)4$ resistors in other areas of the plant from the root to the shoot. In plants which are not known as silica accumulators there may be $Si(OH)_4$ resistors preventing its movement into xylem ( silica may still be deposited in the root) or the $Si(OH)_4$ resistors throughout the plant do not offer sufficient resistance to the movement of $Si(OH)_4$ (relative to water) to support its concentration to a super-saturated level. Such plants may show significant silica deposition when grown in soil solutions containing high levels of $Si(OH)_4$ and almost no evidence of silicification in media which are deficient in $Si(OH)_4$. So, if a plant has the potential to produce a super-saturated solution of $Si(OH)_4$ in xylem tissue across a wide concentration of soil water $Si(OH)_4$ then it is likely to be a known silica accumulator. Some plants may only silicify at high concentrations of soil water $Si(OH)_4$ and other plants may not deposit silica at all

or only deposit silica in the roots. It is probably the case that most plants have the potential for biological silicification and it is the second step, the templating of the silica deposition process which discriminates between those which are highly silicified, such as horsetail, and the rest.

**TEMPLATING SILICIFICATION**

Once a super-saturated concentration of $Si(OH)_4$ is maintained in xylem a steady supply of hydraulically and osmotically driven $Si(OH)_4$ will be available to the rest of the plant tissues. Hydrostatic and osmotic forces drive the radial and axial movement of water from xylem and in following water out of xylem vessels super-saturated $Si(OH)_4$ will encounter various new compartments some of which will support time-dependent formation of dimers, trimers, oligomers, and polymers of $Si(OH)_4$ and eventually the precipitation of silica. These 'compartments' are created by various resistors which influence the relative rates of movement of water and $Si(OH)_4$ (water faster than $Si(OH)_4$) and will include aquaporin-like channels, plasmodesmata and various precursors and constituents of plant tissues, such as those which constitute plant cell walls. The precise nature and abundance of such compartments will be species-specific and the degree to which they may become silicified in any one species will depend upon the soil water content of $Si(OH)_4$ and the extent to which it becomes super-saturated in xylem (and perhaps analogous water conducting tissues).

What may not be generally appreciated is that in plants that are considered as silicon accumulators, for example horsetail and rice, silicification is extensive (Cooke and Leishman, 2011) and the degree to which tissues are silicified cannot always be appreciated using some methods of biological imaging (**Figure 1**). Biogenic silica is extraordinarily stable in acid. When silica-rich plant tissues are digested using a microwave oven at 180°C and 1800W in a 1:1 combination of 15.8M $HNO_3$ and 18.4M $H_2SO_4$ and the resulting clear digests are diluted with ultra pure water and filtered through 0.10 μm membranes the only residue collected by the filters is biogenic silica. When the silica is viewed using the fluor PDMPO and fluorescence microscopy the images obtained are spectacular and in particular they emphasize the myriad structures which are silicified (Law and Exley, 2011). There are structures which appear more heavily silicified than others and their propensities for silicification are probably determined by the respective densities of the molecular structures acting as templates of the precipitation process. We have identified the hemicellulose callose as one such molecular template for biological silicification (Law and Exley, 2011) and others will probably include precursors to and components of plant cell walls (Fleck et al., 2011; Fernández Honaine and Osterrieth, 2012; Yamanaka et al., 2012; Fernández Honaine et al., 2013; Leroux et al., 2013; Zhang et al., 2013).

The mechanism by which callose templates the precipitation of biogenic silica is likely to be entirely passive. Callose is as an amorphous gel-like polymer of glucose units linked by glycosidic bonds and the disorder and flexibility in its structure *in vivo* lends itself to its many functions in plants, including

algae, as well as in yeasts, fungi and lichens (Piršelová and Matušíková, 2013). In plants its intracellular transport is in vesicles and it is continually synthesized and degraded by callose synthases and β-1,3-glucanases, respectively. The adaptability of callose, relative for example, to the more rigid structure of cellulose, makes it ideal as a building material for example in the differentiation of stomata or the development of plasmodesmata. The structure of callose, essentially a loose gel which is rich in hydroxyl functionalities, also makes it an ideal candidate material to provide a constrained environment to template the precipitation of $Si(OH)_4$ as biogenic silica. By way of an example, the differentiation of stomata is a complex process in which callose is involved in almost every step. Apostolakos et al. (2009, 2010) have detailed these stages in fern (*Asplenium nidus* L.), a known silica accumulator (Leroux et al., 2013) and we have shown that silica deposition exactly mimics callose deposition in horsetail (Law and Exley, 2011) and in fern (**Figure 2**). These observations not only support a specific role for callose in silica deposition they demonstrate that the deposition of silica in plants is not simply a one-way process but must involve the modeling, dissolution, and remodeling of silica structures.

**NATURAL SELECTION AND PLANT SILICIFICATION**

Silicification has conferred a range of advantages on silica accumulators and specifically structural support (Hodson et al., 2005), defense against pathogens (Ma, 2004), defense against herbivory (McNaughton et al., 1985), alleviation of micronutrient deficiency (Hernandez-Apaolaza, 2014) and amelioration of metal toxicity (Epstein, 1994). However, the propensity to support the process of silicification, the conversion of a super-saturated solution of $Si(OH)_4$ to amorphous hydrated silica, may also have dictated the success of certain plants to thrive in soil solutions rich in $Si(OH)_4$. To understand what is meant here it needs to be appreciated that saturated solutions of $Si(OH)_4$ which are undergoing rapid auto condensation to form silica nanoparticles are known to be cytotoxic, for example causing rapid haemolysis of red blood cells (Margolis, 1961). Margolis, who described this effect, suggested that the mechanism involved the adsorption and denaturation of a globular protein and that the effect was size-specific and was only observed when silica particles exceeded 5 nm in size (Iler, 1979). Generally the auto condensation of $Si(OH)_4$ is not an issue in biota, it is simply not occurring, and it is only significant in the biosilicifiers and they must achieve the formation of silica without suffering any cytotoxic effects. This suggests two prerequisites to achieving successful and toxicity-free biological silicification; (i) during early stages the size of silica nanoparticles must be maintained below 5 nm and (ii) the assembly of silica structures and frameworks involving silica particles larger than 5 nm must involve biomolecular templates which are not prone to denaturation (perhaps precluding a role for proteins?) or biomolecules which will be sacrificed as part of the silicification process. As mentioned previously, the hemicellulose, callose may be an ideal vehicle for the entrapment of $Si(OH)_4$ and

[Figure]

**FIGURE 1 | Scanning electron microscopy image of silica collected following acid and microwave digestion of rice leaf blade and demonstrating myriad silicified structures.** The silica tube at the center of the image is approximately 100 µm in length and 4 µm in diameter.

[Figure]

**FIGURE 2 | Fluorescent imaging of silica collected following acid and microwave digestion of fern leaf (*Asplenium nidus* L.) and showing multiple stomata in silicified leaf tissue undergoing differentiation.** In particular this image, magnified in the insert, demonstrates how closely silica deposition mimics the deposition of radial fibrillar callose arrays (for example, indicated by star) in stomata in fern (Apostolakos et al., 2009).

the subsequent control of its auto condensation and growth toward nanoparticles (<5 nm) of silica. It has the approximate structure of a sponge being able to soak up $Si(OH)_4$ into myriad constrained spaces each dense with hydroxyl functionality from its constituent glucose units. While the formation of silica may be allowed within these spaces its growth will probably be

significantly delayed or constrained. As was alluded to earlier the extremely detailed way in which silica deposition appears to mirror the role of callose in the differentiation of stomata, cytokinesis and the structure of plasmodesmata (Law and Exley, 2011) would suggest significant plasticity within the callose-silica system with silica both forming and dissolving to mimic the role of callose in these processes. Biological silicification is not a one-way process as it is known to be reversible to a significant extent when the source of $Si(OH)_4$ to the organism is removed (Law and Exley, 2011; Yamada et al., 2014). It is a highly dynamic process and the processes which underlie callose biochemistry may also underlie biological silicification but only in those plants where a super-saturated concentration of $Si(OH)_4$ is maintained in xylem and, perhaps, other conducting tissues. By way of contrast those plants which maintained a super-saturated concentration of $Si(OH)_4$ in xylem tissue but did not also utilize callose (or equivalent biomolecule) have already been selected out of those environments which today support biosilicifiers.

**STEP BY STEP GUIDE TO BIOLOGICAL SILICIFICATION IN PLANTS**

The biologically available form of silicon in soil waters is $Si(OH)_4$ and it follows water into the plant root.

The relative rates of movement of solute [$Si(OH)_4$] and solvent (water) into xylem and other conducting tissues under hydrostatic pressure are governed by water channels, such as aquaporins, and not transporters. Where these channels present significant resistance to the movement of $Si(OH)_4$, relative to water, the solute is progressively concentrated with the result that some plants maintain a super-saturated concentration of $Si(OH)_4$ in these tissues, the degree of super-saturation being governed by the concentration of $Si(OH)_4$ in soil water.

Super-saturated $Si(OH)_4$ in the vascular system acts as a source of $Si(OH)_4$ to all other tissues. Some of this $Si(OH)_4$ leaves the plant through guttation. Transcellular movement of $Si(OH)_4$ following concentration gradients will result in auto condensation of $Si(OH)_4$ upon entering constrained environments, for example, such as those presented by the vesicular transport of callose. Silicification piggy-backing the metabolism and deposition of callose presents sophisticated cellular machinery for the controlled and specific deposition of biogenic silica. This is evident in the highly specialized silicification seen in horsetail and other biosilicifiers. However, silicification is significantly more widespread throughout plant tissues than is generally appreciated and other constrained environments, usually created by biomolecules involved in structures associated with cell walls, will also promote biological silicification to differing extents and degrees of sophistication depending upon the substrate and the delivery of $Si(OH)_4$.

Silicification is a passive process in that it occurs simply as a consequence of biochemistry and cellular machinery which evolved to fulfill entirely different requirements, such as the movement of water and the differentiation of cell walls. We know that this is true as while silicification does confer advantage on some organisms it is not essential for any organism. For example, horsetail grows perfectly well in the complete absence of silica deposition in its tissues though such silica-free plants are more prone to fungal infection (Fauteux et al., 2005; Law and Exley, 2011). There is no known silicon biochemistry (Exley, 1998) and there is a simple reason for this in that the biologically available form of silicon, $Si(OH)_4$, has no organic chemistry and an extremely limited inorganic chemistry. These simple facts explain the non-selection of silicon in the biochemistry of life (Exley, 2009a).

**REFERENCES**

Apostolakos, P., Livanos, P., and Galatis, B. (2009). Microtubule involvement in the deposition of radial fibrillar callose arrays in stomata of the fern *Asplenium nidus* L. Cell Motil. Cytoskeleton. 66, 342–349. doi: 10.1002/cm.20366

Apostolakos, P., Livanos, P., Nikolakopoulou, T. L., and Galatis, B. (2010). Callose implication in stomatal opening and closure in the fern *Asplenium nidus*. New Phytol. 186, 623–635. doi: 10.1111/j.1469-8137.2010.03206.x

Casey, W. H., Kinrade, S. D., Knight, C. T. G., Rains, D. W., and Epstein, E. (2003). Aqueous silicate complexes in wheat *Triticum aestivum* L. Plant Cell Environ. 27, 51–54. doi: 10.1046/j.0016-8025.2003.01124.x

Cooke, J., and Leishman, M. R. (2011). Is plant ecology more siliceous than we realise? Trends Plant Sci. 16, 61–68. doi: 10.1016/j.tplants.2010.10.003

Coradin, T., Eglin, D., and Livage, J. (2004). The silicomolybdic acid spectrophotometric method and its application to silicate/biopolymer interaction studies. Spectroscopy 18, 567–576. doi: 10.1155/2004/356207

Epstein, E. (1994). The anomaly of silicon in plant biology. Proc. Natl. Acad. Sci. U.S.A. 91, 11–17. doi: 10.1073/pnas.91.1.11

Exley, C. (1998). Silicon in life: a bioinorganic solution to bioorganic essentiality. J. Inorg. Biochem. 69, 139–144. doi: 10.1016/S0162-0134(97)10010-1

Exley, C. (2009a). Darwin, natural selection and the biological essentiality of aluminium and silicon. Trends Biochem. Sci. 34, 589–593. doi: 10.1016/j.tibs.2009.07.006

Exley, C. (2009b). "Silicon in life: whither biological silicification?," in *Biosilica in Evolution, Morphogenesis and Nanobiotechnology*, Vol. 47, eds W. E. G. Müller and M. A. Grachev (Berlin: Springer-Verlag Berlin Heidelberg), 173–184.

Exley, C., and Sjöberg, S. (2014). Silicon species in seawater. Spectrochim. Acta Part A Mol. Biomol. Spectrosc. 117, 820–821. doi: 10.1016/j.saa.2013.09.002

Fauteux, F., Remus-Borel, W., Menzies, J. G., and Belanger, R. R. (2005). Silicon and plant disease resistance against pathogenic fungi. FEMS Microbiol. Lett. 249, 1–6. doi: 10.1016/j.femsle.2005.06.034

Fernández Honaine, M., Borrelli, N. L., Osterrieth, M., and Del Rio, L. (2013). Amorphous silica biomineralisations in *Schoenoplectus californicus* (Cyperaceae): their relation with maturation stage and silica availability. Bull. Argentinian Soc. Bot. 48, 247–259.

Fernández Honaine, M., and Osterrieth, M. L. (2012). Silicification of the adaxial epidermis of leaves of a panicoid grass in relation to leaf position and section and environmental conditions. Plant Biol. 14, 596–604. doi: 10.1111/j.1438-8677.2011.00530.x

Fleck, A. T., Nye, T., Repenning, C., Stahl, F., Zahn, M., and Schenk, M. K. (2011). Silicon enhances suberization and lignification in roots of rice (*Oryza sativa*). J. Exp. Bot. 62, 2001–2011. doi: 10.1093/jxb/erq392

Hartley, R. D., and Jones, L. H. P. (1972). Silicon compounds in xylem exudates of plants. J. Exp. Bot. 23, 637–640. doi: 10.1093/jxb/23.3.637

Hernandez-Apaolaza, L. (2014). Can silicon partially alleviate micronutrient deficiency in plants? Planta 240, 447–458. doi: 10.1007/s00425-014-2119-x

Hodson, M. J., White, P. J., Mead, A., and Broadley, M. R. (2005). Phylogenetic variation in the silicon composition of plants. *Ann. Bot.* 96, 1027–1046. doi: 10.1093/aob/mci255

Iler, R. K. (1979). *The Chemistry of Silica*. Hoboken, NJ: John Wiley & Sons, 866.

Knipfer, T., Besse, M., Verdeil, J.-L., and Fricke, W. (2011). Aquaporin-facilitated water uptake in barley (*Hordeum vulgare* L.) roots. *J. Exp. Bot.* 62, 4115–4126. doi: 10.1093/jxb/err075

Knipfer, T., and Fricke, W. (2011). Water uptake by seminal and adventitious roots in relation to whole-plant water flow in barley (*Hordeum vulgare* L.). *J. Exp Bot.* 62, 717–733. doi: 10.1093/jxb/erq312

Law, C., and Exley, C. (2011). New insight into silica deposition in horsetail (*Equisetum arvense*). *BMC Plant Biol.* 11:112. doi: 10.1186/1471-2229-11-112

Leroux, O., Leroux, F., Mastroberti, A. A., Santos-Silva, F., Van Loo, D., Bagniewska-Zadworna, A., et al. (2013). Heterogeneity of silica and glycan-epitope distribution in epidermal idioblast cell walls in *Adiantum raddianum* laminae. *Planta* 237, 1453–1464. doi: 10.1007/s00425-013-1856-6

Li, G., Santoni, V., and Maurel, C. (2014). Plant aquaporins: roles in plant physiology. *Biochim. Biophys. Acta* 1840, 1574–1582. doi: 10.1016/j.bbagen.2013.11.004

Liang, Y., Hua, H., Zhu, Y.-G., Zhang, J., Cheng, C., and Römheld, V. (2006). Importance of plant species and external silicon concentration to active silicon uptake and transport. *New Phytol.* 172, 63–72. doi: 10.1111/j.1469-8137.2006.01797.x

Liang, Y., Si, J., and Römheld, V. (2005). Silicon uptake and transport is an active process in *Cucumis Sativus*. *New Phytol.* 167, 797–804. doi: 10.1111/j.1469-8137.2005.01463.x

Ludewig, U., and Dynowski, M. (2009). Plant aquaporin selectivity: where transport assays, computer simulations and physiology meet. *Cell Mol. Life Sci.* 66, 3161–3175. doi: 10.1007/s00018-009-0075-6

Ma, J. F. (2004). The role of silicon in enhancing the resistance of plants to biotic and abiotic stresses. *Soil Sci. Plant Nutr.* 50, 11–18. doi: 10.1080/00380768.2004.10408447

Ma, J. F., Tamai, K., Ichii, M., and Wu, G. F. (2002). A rice mutant defective in silicon uptake. *Plant Physiol.* 130, 2111–2117. doi: 10.1104/pp.010348

Ma, J. F., Tamai, K., Yamaji, N., Mitani, N., Konishi, S., Katsuhara, M., et al. (2006). A silicon transporter in rice. *Nature* 440, 688–691. doi: 10.1038/nature04590

Ma, J. F., Yamaji, N., Mitani, N., Tamai, K., Konishi, S., Fujiwara, T., et al. (2007). An efflux transporter of silicon in rice. *Nature* 448, 209–212. doi: 10.1038/nature05964

Margolis, J. (1961). The effect of colloidal silica on blood coagulation. *Aust. J. Exp. Biol. Med.* 39, 249–258. doi: 10.1038/icb.1961.25

McNaughton, S. J., Tarrants, J. L., McNaughton, M. M., and Davis, R. H. (1985). Silica as a defence against herbivory and a growth promoter in African grasses. *Ecology* 66, 528–535. doi: 10.2307/1940401

Mitani, N., Ma, J. F., and Iwashita, T. (2005). Identification of the silicon form in xylem sap of rice (*Oryza sativa* L.). *Plant Cell Physiol.* 46, 279–283. doi: 10.1093/pcp/pci018

Mitani, N., Yamaji, N., and Ma, J. F. (2008). Characterisation of substrate specificity of a rice silicon transporter Lsi1. *Pflugers Archiv.* 456, 679–686. doi: 10.1007/s00424-007-0408-y

Piršelová, B., and Matušíková, I. (2013). Callose: the plant cell wall polysaccharide with multiple biological functions. *Acta Physiol. Plant* 35, 635–644. doi: 10.1007/s11738-012-1103-y

Raats, P. A. C. (2007). Uptake of water from soils by plant roots. *Transp. Porous Media* 68, 5–28. doi: 10.1007/s11242-006-9055-6

Raven, J. A. (1993). The transport and function of silicon in plants. *Biol. Rev.* 58, 179–207. doi: 10.1111/j.1469-185X.1983.tb00385.x

Yamada, K., Yoshikawa, S., Ichinomiya, M., Kuwata, A., Kamiya, M., and Ohki, K. (2014). Effects of silicon-limitation on growth and morphology of *Triparma laevis* NIES-2565 (Parmales. *Heterokontophyta*) *PLoS ONE* 9:e103289. doi: 10.1371/journal.pone.0103289

Yamaji, N., Mitatni, N., and Ma, J. F. (2008). A transporter regulating silicon distribution in rice shoots. *Plant Cell* 20, 1381–1389. doi: 10.1105/tpc.108.059311

Yamanaka, S., Sato, K., Ito, F., Komatsubara, S., Ohata, H., and Yoshino, K. (2012). Roles of silica and lignin in horsetail (*Equisetum hyemale*) with special reference to mechanical properties. *J. Appl. Phys.* 111, 044703. doi: 10.1063/1.3688253

Zhang, C., Wang, L., Zhang, W., and Zhang, F. (2013). Do lignification and silicification of the cell wall precede silicon deposition in the silica cell of the rice (*Oryza sativa* L.) leaf epidermis? *Plant Soil* 372, 137–149. doi: 10.1007/s11104-013-1723-z

**Conflict of Interest Statement:** The author declares that the research was conducted in the absence of any commercial or financial relationships that could be construed as a potential conflict of interest.

---

## Author Comment (AC3) · 18 May 2020

Dear Prof. Hodson

Thanks for taking your time to review our manuscript. In the following we are responding to your questions and recommended improvements in detail. We will supply the improved manuscript with track changes in the later process after the discussion has ended.

**This paper represents an interesting investigation into Si isotope fractionation in three contrasting crop plants. I am not aware that anyone has taken this ap-**
**proach before. I have seen that another referee has concentrated on the methodology, and I will not go over these points again. Rather I will look mostly at the interpretation of the results, and give some suggestions for improvements in the discussion.**

We have responded to the comments from *Anonymous Reviewer 2* and Dr. Delvigne and have clarified our Materials and Method section. We will supply the improved manuscript with track changes in the later stage.

**Major Points**

**Line 12 and elsewhere. I am not sure that I would use "a variety of strategies (rejective, passive and active)." As we have come to understand Si uptake by plants it has become obvious that the different species form a spectrum. You mentioned Hodson et al. (2005) and the spectrum is very evident there. I would just say that you took species that take up Si to different extents.**

We understand that the silicic acid uptake classification (active, passive or rejective) is not a strict metric and still source of an intense debate (see also Anonymous Reviewer 2 comment RC1 and RC4 regarding this topic). We have made adaptions and accounted for this throughout the manuscript. The major changes are:

Line 12: *The incorporation of silicic acid from the soil solution into the plants forms a broad spectrum, from varieties which reject to species which actively incorporate silicic acid, these classifications are however to subject to an intense debate.*

*Line 40: Higher plant species form a continuous spectrum to what extend Si is incorporated, depending on the relative amounts of Si taken up they are grouped into three categories: active, passive and rejective (Marschner and Marschner, 2012).*

Line 185: *Plants form a continuous spectrum of different uptake characteristics, from almost no silicic acid incorporation, to actively accumulating silicic acid. The uptake characteristics were can be classified based on the ratio of measured and theoretical*

*Si uptake. A ratio of greater than 1 indicates an active uptake mechanism, a ratio much smaller than 1 a rejective strategy, and a ratio of 1 indicates passive uptake. The theoretical Si uptake was calculated based on the amount of transpired water taken up and the nutrient solution Si concentration.*

**Line 22. Not always at the endodermis (rice)- some species have much more dispersed transporters in the root.**

*We have accounted for this and rephrased the sentence:*

"*In contrast, the transport of silicic acid from the roots to the shoots depends on the preceding precipitation of silicic acid in the roots and the presence of active transporters in the roots.*"

**Line 24 and elsewhere. The finding of significant biogenic silica deposition in the roots of mustard is novel. As far as I am aware it is the first time in a non-woody dicot. The only dicot mentioned in the recent review of silicification of roots by Lux et al. (2020) is beech. I don't think you can really just say "unpublished observations". We need to know more about this- is it endodermal deposition? A picture would help.**

We have currently gathered only little data regarding the mustard root phytoliths and decided not to include these results. One of the reasons is, that we do not have analysed '*fresh*' mustard roots and can thus not provide in depth review where those phytoliths are deposited. Based on your recommendation however, we have added our observations (SEM-EDX measurements of phytoliths extracted from dried mustard roots). Our colleague Danuta Kaczorek has obtained these results and we will include her in the author list.

The following changes are made to the manuscript:

Line 327: Phytolith formation, which was observed in mustard roots (see Fig. S2) could explain the lower Si transfer efficiency of mustard.

Line 370: The isotopic difference between the Si in the shoots and in the roots ($^{30}\Delta$ Root-Shoot) for mustard and wheat, amounts to 0.72 and 0.98 ‰ respectively, and could be explained by precipitation reactions in the roots (see Fig S2 for the observed mustard root phytoliths, for wheat mineral depositions in the roots have also been observed see Hodson and Sangster, 1989, supporting hypothesis 3).

Added the following items to the supplement:

"*Figure S2: SEM-EDX micrograph of Si precipitates (phytoliths) in mustard roots extracted from dried root samples. See Method S4 Phytolith separation and SEM-EDX analysis for detailed extraction and measurement methods.*"

*Method S4 Phytolith separation and SEM-EDX analysis*

*Phytolith separation: One gram of plant material (roots, shoots) was taken for analysis. Removal of organic matter was conducted by burning the samples in a muffle furnace at 500 °C for 5h. Next, the residue material was subject to additional oxidation using 30% $H_2O_2$ for 0.5h, the carbonates were dissolved by 80 °C in HCl (10Vol.% ) for 10 min. The plants residue was washed with water, bulked, and dried at 105 °C. SEM-EDX analysis: ZEISS EVO MA10 (HV, LV, LaB6 cathode) equipped with a Bruker QUANTAX EDS system including a liquid nitrogen free XFlash R 5010 Detector (energy resolution of 123 eV for MNKa at 100,000cps). The SEM operated at 20keV, with an average working distance of 10.5 mm. Software: Esprit 2.1.1., incl Qmap.*

**Line 95 onwards: Sun et al. (2019) found that while there are two Si transporter homologues present in tomato (SlLsi1, a homologue of the rice LSi1 influx transporter; and SlLsi2, a homologue of the rice LSi2 efflux transporter), only SlLsi1 is active. They suggest that the absence of active SlLsi2 explains the low levels of Si accumulation in this species.**

Thanks for bringing this study to our attention: we have included it:

Line 102: "*Conversely, the alleged active Si efflux transporter (Lsi2-like) are present in*

*the family of Brassicacea (Sonah et al., 2017), but not in tomato (Sun et al., 2020).*"

**Line 321 onwards. As already stated phytoliths in the mustard root is a novel finding, and "data not shown" is not really good enough.**

See comment on Line 24. We have added SEM-EDX images of the root phytoliths of mustard.

**Line 340 and elsewhere: I really don't like reviewers that try to increase their citations by recommending their own papers! However, there are some cases where this is justified. I am very surprised that you did not mention the work of Hodson et al. (2008) on Si isotopes in wheat. Our plants were grown in soil to maturity, and so it was a different setup. But one thing is very clear: there is significant fractionation within the wheat shoot. This does not invalidate your results, but it should be noted (our culm d30Si is negative, but leaf sheaths and blades are positive leading to a positive value overall). The second point is that we also found that the lighter isotopes were deposited first. We could not measure Si in the roots because of soil contamination, but we said "It is apparent that there are two main routes for Si transport within the wheat plant, and that heavier isotopes increase towards the end of both routes: (1) culm > leaf sheath > leaf blade; and (2) culm > rachis > inflorescence bracts. A similar pattern was reported by Ding et al. (2005) working on rice. They considered that the process of Rayleigh fractionation explained the accumulation of heavy isotopes in the upper parts of the plant. Essentially, this would involve the lighter 28Si isotope being more reactive, and thus more likely to be deposited in phytoliths. Thus, in wheat, proportionately more 28Si isotope would be deposited in the culm phytoliths, and a greater proportion of 30Si and 29Si would continue in the transpiration stream to the leaf sheath. In the sheath the same fractionation occurs, leading to an even greater concentration of heavier isotopes in the leaf blade." This is exactly the same process that you postulate is happening in the wheat roots before Si flows on to the shoots. So our work confirms your ideas in section 4.4.**

In the paragraph starting on line 338ff we discuss the literature for which we were able to report fractionation factors. The Hodson *et al.* 2008 manuscript does unfortunately not provide the soil water or soil silicon isotope composition to calculate the fractionation factor. We acknowledge that there is significant internal fractionation observed (e.g. Ding et al., 2005; Hodson et al., 2008) which is one of the reasons we have decided to investigate the silicon isotope fractionation on bulk shoots and roots and not in greater detail. We made changes in section 4.4 and 4.5 to highlight this confirmation and included the Hodson et al. 2008 reference:

Line 390: "*Several researcher have observed an enrichment of $^{30}$Si along the transpiration stream* (Ding et al., 2005; Hodson et al., 2008; Sun et al., 2016)*, which represents a second Rayleigh-like fractionation process internally within the shoots. A possible explanation for this observation is the formation of phytoliths. Early in the transpiration stream, the kinetically controlled condensation of silicic acid leads to the preferential incorporation of $^{28}$Si into phytoliths, whereas the remaining silicic acid in the fluid is enriched in $^{30}$Si and further transported along the transpiration stream. This process could be analysed spatially resolved using an in situ technique to target individual phytoliths (e.g. Frick et al., 2019).*"

Line 404: "*Our results demonstrate that the fractionation between roots and shoots is variable in direction and is controlled by internal plant processes, which are also present within subparts of the roots and shoots (Ding et al., 2005; Hodson et al., 2008).*"

**Minor correction Line 43 Yan**

We have used the official notation used by *Journal of Integrative Agriculture* (https://www.sciencedirect.com/science/article/pii/S2095311918620374) for Gua-chao YANs last name.

We hope that our answers clarify your questions and remarks. Thank you for the

suggested improvements and clarification for our manuscript.

Best regards on behalf of all my co-authors,
Daniel A. Frick

**Literature:**

Ding, T. P., Ma, G. R., Shui, M. X., Wan, D. F. and Li, R. H.: Silicon isotope study on rice plants from the Zhejiang province, China, Chem. Geol., 218(1-2 SPEC. ISS.), 41–50, doi:10.1016/j.chemgeo.2005.01.018, 2005.

Frick, D. A., Schuessler, J. A., Sommer, M. and Blanckenburg, F.: Laser Ablation In Situ Silicon Stable Isotope Analysis of Phytoliths, Geostand. Geoanalytical Res., 43(1), 77–91, doi:10.1111/ggr.12243, 2019.

Hodson, M. J. and Sangster, A. G.: Subcellular localization of mineral deposits in the roots of wheat (Triticum aestivum L.), Protoplasma, 151(1), 19–32, doi:10.1007/BF01403298, 1989.

Hodson, M. J., Parker, A. G., Leng, M. J. and Sloane, H. J.: Silicon, oxygen and carbon isotope composition of wheat (Triticum aestivum L.) phytoliths: implications for palaeoecology and archaeology, J. Quat. Sci., 23(4), 331–339, doi:10.1002/jqs.1176, 2008.

Marschner, H. and Marschner, P.: Marschner's mineral nutrition of higher plants, Academic Press. [online] Available from: https://www.sciencedirect.com/science/book/9780123849052 (Accessed 5 April 2018), 2012.

Sonah, H., Deshmukh, R. K., Labbé, C. and Bélanger, R. R.: Analysis of aquaporins in Brassicaceae species reveals high-level of conservation and dynamic role against biotic and abiotic stress in canola, Sci. Rep., 7(1), 1–17, doi:10.1038/s41598-017-02877-9, 2017.

Sun, H., Duan, Y., Mitani-Ueno, N., Che, J., Jia, J., Liu, J., Guo, J., Ma, J. F. and Gong, H.: Tomato roots have a functional silicon influx transporter but not a functional silicon efflux transporter, Plant Cell Environ., 43(3), 732–744, doi:10.1111/pce.13679, 2020.

Sun, Y., Wu, L., Li, X., Sun, L., Gao, J. and Ding, T.: Silicon Isotope Fractionation in Rice and Cucumber Plants over a Life Cycle: Laboratory Studies at Different External Silicon Concentrations, J. Geophys. Res. Biogeosciences, 2829–2841, doi:10.1002/2016JG003443, 2016.

---

## Author Comment (AC4) · 18 May 2020

Dear Dr. Delvigne

We greatly appreciate your time which you took to review our manuscript.

**The manuscript "Silicon isotope fractionation and uptake dynamics of three crop plants: laboratory studies with transient silicon concentrations" by Daniel Frick et al. brings out two important points: (a) Si isotope fractionations during plant uptake are similar no matter the Si is taken up actively or passively with water flux; (b) contrasted Si isotopes fractionations at the root-shoot interface reveal**

**different plant Si accumulation strategies. Until now, this could only be speculated from data in the literature and for once it is clearly demonstrated. This conclusion is of great interest for the community and I'm convinced that this study will be really helpful for a large number of studies. Also, I would like to thank the authors for the high quality of their study at all steps. The experiment is well-designed and fit-for-purpose, the dataset is of high quality, the manuscript is very well written and easy to follow. It's a pleasure to read this work that is perfectly adapted to Biogeosciences. Overall, there is very little to suggest in terms of improvements but here are some minor comments.**

Thank you very much for the validation of our work. It means a lot. In the following we are responding to your questions and recommended improvements in detail. We will supply the improved manuscript with track changes in the later process after the discussion has ended.

**Title: I'm not sure that the term** "**transient**" **is the best one. To me, it's not appropriate but I'm not a native speaker. I would prefer something like** "**exhaustible**" **or** "**finite**" **. Also, I would have loved a title less technical to attract more readers but it's a safe choice.**

We agree that the title is a very technical description of the paper – after some intense discussions we have come up with a shorter title for the manuscript which still grasps the essence of our work:

*Silicon stable isotope fractionation and uptake dynamics of crop species*

**Material and methods: I agree with Reviewer 2 that it would be useful to add details on how transpiration was measured.**

We have added the information in a revised version of the manuscript. The following amendments were made in section 2.3 regarding the details how we measured the transpiration:

[Figure]

"*Each week the pots were weighted without the lid and the plants, and the mass of transpired water was replenished with ultrapure water (18.2 M$\Omega \cdot$ cm). The weight difference to the previous week is considered the mass of water transpired by the plants. The pots were closed with a lid, and we thus neglect evaporation.*"

Additionally, we have also given our definition of transpiration in *ch.* 2.6.1:

"*We define the plant transpiration as the amount of water taken up by the plants and successively transferred into the atmosphere. The transpiration is measured weekly by weighing the pots without the lids and plants, the weight difference to the previous week is considered the mass of water transpired by the plants. The gravimetrically determined transpiration does not account for the amount of water present in the plants at harvest and the negligible amount of guttation* (Joachimsmeier et al., 2012)."

**L89: Have you checked the Si solubility limit at 15C? No sign of polymerization?**

We have not spectroscopically searched for absence of polymerisation in the nutrient solutions. The solubility for amorphous silica at 25 $^\circ$ C is reported to be $\sim$ 116 µg/g (Gunnarsson and Arnórsson, 2000), using their reported temperature dependence the solubility of SiO2 at 15 $^\circ$ C is $\sim$ 95 µg/g and for 18 $^\circ$ C $\sim$ 101 µg/g. Our starting concentration is slightly above the solubility limit between 15-18 $^\circ$ C (by 2-5 µg/g Si). We did however not observe a significant change in the silicon isotope composition during the early course of the experiments. We would expect this when a significant amount of silicic acid polymerises. We have made an amendment to section 4.1 to describe this concern:

"*The initial concentration of Si at the start of the experiment (49.5 µg/g) was slightly above the solubility limits of amorphous silica at 15-18 $^\circ$C (44.2 – 47.1 µg/g), part of the silicon could thus also be lost to polymerisation and precipitation.*"

**Section 2.5.1 and 2.5.3. Why don't you analyse Si isotopes of nutrient solutions directly after a cationic purification? The content of anions is too high? As**

**salts are not detailed in section 2.1 it's not obvious what could compromise the analysis. It's worth mentioning what you feared with these samples. I guess you did not choose the easy way for a reason.**

We have followed the procedure of (Steinhoefel et al., 2017) due to two concerns: the possible interference of the organic content which could be excreted by the roots and the relative high content anions. High temperature NaOH fusion is our 'go-to-method' and we have not evaluated a direct cationic purification. We have rephrased this: see below for the improved passage.

**L126: It might be useful to rephrase this sentence that is a bit confusing. I had to read the sentence a few times to understand that the important thing is the amount of NaOH/g Si and not the molarity of the solution. It's worth explaining why you add a solution and not a powder directly as for solid samples. I guess it's to recover Si left on the crucible sides.**

Well observed, this is exactly the reason we use a solution of NaOH instead of the pellets/powder. We have rephrased this: see below for the improved passage.

**Have you tried this protocol with dissolved references like a solution of BHVO-2? There are so many different protocols for solution with a complex matrix that a quality check is useful. Alternatively, it is worth mentioning that your protocol is equivalent to the one of Steinhoefel et al 2017 (excluding the destruction of DOC) as you both use 1mmol of NaOH / 100g Si (if my calculations are correct).**

Throughout the NaOH fusion and chromatographic separation we have used BHVO-2 and ERM-CD281 as a quality control. However, for the drying step we could not find an appropriate reference sample in liquid form which could act as an independent control (dissolved but unpurified BHVO-2 would contain already a large amount of Na due to the fusion). We have however taken some measures to assure that the drying is not affecting the silicon isotope composition:

- We controlled the yield based on the amounts we dried down and the concentration measured after the NaOH fusion to assure no loss or gain.

- The overall blank levels were contributing less than 1% to the total amount of Si processed.

The passage 2.5.1 reads now:

"*The relative complex matrix (high nutrient content and organic acids) could interfere with the chromatographic purification of Si, thus the nutrient solution was digested following the* "*Sample preparation of water samples*" *by Steinhoefel et al., 2017 without the additional dissolved organic removal step. Briefly, based on the concentration measured, an aliquot of each nutrient solution containing approximately 1000 µg Si was dried down in silver crucibles on a hotplate at 80-95 ° C. The crucibles were then filled with 400 mg NaOH (Merck pellets, p.a. grade, previously checked for low Si blank levels) and ultrapure water to the initial fill level and dried down. This step ensured that Si adhered to the crucible walls was also covered with NaOH. A blank containing ultrapure water and NaOH was processed together with the samples to check for contaminations.*"

**L 150: It would be useful to add some references (e.g., Savage et al., 2014 for BHVO-2 and Delvigne et al., 2019 for ERM-CD281) Camille Delvigne, Abel Guihou, Jan A. Schuessler, Paul Savage, Sebastian Fischer, Jade E. Hatton, Kate R. Hendry, Germain Bayon, Emmanuel Ponzevera, Bastian Georg, Alisson Akerman, Oleg Pokrovsky, Frank Poitrasson, Jean-Dominique Meunier, and Isabelle Basile-Doelsch (2019). An inter-comparison exercise of the Si isotope composition of soils and plant reference materials. Geophysical Research Abstracts, Vol. 21, EGU2019-18488, 2019.**

I'm hesitant to cite a single selected publication for BHVO-2 since more than 27 publications (to my knowledge) have helped to characterise BHVO-2 for its silicon isotope

composition, I have thus opted for the GeoReM database. Regarding ERMC-CD281, I am happy to include the tremendous effort you and your colleagues made to characterise plants and soils for their silicon isotope composition and cite your EGU abstract as a reference:

*"ERM-CD281 resulted in $\delta^{30}Si$ = 0.34 $\pm$ 0.20 ‰ 2s, n=13 and BHVO-2 in $\delta^{30}Si$ = -0.29 $\pm$ 0.09 ‰ 2s, n=40, in line with literature values (Jochum et al., 2005 for BHVO-2 and Delvigne et al., 2019 for ERM-CD281)."*

**L338: It is hard to find its way with all these data as you mix 30/28 and 29/28. It would be less confusing for the reader if you stick only to 30/28 fractionation factors and just specify when it is recalculated from 29/28. Also, it may be useful to remind here your own 30/28 fractionation factors to directly see that your data are within the literature range. It's also worth mentioning that all species in your list are Si accumulators.**

We agree that it is a very crowed section, we have taken your advice and only report 30/28 and indicate where we re-calculate the fractionation factor from a reported 29/28 ratio. Thank you also for pointing this out that we have likely measured the first Si fractionation factors for non-accumulating Si species, we have added this information:

*"Our new Si fractionation factors (tomato 0.33 ‰ and mustard 0.55 ‰ are the first to be reported for non Si accumulator plants and together with wheat (0.43 ‰ are similar to those measured in other Si accumulator species, including rice, 0.30 ‰ (Sun et al., 2008), -1.02 $\pm$ 0.33 ‰ \* (\* indicates results recalculated from $^{29/28}Si$ to $^{30/28}Si$, Ding et al., 2005) and -0.79 $\pm$ 0.07 (Sun et al., 2016), banana, -0.77 $\pm$ 0.21 ‰ \* (Opfergelt et al., 2006) and -0.68 ‰ \* (Delvigne et al., 2009), and corn and wheat, -1.00 $\pm$ 0.31 ‰ \* (Ziegler et al., 2005)."*

**L384-392: The link with the previous section is a bit poor. It's too bad to end the discussion with a weak paragraph:**

We shortened this paragraph and tried to better link it to the discussion before:

"*This implies that Si which is liberated through weathering reactions, may be recycled multiple times through plants, re-dissolved into soil solution and precipitated into secondary minerals before being exported from the ecosystem. Based on plant and phytolith data aggregated by Frings et al., 2016 biogenic silica is unlikely one of the main export flux of Si from the ecosystems. Plants are thus an important factor for the internal ecosystem element cycling (Uhlig and von Blanckenburg, 2019), but not for the particulate Si export. The plant internal processes which distribute, and deposit Si have however, influence on the amounts and chemical form of Si which is cycled through the ecosystem, and these processes can be traced using stable isotopes to identify the mechanism.*"

**L 394: It would be more careful with the** "**species-specific**" **term as your study demonstrates that your fractionation factors are rather similar despite your 3 plants have very different Si strategies. This might be confusing and sounds contradictory.**

This is true and was not the intended meaning of species-specific. We have rephrased the sentence and hope this is now clearer:

"*We have confirmed that the amount of Si uptake into crop plants and the distribution within is species-specific and complex, involving uptake mechanism in varying proportions. However, regardless of the uptake strategy (active and rejective) all three crop species preferentially incorporate light silicon ($^{28}$Si) with a fractionation factor $1000 \cdot \ln(\alpha)$ for tomato 0.33 ‰ for mustard 0.55 ‰ and for wheat 0.43 ‰ . Within uncertainty, the fractionation factors between these species are indistinguishable.*"

We hope that these answers clarify your questions and thank-you for helping with the improvement of this manuscript.

Best regards on behalf of all my co-authors,
Daniel A. Frick

Literature:

Delvigne, C., Opfergelt, S., Cardinal, D., Delvaux, B. and André, L.: Distinct silicon and germanium pathways in the soil-plant system: Evidence from banana and horsetail, J. Geophys. Res. Biogeosciences, 114(G2), n/a-n/a, doi:10.1029/2008JG000899, 2009.

Delvigne, C., Guihou, A., Schuessler, J. A., Savage, P., Fischer, S., Hatton, E., Hendry, K. R., Bayon, G., Ponzevera, E. and Georg, B.: An inter-comparison exercise of the Si isotope composition of soils and plant reference materials, , 21, 18488, 2019.

Ding, T. P., Ma, G. R., Shui, M. X., Wan, D. F. and Li, R. H.: Silicon isotope study on rice plants from the Zhejiang province, China, Chem. Geol., 218(1-2 SPEC. ISS.), 41–50, doi:10.1016/j.chemgeo.2005.01.018, 2005.

Frings, P. J., Clymans, W., Fontorbe, G., De La Rocha, C. L. and Conley, D. J.: The continental Si cycle and its impact on the ocean Si isotope budget, Chem. Geol., 425, 12–36, doi:10.1016/j.chemgeo.2016.01.020, 2016.

Gunnarsson, I. and Arnórsson, S.: Amorphous silica solubility and the thermodynamic properties of H4SiO° 4 in the range of 0° to 350° C at Psat, Geochim. Cosmochim. Acta, 64(13), 2295–2307, doi:10.1016/S0016-7037(99)00426-3, 2000.

Joachimsmeier, I., Pistorius, J., Heimbach, U., Schenke, D., Kirchner, W. and Zwerger, P.: Frequency and intensity of guttation events in different crops in Germany, in 11th International Symposium of the ICP-BR Bee Protection Group, Wageningen (The Netherlands), November 2-4, 2011, vol. 437, pp. 87–90., 2012.

Jochum, K. P., Nohl, U., Herwig, K., Lammel, E., Stoll, B. and Hofmann, A. W.: GeoReM: A New Geochemical Database for Reference Materials and Isotopic Standards, Geostand. Geoanalytical Res., 29(3), 333–338, doi:10.1111/j.1751-

908X.2005.tb00904.x, 2005.

Opfergelt, S., Cardinal, D., Henriet, C., Draye, X., André, L. and Delvaux, B.: Silicon Isotopic Fractionation by Banana (Musa spp.) Grown in a Continuous Nutrient Flow Device, Plant Soil, 285(1–2), 333–345, doi:10.1007/s11104-006-9019-1, 2006.

Steinhoefel, G., Breuer, J., von Blanckenburg, F., Horn, I. and Sommer, M.: The dynamics of Si cycling during weathering in two small catchments in the Black Forest (Germany) traced by Si isotopes, Chem. Geol., 466(January), 389–402, doi:10.1016/j.chemgeo.2017.06.026, 2017.

Sun, L., Wu, L. H., Ding, T. P. and Tian, S. H.: Silicon isotope fractionation in rice plants, an experimental study on rice growth under hydroponic conditions, Plant Soil, 304(1–2), 291–300, doi:10.1007/s11104-008-9552-1, 2008.

Sun, Y., Wu, L. and Li, X.: Experimental Determination of Silicon Isotope Fractionation in Rice, edited by H. Gerós, PLoS One, 11(12), e0168970, doi:10.1371/journal.pone.0168970, 2016.

Uhlig, D. and von Blanckenburg, F.: How slow rock weathering balances nutrient loss during fast forest floor turnover in montane, temperate forest ecosystems, Front. Earth Sci., 7(July), doi:10.3389/feart.2019.00159, 2019.

Ziegler, K., Chadwick, O. A., Brzezinski, M. A. and Kelly, E. F.: Natural variations of $\delta$ 30Si ratios during progressive basalt weathering, Hawaiian Islands, Geochim. Cosmochim. Acta, 69(19), 4597–4610, doi:10.1016/j.gca.2005.05.008, 2005.

---

## Author Comment (AC5) · 25 May 2020

Dear Anonymous Reviewer # 2

Regarding your comment:

**I think that what is actually demonstrated is that silicon as silicic acid follows water and that this is only a passive process. See attached.**

We do not agree with your observation. Our results, comparing the theoretical and the actual amount of Si that plants taken up during growth (Fig. 1c), show a clear evidence that active, metabolism-driven processes or mechanisms must have been involved for

wheat. There is no other explanation for the 2-fold excess of the theoretically taken up amount of Si which we observe for wheat. Of course, this does not mean that the sub-processes you have indicated did not also occur passively.

We will discuss shortly your and our arguments in the revised manuscript.

Best regards on behalf of all co-authors,

Daniel A. Frick

---

## Author Response (AR1)

Dear Prof. Bahn

In the following we summarise our responses to Dr. Delvigne, Prof. Hodson and the Anonymous Reviewer 2. We are confident that our answers provide a significant improvement of our manuscript and would like to thank the reviewers for their suggestions.

**Detailed response to Camille Delvigne**

**The manuscript " Silicon isotope fractionation and uptake dynamics of three crop plants: laboratory studies with transient silicon concentrations" by Daniel Frick et al. brings out two important points: (a) Si isotope fractionations during plant uptake are similar no matter the Si is taken up actively or passively with water flux; (b) contrasted Si isotopes fractionations at the root-shoot interface reveal different plant Si accumulation strategies. Until now, this could only be speculated from data in the literature and for once it is clearly demonstrated. This conclusion is of great interest for the community and I'm convinced that this study will be really helpful for a large number of studies. Also, I would like to thank the authors for the high quality of their study at all steps. The experiment is well-designed and fit-for-purpose, the dataset is of high quality, the manuscript is very well written and easy to follow. It's a pleasure to read this work that is perfectly adapted to Biogeosciences. Overall, there is very little to suggest in terms of improvements but here are some minor comments.**

Thank you very much for the validation of our work. It means a lot. In the following we are responding to your questions and recommended improvements in detail. We will supply the improved manuscript with track changes in the later process after the discussion has ended.

**Title: I'm not sure that the term "transient" is the best one. To me, it's not appropriate but I'm not a native speaker. I would prefer something like "exhaustible" or "finite". Also, I would have loved a title less technical to attract more readers but it's a safe choice.**

We agree that the title is a very technical description of the paper – after some intense discussions we have come up with a shorter title for the manuscript which still grasps the essence of our work:

Silicon stable isotope fractionation and uptake dynamics of crop species

**Material and methods: I agree with Reviewer 2 that it would be useful to add details on how transpiration was measured.**

We have added the information in a revised version of the manuscript. The following amendments were made in section 2.3 regarding the details how we measured the transpiration:

*"Each week the pots were weighed without the lid and the plants, and the mass of transpired water was replenished with ultrapure water (18.2 MΩ·cm). The weight difference to the previous week is considered to quantify the mass of water transpired by the plants. The pots were closed with a lid, and we thus neglect evaporation."*

Additionally, we have also given our definition of transpiration in *ch.* 2.6.1:

*"We define the plant transpiration as the amount of water taken up by the plants followed by transpiration. The transpiration is measured weekly by weighing the pots without the lids and plants. The difference in mass to the previous week is considered the mass of water transpired by the plants. The gravimetrically determined transpiration does not account for the amount of water present in the plants at harvest and the negligible amount of guttation (Joachimsmeier et al., 2012)."*

**L89: Have you checked the Si solubility limit at 15C? No sign of polymerization?**

We have not spectroscopically searched for absence of polymerisation in the nutrient solutions. The solubility for amorphous silica at 25 °C is reported to be ~116 µg/g (Gunnarsson and Arnórsson, 2000), using their reported temperature dependence the solubility of $SiO_2$ at 15 °C is ~95 µg/g and for 18 °C ~101 µg/g. Our starting concentration is slightly above the solubility limit between 15-18 °C (by 2-5 µg/g Si). We did however not observe a significant change in the silicon isotope composition during the early course of the experiments. We would expect this when a significant amount of silicic acid polymerises. We have made an amendment to section 4.1 to describe this concern:

*"As the initial concentration of Si at the onset of the experiment (49.5 µg/g) was slightly above the solubility limits of amorphous silica at 15-18 °C (44.2 – 47.1 µg/g), a fraction of the silicon could also have been lost to polymerisation and precipitation."*

**Section 2.5.1 and 2.5.3. Why don't you analyse Si isotopes of nutrient solutions directly after a cationic purification? The content of anions is too high? As salts are not detailed in section 2.1 it's not obvious what could compromise the analysis. It's worth mentioning what you feared with these samples. I guess you did not choose the easy way for a reason.**

We have followed the procedure of (Steinhoefel et al., 2017) due to two concerns: the possible interference of the organic content which could be excreted by the roots and the relative high content anions. High temperature NaOH fusion is our 'go-to-method' and we have not evaluated a direct cationic purification. We have rephrased this: see below for the improved passage.

**L126: It might be useful to rephrase this sentence that is a bit confusing. I had to read the sentence a few times to understand that the important thing is the amount of NaOH/g Si and not the molarity of the solution. It's worth explaining why you add a solution and not a powder directly as for solid samples. I guess it's to recover Si left on the crucible sides.**

Well observed, this is exactly the reason we use a solution of NaOH instead of the pellets/powder. We have rephrased this: see below for the improved passage.

**Have you tried this protocol with dissolved references like a solution of BHVO-2? There are so many different protocols for solution with a complex matrix that a quality check is useful. Alternatively, it is worth mentioning that your protocol is equivalent to the one of Steinhoefel et al 2017 (excluding the destruction of DOC) as you both use 1mmol of NaOH / 100g Si (if my calculations are correct).**

Throughout the NaOH fusion and chromatographic separation we have used BHVO-2 and ERM-CD281 as a quality control. However, for the drying step we could not find an appropriate reference sample in liquid form which could act as an independent control (dissolved but unpurified BHVO-2 would contain already a large amount of Na due to the fusion). We have however taken some measures to assure that the drying is not affecting the silicon isotope composition:

- We controlled the yield based on the amounts we dried down and the concentration measured after the NaOH fusion to assure no loss or gain.
- The overall blank levels were contributing less than 1% to the total amount of Si processed.

The passage 2.5.1 reads now:

*"The high nutrient content and the organic acids in the nutrient solution potentially impair the chromatographic purification of Si. Thus the nutrient solution was digested following the "Sample preparation of water samples" by Steinhoefel et al., 2017 without employing an additional step for the removal of dissolved organic carbon. Briefly, based on the concentration measured, an aliquot of each nutrient solution containing approximately 1000 µg Si was dried down in silver crucibles on a hotplate at 80-95 °C. The crucibles were then filled with 400 mg NaOH (Merck pellets, p.a. grade, previously*

*checked for low Si blank levels) and ultrapure water to the initial fill level and dried down. This step ensured that Si attached to the crucible walls was also immersed in NaOH. A blank containing ultrapure water and NaOH was processed in parallel to the samples to check for contamination of Si and other elements introduced in the procedure."*

**L 150: It would be useful to add some references (e.g., Savage et al., 2014 for BHVO-2 and Delvigne et al., 2019 for ERM-CD281) Camille Delvigne, Abel Guihou, Jan A. Schuessler, Paul Savage, Sebastian Fischer, Jade E. Hatton, Kate R. Hendry, Germain Bayon, Emmanuel Ponzevera, Bastian Georg, Alisson Akerman, Oleg Pokrovsky, Frank Poitrasson, Jean-Dominique Meunier, and Isabelle Basile-Doelsch (2019). An inter-comparison exercise of the Si isotope composition of soils and plant reference materials. Geophysical Research Abstracts, Vol. 21, EGU2019-18488, 2019.**

I'm hesitant to cite a single selected publication for BHVO-2 since more than 27 publications (to my knowledge) have helped to characterise BHVO-2 for its silicon isotope composition, I have thus opted for the GeoReM database. Regarding ERMC-CD281, I am happy to include the tremendous effort you and your colleagues made to characterise plants and soils for their silicon isotope composition and cite your EGU abstract as a reference:

*"ERM-CD281 resulted in $\delta^{30}$Si = -0.34 ± 0.20 ‰, 2s, n=13 and BHVO-2 in $\delta^{30}$Si = -0.29 ± 0.09 ‰, 2s, n=40, in line with literature values (Jochum et al., 2005 for BHVO-2 and Delvigne et al., 2019 for ERM-CD281)."*

**L338: It is hard to find its way with all these data as you mix 30/28 and 29/28. It would be less confusing for the reader if you stick only to 30/28 fractionation factors and just specify when it is recalculated from 29/28. Also, it may be useful to remind here your own 30/28 fractionation factors to directly see that your data are within the literature range. It's also worth mentioning that all species in your list are Si accumulators.**

We agree that it is a very crowed section, we have taken your advice and only report 30/28 and indicate where we re-calculate the fractionation factor from a reported 29/28 ratio. Thank you also for pointing this out that we have likely measured the first Si fractionation factors for non-accumulating Si species, we have added this information:

*"Our new Si fractionation factors (tomato -0.33 ‰, and mustard -0.55 ‰) are the first to be reported for non-Si accumulator plants and together with wheat (-0.43 ‰) are similar to those measured in other Si accumulator species. These include rice: -0.30 ‰ (Sun et al., 2008), -1.02 ± 0.33 ‰\* (\* indicates results recalculated from $^{29/28}$Si to $^{30/28}$Si, Ding et al., 2005) and -0.79 ± 0.07 (Sun et al., 2016a); banana: -0.77 ± 0.21 ‰\* (Opfergelt et al., 2006) and -0.68 ‰\* (Delvigne et al., 2009); and corn and wheat: -1.00 ± 0.31 ‰\* (Ziegler et al., 2005)."*

**L384-392: The link with the previous section is a bit poor. It's too bad to end the discussion with a weak paragraph:**

In retrospective we agree and have decided to remove the paragraph.

**L 394: It would be more careful with the "species-specific" term as your study demonstrates that your fractionation factors are rather similar despite your 3 plants have very different Si strategies. This might be confusing and sounds contradictory.**

This is true and was not the intended meaning of species-specific. We have rephrased the sentence and hope this is now clearer:

*"The amount of Si uptake into crop plants and the distribution of Si within them is species-specific, and the uptake strategies are in operation in variable relative proportions. However, regardless of the uptake strategy (active and rejective) all three crop species studied preferentially incorporate light silicon ($^{28}Si$) with a fractionation factor $1000 \cdot \ln(\alpha)$ for tomato -0.33 ‰, for mustard -0.55 ‰ and for wheat -0.43 ‰ which are indistinguishable within uncertainty."*

**Detailed response to Martin Hodson**

**This paper represents an interesting investigation into Si isotope fractionation in three contrasting crop plants. I am not aware that anyone has taken this approach before. I have seen that another referee has concentrated on the methodology, and I will not go over these points again. Rather I will look mostly at the interpretation of the results, and give some suggestions for improvements in the discussion.**

We have responded to the comments from *Anonymous Reviewer 2* and Dr. Delvigne and have clarified our Materials and Method section.

**Major Points**

**Line 12 and elsewhere. I am not sure that I would use "a variety of strategies (rejective, passive and active)." As we have come to understand Si uptake by plants it has become obvious that the different species form a spectrum. You mentioned Hodson et al. (2005) and the spectrum is very evident there. I would just say that you took species that take up Si to different extents.**

We understand that the silicic acid uptake classification (active, passive or rejective) is not a strict metric and still source of an intense debate (see also Anonymous Reviewer 2 comment RC1 and RC4 regarding this topic). We have made adaptions and accounted for this throughout the manuscript. The major changes are:

Abstract: "*However, plants differ in the way they take up silicic acid from soil solution. Correspondingly species encompass a broad spectrum, from varieties that reject silicic acid to species that actively incorporate it. Yet these classifications are subject to intense debate.*"

Ch. 1: "*Higher plant species form a continuous spectrum in the extent to which Si is incorporated. According to the amount of Si taken up they are grouped into three categories: active, passive and rejective (Marschner and Marschner, 2012).*"

Ch. 2.6.1: "*The plant Si uptake characteristics can be classified based on the ratio between the measured and the expected Si uptake. A ratio of greater than 1 indicates an active uptake mechanism, a ratio much smaller than 1 a rejective strategy, and a ratio of 1 indicates passive uptake.*"

**Line 22. Not always at the endodermis (rice)- some species have much more dispersed transporters in the root.**

*We have accounted for this and rephrased the sentence:*

"*In contrast, the transport of silicic acid from the roots to the shoots depends on the amount of silicon previously precipitated in the roots and the presence of active transporters in the roots.*

**Line 24 and elsewhere. The finding of significant biogenic silica deposition in the roots of mustard is novel. As far as I am aware it is the first time in a non-woody dicot. The only dicot mentioned in the recent review of silicification of roots by Lux et al. (2020) is beech. I don't think you can really just**

**say "unpublished observations". We need to know more about this- is it endodermal deposition? A picture would help.**

We have currently gathered only little data regarding the mustard root phytoliths and have decided not to include these results. One of the reasons is, that we do not have analysed *'fresh'* mustard roots and can thus not provide in depth review where those phytoliths are deposited. Based on your recommendation we have added our observations (SEM-EDX measurements of phytoliths extracted from dried mustard roots). Our colleague Danuta Kaczorek has obtained these results and we will thus include her in the author list.

The following changes are made to the manuscript:

Ch. 2.7 Method description for the phytolith extraction and SEM-EDX measurement.

Ch. 3.5 Results of the SEM-EDX measurements.

Ch. 4.2: *"The remarkably high Si concentration and amounts in mustard roots, and thus the lower Si transfer efficiency of mustard can be explained by phytolith formation (see Fig. S2)."*

Ch. 4.4: *"The isotopic difference between the Si in the shoots and in the roots ($^{30}\Delta_{Root\text{-}Shoot}$) for mustard and wheat amounts to -0.72 and -0.98 ‰, respectively, and can be explained by Si precipitation in the roots. Indeed, we observed mustard root phytoliths; see Fig. S2. Mineral deposition in wheat roots has also been observed by Hodson & Sangster, (1989), supporting hypothesis (3)."*

Added the following items to the supplement:

*Figure S 2: "Representative SEM-EDX micrograph of Si precipitates (phytoliths) in mustard roots extracted from dried root samples. See SEM-EDX analysis of mustard root phytoliths for detailed extraction and measurement methods."*

**Line 95 onwards: Sun et al. (2019) found that while there are two Si transporter homologues present in tomato (SlLsi1, a homologue of the rice LSi1 influx transporter; and SlLsi2, a homologue of the rice LSi2 efflux transporter), only SlLsi1 is active. They suggest that the absence of active SlLsi2 explains the low levels of Si accumulation in this species.**

Thanks for bringing this study to our attention: we have included it:

Line 102: *"Conversely, the alleged active Si efflux transporter (Lsi2-like) are present in the family of Brassicacea (Sonah et al., 2017), but not in tomato (Sun et al., 2020). An ongoing controversy surrounds the significance of the Lsi1 homologue in tomato. Whereas Deshmukh et al., 2015 used Si uptake studies to infer the transporter to be non-functional, Sun et al., 2020 observed the contrary using Ge as homologue element. Sun and co-workers concluded that the low Si uptake is caused by the lack of a functional Si efflux transporter Lsi2 at the root endodermis."*

**Line 321 onwards. As already stated phytoliths in the mustard root is a novel finding, and "data not shown" is not really good enough.**

See comment on Line 24. We have added SEM-EDX images of the root phytoliths of mustard.

**Line 340 and elsewhere: I really don't like reviewers that try to increase their citations by recommending their own papers! However, there are some cases where this is justified. I am very surprised that you did not mention the work of Hodson et al. (2008) on Si isotopes in wheat. Our plants were grown in soil to maturity, and so it was a different setup. But one thing is very clear: there is significant fractionation within the wheat shoot. This does not invalidate your results, but it should be noted (our culm d30Si is negative, but leaf sheaths and blades are positive leading to a**

positive value overall). **The second point is that we also found that the lighter isotopes were deposited first. We could not measure Si in the roots because of soil contamination, but we said "It is apparent that there are two main routes for Si transport within the wheat plant, and that heavier isotopes increase towards the end of both routes: (1) culm » leaf sheath » leaf blade; and (2) culm » rachis » inflorescence bracts. A similar pattern was reported by Ding et al. (2005) working on rice. They considered that the process of Rayleigh fractionation explained the accumulation of heavy isotopes in the upper parts of the plant. Essentially, this would involve the lighter 28Si isotope being more reactive, and thus more likely to be deposited in phytoliths. Thus, in wheat, proportionately more 28Si isotope would be deposited in the culm phytoliths, and a greater proportion of 30Si and 29Si would continue in the transpiration stream to the leaf sheath. In the sheath the same fractionation occurs, leading to an even greater concentration of heavier isotopes in the leaf blade." This is exactly the same process that you postulate is happening in the wheat roots before Si flows on to the shoots. So our work confirms your ideas in section 4.4.**

In the paragraph starting on line 338ff we discuss the literature for which we were able to report fractionation factors. The Hodson *et al.* 2008 manuscript does unfortunately not provide the soil water or soil silicon isotope composition to calculate the fractionation factor. We acknowledge that there is significant internal fractionation observed (e.g. Ding et al., 2005; Hodson et al., 2008) which is one of the reasons we have decided to investigate the silicon isotope fractionation on bulk shoots and roots and not in greater detail. We made changes in section 4.4 to highlight this confirmation and included the Hodson et al. 2008 reference:

Line 390: *"Within the shoots, Si is not homogenously distributed. Several researcher have observed an enrichment of $^{30}$Si along the transpiration stream (Ding et al., 2005; Hodson et al., 2008; Sun et al., 2016b), compatible with a Rayleigh-like fractionation within the shoots. A possible explanation for this observation is the formation of phytoliths. Early in the transpiration stream, the kinetically controlled condensation of silicic acid leads to the preferential incorporation of $^{28}$Si into phytoliths (e.g. Frick et al., 2019), whereas the remaining silicic acid in the fluid is enriched in $^{30}$Si and further transported along the transpiration stream."*

**Minor correction Line 43 Yan**

We have used the official notation used by *Journal of Integrative Agriculture* (https://www.sciencedirect.com/science/article/pii/S2095311918620374) for Gua-chao YANs last name.

**Detailed response to the Anonymous Referee #2**

We have considered your suggestion and have further clarified the materials and methods section. In detail we provide our answer to your questions and suggestions:

**Line 89: What is this? A somewhat unconventional unit. Do you mean & Line 90: Do you mean 49.5 mg/L? Is this the concentration of Si or the salt?**

µg/g is a SI unit for concentration. Our measurements are based on weighing the solutions, thus we report the concentration as 'per g' and not as 'per mL'. For the convenience of the reader we expanded the sentence and provide the concentration in mM and specified that the concentration refers to Si:

*"Silicon was added in the form of NaSiO₄ to an initial Si starting concentration of 49.5 µg·g$^{-1}$ (1.76 mM). Detail composition can be found in supplementary methods S1. Ultrapure water (resistivity 18.2*

*MΩ·cm) was used to prepare the nutrient solutions and to weekly restock water transpired by the plants."*

**Line 94: What does this mean? How can they 'reject' silicic acid? & Line 94: Active uptake of silicic acid? Where is the evidence that this occurs?**

Active, passive and rejective Si uptake is a concept which has been proposed by several groups: see e.g. (Hodson et al., 2005; Takahashi et al., 1990) and also the review by M. Hodson for this manuscript: https://www.biogeosciences-discuss.net/bg-2020-66/bg-2020-66-RC2.pdf). The classification is based on the amount of silicon is taken up in relation to the water uptake and is also explained in Line 171ff. As also M. Hodson remarked in his review, the uptake of Si is not a strict classification, but a spectrum which allows to qualitatively describe the Si uptake. We have accounted for this and clarified it:

Abstract: *"However, plants differ in the way they take up silicic acid from soil solution. Correspondingly species encompass a broad spectrum, from varieties that reject silicic acid to species that actively incorporate it. Yet these classifications are subject to intense debate."*

Ch. 1: *"Higher plant species form a continuous spectrum in the extent to which Si is incorporated. According to the amount of Si taken up they are grouped into three categories: active, passive and rejective (Marschner and Marschner, 2012)."*

Ch. 2.6.1: *"The plant Si uptake characteristics can be classified based on the ratio between the measured and the expected Si uptake. A ratio of greater than 1 indicates an active uptake mechanism, a ratio much smaller than 1 a rejective strategy, and a ratio of 1 indicates passive uptake."*

**Line 99: Really, so silicic acid does not follow water into either mustard or tomato? Do you have evidence to support this?**

This is not what has been stated in the text. We justify the selection of the plant species and provide information which additional transporter channels / proteins are present in the investigated plants.

**Line 102: Added Si, but how much Si was present in these solutions?**

The amount of Si introduced by the other nutrient salts and the water was not resolvable using the ICP-OES, thus we considered these negligible. We have changed the sentence to:

*"Plant seeds were germinated in Petri dishes with half-strength nutrient solution used for the later growth experiment that contained no added $NaSiO_4$."*

**Line 104: What about the significant increase in sodium content, did you have a control for this?**

We did not counterbalance or remove the Na which has been introduced by the addition of $NaSiO_4$.

**Line 105: How did you measure the volume of transpired water?**

The pots were weighted weekly without the lid and plants, using a balance. The weight difference to the previous week is reported as volume taken up by the plants, assuming a density of 1 g/mL. We replenished the pots by filling up with ultra-pure water to the weight from the previous week. The pots were closed with a lid, and we thus neglect evaporation. The term transpiration is thus referred to the water taken up, which is either lost by transpiration and guttation or stored in the biomass. Based on previous reports (e.g. Joachimsmeier et al., 2012) the amount of fluid lost through guttation, was considered negligible during the course of the experiment. We have added this information:

*"Each week the pots were weighed without the lid and the plants, and the mass of transpired water was replenished with ultrapure water (18.2 MΩ·cm). The weight difference to the previous week is*

*considered to quantify the mass of water transpired by the plants. The pots were closed with a lid, and we thus neglect evaporation."*

Line 112: What about other forms of water loss such as guttation?

See question before. We considered water loss through guttation negligible. We specified how we defined plant transpiration in Ch. 2.6.1:

*"We define the plant transpiration as the amount of water taken up by the plants followed by transpiration. The transpiration is measured weekly by weighing the pots without the lids and plants. The difference in mass to the previous week is considered the mass of water transpired by the plants. The gravimetrically determined transpiration does not account for the amount of water present in the plants at harvest and the negligible amount of guttation (Joachimsmeier et al., 2012)"*

**Line 115: What kind of extracellular Si deposits? Do you simply mean that you washed off the nutrient solution?**

Thanks for bringing this to our attention, we have clarified the sentence:

*"The roots were immersed multiple times in ultrapure water to remove potential extracellular Si deposits and attached nutrients."*

**Line 118: how?**

We have added a link to chapter 2.5.2 where the digestion procedure is explained.

**Line 123: Why are all essential details of methods in Supplementary files, they need to be here in M&M.**

We have expanded the section and explained how we have performed the concentration measurements by ICP-OES:

*"Samples and standard were analysed following a procedure by Schuessler et al., 2016. Briefly, the samples and standards were doped with an excess of $CsNO_3$ (1 mg $g^{-1}$) to reduce matrix effects in the ICP source that are likely to be caused from the high nitrogen content of the samples and quantified applying an external calibration. The relative analytical uncertainties are estimated to be below 10% and agreed with the nominal concentration of the starting solutions."*

**Line 125: What do you mean? How do you know that the aliquot contains this amount of Si? Where are the methods?**

The concentration is known from the measurement by ICP-OES, we have clarified this part:

*"Briefly, based on the concentration measured, an aliquot of each nutrient solution containing approximately 1000 µg Si was dried down in silver crucibles on a hotplate at 80-95 °C."*

**Line 131: estimates based upon what?**

The concentration was estimated by analysing an exploratory experiment, we have clarified this:

*"50-800 mg of plant material, depending on the Si concentration determined in an exploratory study, was weighed into Ag crucibles and combusted overnight (2h at 200 °C, 4h at 600 °C, then cooled to room temperature) in a furnace (LVT 5/11/P330, Nabertherm)."*

**Line 133: what does this mean?**

We removed this information since the results were not presented in this study.

**Line 134: what is the Si content of this salt?**

We have specified what the Si content of NaOH was:

*"After cooling 400 mg NaOH (TraceSELECT, Sigma-Aldrich, checked for low Si blank levels) was added."*

**Line 137: Does plant silica dissolve under these conditions?**

The high temperature fusion of silicates, silicon, and bio silica (e.g. diatoms, phytoliths) using NaOH has been proven to be quantitative. The silicate is transformed in this fusion into its silicic form which can be dissolved in water.

**Line 138: How? You convert Si to a cation? You need to fully explain these methods.**

The dissolution procedure of silicates, silicon and bio silica is state of the art in geosciences. Si is present in SiO2 as $Si^{4+}$, counterbalanced by 2 $O^{2-}$. Therefore, we do not need to convert Si into a cation. The NaOH accelerates the dissolution of the oxide, and after the addition of water silicon is present as silicic acid ($H_4SiO_4$ and depending on the pH also in the form of $H_3SiO_4^-$ (see e.g. (Stamm et al., 2019), their Fig. 1 for an aqueous Si species in equilibrium diagram).

We hesitate to include the entire Supplementary Method S3 into the main text, since this is a routine method applied in chemistry and geochemistry to dissolve silicates quantitatively.

**Line 140: Again, the methods should be here and not in Supplementary files.**

We have clarified that in the supplementary files a step-by-step procedure can be found. We hesitate to include the entire Supplementary Method S3 into the main text, since this is a routine method applied in chemistry and geochemistry to dissolve silicates quantitatively.

**Line 161: It would seem that all measurements rely upon accurate measurements of water intake. Where have you written about how you measured the amount of transpired water? Why do you assume that all water uptake is reflected by this transpired volume? Again, what about processes like gutation. Even if your measurements of transpiration are accurate, they do not represent water uptake into the plant.**

See response to your question on Line 105.

**I think that what is actually demonstrated is that silicon as silicic acid follows water and that this is only a passive process. See attached.**

We do not agree with your observation.

The expected Si uptake was calculated based on the amount of transpired water and the nutrient solution Si concentration. This expected Si uptake equals the amount of the passive process, where silicic acid follows the water. Our results, comparing the expected and the actual amount that plants taken up during growth (Fig. 1c), show a clear evidence that active, metabolism-driven processes or mechanisms must have been involved for wheat. There is no other explanation for the 2-fold excess of the theoretically taken up amount of Si which we observe for wheat. Of course, this does not mean that the sub-processes you have indicated did not also occur passively.

**Additional changes**

We have made the following additional changes to clarify the manuscript.

**New-Coauthor:** Danuta Kaczorek from the Leibniz Centre for Agricultural Landscape Research (ZALF) has extracted the phytoliths and obtained the SEM images.

**Abstract and introduction:** we have revised the language in those two chapters.

**Fig 1c:** Changed the axis label to "Expected Si uptake (mg)" and "Measured : expected Si uptake" and clarified this in section 2.6.1.

**Fig 2, caption:** We have rephrased the caption of Figure 2.

**Literature**

Delvigne, C., Opfergelt, S., Cardinal, D., Delvaux, B. and André, L.: Distinct silicon and germanium pathways in the soil-plant system: Evidence from banana and horsetail, J. Geophys. Res. Biogeosciences, 114(G2), n/a-n/a, doi:10.1029/2008JG000899, 2009.

Delvigne, C., Guihou, A., Schuessler, J. A., Savage, P., Fischer, S., Hatton, E., Hendry, K. R., Bayon, G., Ponzevera, E. and Georg, B.: An inter-comparison exercise of the Si isotope composition of soils and plant reference materials, , 21, 18488, 2019.

Ding, T. P., Ma, G. R., Shui, M. X., Wan, D. F. and Li, R. H.: Silicon isotope study on rice plants from the Zhejiang province, China, Chem. Geol., 218(1-2 SPEC. ISS.), 41–50, doi:10.1016/j.chemgeo.2005.01.018, 2005.

Frick, D. A., Schuessler, J. A., Sommer, M. and Blanckenburg, F.: Laser Ablation In Situ Silicon Stable Isotope Analysis of Phytoliths, Geostand. Geoanalytical Res., 43(1), 77–91, doi:10.1111/ggr.12243, 2019.

Frings, P. J., Clymans, W., Fontorbe, G., De La Rocha, C. L. and Conley, D. J.: The continental Si cycle and its impact on the ocean Si isotope budget, Chem. Geol., 425, 12–36, doi:10.1016/j.chemgeo.2016.01.020, 2016.

Gunnarsson, I. and Arnórsson, S.: Amorphous silica solubility and the thermodynamic properties of H4SiO°4 in the range of 0° to 350°C at Psat, Geochim. Cosmochim. Acta, 64(13), 2295–2307, doi:10.1016/S0016-7037(99)00426-3, 2000.

[revised manuscript text omitted]
. The plant-internal processes whichthat distribute, and deposit Si have however, influence on the amounts and chemical form of Si which is cycled through the ecosystem, and these processes can be traced using stable isotopes to identify the mechanism. as biogenic silica, secondary clays, or as dissolved Si. The relative magnitude

460 between these fluxes depends however on the environmental conditions (Frings et al., 2016; Sommer et al., 2006, 2013). The isotope composition of the dissolved Si in river water shows almost exclusively a heavier silicon isotope signature than the bedrock they drain (Frings et al., 2016; Opfergelt and Delmelle, 2012). To close the Si isotopic mass balance therefore requires an isotopically light solid counterpart (Bouchez et al., 2013). The plant and phytolith data aggregated by Frings et al., 2016

465

**5 Conclusion**

The amount of Si uptake into crop plants and the distribution of Si within them is species-specific  and the rejective, passive and activestrategies are in operation in  relative proportions. However, regardless of the uptake strategy (active and rejective) all three crop species studied

[revised manuscript text omitted]

75 chemistry was determined in a 1:10-fold dilution by ICP-OES.

---

## Author Response (AR2)

Dear Prof. Bahn

We summarise in this response the main points raised by reviewer 2 and how we have approached them. They can be grouped into three main issues.

1) Plants do not have an active uptake mechanisms and Si uptake is in principle unspecific and passively taken up and passively transported in plants (Exley et al., 2020; Exley and Guerriero, 2019).

While we retain a neutral position in this debate we note that there are numerous articles that assume a highly selective Si uptake mechanism in Si accumulator plants. Assuming a passive uptake mechanism and passive transport within the plant, the amount of Si incorporated is set by transpiration, provided that the availability of soluble Si in the root space does not differ between species. On the other hand, if we can observe species with higher transpiration rates which incorporate less Si than plant species with a lower transpiration rate (under the same environmental conditions), this demonstrates that plant species must differ in their Si uptake mechanism or in their Si transport within the plant. This is what we observe. In the revised version we however point at several occasions at the controversy surrounding the debate over uptake mechanisms, and in particular to the points made in the papers that question the active uptake (Exley et al., 2020; Exley and Guerriero, 2019).

2) The lack of a control experiment.

The aim of our study is to trace and quantify the uptake of silicon in three different crop species using the naturally occurring shifts (=isotope fractionation) in the abundances of the stable isotopes of silicon and physiological parameters (plant Si amounts and concentrations, transpiration). Performing a control experiment in the absence of plants makes no sense, as Si will not change compartments hence there is no isotope fractionation. However, what we have done can be seen as a variant of a control experiment: We have grown (in triplicate) spring wheat, tomato, and mustard under the ***same*** environmental conditions (humidity, light, temperature, and the initial nutrient composition), and thus the differences observed are controls over differences in plant physiology.

3) Methodological issues and missing information on methods.

One set of questions by reviewer 2 pertains to explanations that were in fact provided elsewhere in the manuscript, and hence no changes were made. The reviewer may not have read these sections. Another set of questions pertains to the isotope analytical sample preparations employed, and to basic isotope data interpretation. All of this is widely available in isotope geochemical literature, and is standard in isotope geochemistry. We believe that providing this background information in this paper would dilute the flow of information and is in any case beyond the scope of this paper.

Still, both reviewers pointed at deficits in the presentations and suggested clarifications. We implemented these. Below we provide a detailed response to all of the points raised by reviewer 2. For many of those points however we rebutted the requests, since the information is already present in the manuscript, or the request is not within the scope of our work.

We hope that our answers together with the (improved) manuscript is now ready for publication in 'Biogeosciences'.

Daniel A. Frick on behalf of all coauthors.

**Response to Anonymous Reviewer 2**

In the following we respond individually to all the comments:

**Title: I appreciate that you changed the title to accomodate another reviewer but the new title really does not make sense in English.**

Thanks for bringing this to our attention, after some consultations with native speakers we have changed the title of the manuscript to: *Silicon uptake and isotope fractionation dynamics by crop species*

**Line 35: There are excellent papers on the biogeochemical cycle of silicon and silicic acid that could be cited here.**

We have added and rearranged citations of excellent papers on the biogeochemical cycle of silicic acid and silica: *"One crucial but poorly understood aspect of terrestrial Si biogeochemistry is biological cycling (Carey and Fulweiler, 2012; Derry et al., 2005; Sommer et al., 2006, 2013)."*

**Line 36: What do you mean by this statement? How is silicon 'recycled multiple times' in plants?**

The sentence does not claim that Si is recycled *within* plants, but within the ecosystem by plants (e.g. Si from decomposing phytoliths (litter fall) will be available for re-incorporate into plants, before it is lost from ecosystems. We have clarified the statement: *"Si has well documented biological roles, and Si may be recycled multiple through higher plants before being lost from an ecosystem."*

**Line 42: Sequestration of heavy metals? Evidence of this? What about Al, this is not a heavy metal.**

We have clarified : *"Despite having a disputed biochemical role, Si is considered beneficial for plant growth, including crops: Si increases abiotic stress mediation (aluminium and heavy metal toxicity, salinity), biotic stress resistance (defence against herbivores), and improves the plants' structural stability* (Coskun et al., 2019b; Epstein, 1994, 1999, 2001; Exley and Guerriero, 2019; Ma, 2004; Richmond and Sussman, 2003).*"*

**Line 46: Actually plants are classified according to how much 'silica' they deposit in their tissues, see work by Hodson. There is certainly no consensus on their classification according to the parameters you suggest. There is equally good evidence that the movement of silicic acid into and in plants is entirely passive and even if this is not your point of view you should acknowledge this. See for example, most recently; https://link.springer.com/article/10.1007%2Fs12633-019-00360-w**

We describe here a very widely used and applied scheme to classify **_the amounts_** of silicon incorporated by higher plant species (see e.g. Coskun et al., 2019b; Guerriero et al., 2020; Handreck and P Jones, 1967; Takahashi et al., 1990). The sentence it is not about **_how_** silicon is taken up. How Si is incorporated is exposed in Line 50/51, see also our next response. No changes were made.

**Line 51: The discussion is not simply about the definition of 'active' it is whether or not previous research has identifed active transport of silicic acid.**

We made changes to the sentence to include these aspects: *"However, the term "active uptake" is still widely debated. In particular whether the classification as active or passive is justified, as well as the evidence for involvement of an active, metabolically controlled process in some plant species is subject of an intense discussion (Coskun et al., 2019a; Exley, 2015; Exley et al., 2020)."*

**Line 59: But these studies have recently been questioned and as yet no unequivocal evidence of active transport of silicic acid has been demonstrated.**

We have already highlighted the ongoing dispute regarding the 'active' nature of silicic acid uptake in Line 50. Our study design and results however cannot provide evidence whether the active transport requires the expense of energy, thus we cannot provide answers towards either view of this ongoing discussion. We are thus taking a neutral standpoint in this discussion. Thus, we believe the sentence the reviewer comments on is adequate. No changes were done.

**Line 70: But, see Exley et al 2020 why Ge is not suitable as an analogue for movement of silicic acid.**

We have added this citation to Line 62ff *"Both techniques impose limitations on growth experiments, either due to safety concerns arising from radioactivity or due to physiological differences between the homologue element Ge and Si (Exley et al., 2020; Takahashi et al., 1990)."*

**Line 82: Makes sense since lighter molecules of silicic acid will diffuse more easily than heavier ones.**

No change made.

**Line 88: No evidence of transporters, better 'channels'.**

For the sake of consistency throughout the manuscript we use transporter when talking about the homologues of Lsi1 and Lsi2. See e.g. in the introduction: *"Lsi1, Low Silicon 1 transporter, a thermodynamically passive transporter from the family of aquaporin-like proteins) incorporates Si, whereas a metabolically active efflux transporter (Lsi2, a putative anion-channel transporter)"*

No change made.

**Line 99: Na4SiO4**

Thanks for bringing this to our attention, we have corrected it with the name (sodium silicate trihydrate) and the corresponding sum formula ($Na_2O_7Si_3 \cdot 3H_2O$).

**Line 99: This concentration of sodium orthoscilicate when dissolved in pure water will produce a highly alkaline solution, pH above 12. How did you accomodate this? 1/6th strength MS nutrient soln includes about 0.012 mM silicic acid. What was the Si content of your control?**

This information is provided in the supplementary methods S1. The nutrient solution was acidified with $HNO_3$ to a pH of 6 prior to the start of the experiment and was measured at the end of the experiment and it was on average pH $7.27 \pm 0.29$. We did not use any 'MS nutrient solution', see the supplement and chapter 2.1 for the composition of the nutrient solution used in our experiment. In terms of the control experiment the strategy of our experiment is to perform growth experiments that compare Si uptake and isotope fractionation between species. Performing a negative control experiment (meaning plants growing in Si free growth solution) makes no sense as Si is not transferred between compartments hence there would be no isotope fractionation.

**Line 114: Actually recent research would contend that these aquaporins are barely permeable to silicic acid since their pore size is less than that of silicic acid. See recent research by Guerriero et al and Exley et al. You do not have to agree with this new research but you should include it in your deliberations.**

This point is not relevant in the context of the statement made here, as we are discussing the functionality of Lsi1 homologue in tomato and not in *Cannabis sativa*. We have referred to the general controversies regarding the uptake of Si in line 50ff and 56ff, this includes the work by Exley and co-worker and their concerns raised that Lsi1 homologues found in Cannabis sativa have a modelled maximum pore size of 1.77 Å (Exley et al., 2020) whereas the maximum radius of silicic acid is estimated to be 4 Å (Exley et al., 2019).

**Line 117: Na4SiO4**

See above.

**Line 119: Na4SiO4**

See above.

**Line 123: Can you show how this was achieved as you cannot have a complete lid since the plant has to be accomodated. It is very difficult to make such a seal. A good positive control should be pots without plants sealed in an identical way but using, for example a plastic tube instead of the plant.**

The lids were placed inside the pots on an internal rim. The plants were fixed in their ports with a 3 cm thick foam rubber disc. All unused ports in the lid were sealed with rubber stops (see Figure 1 for the experimental setup). Because of all these measures, the amount of evaporation is extremely low. Even in the unlikely case that evaporation was non-negligible our experiments were made under the same conditions, in the same containers using the same sealing measures to minimise evaporation. A potential bias is thus identical for all three plant species and all triplicates. For this reason, we did not consider it advisable to carry out control tests. Using a plastic tube would raise other methodological questions, the clarification of which is far beyond the focus of our investigations.

We have rephrased it to make this clear: *"The pots were closed with a fixed and completely sealed lid, and thus evaporation is considered very small and, in any case, identical between the plant species and triplicates."*

[Figure]

*Figure 1: Overview of the arrangements in the growth chamber. The containers were closed with a lid, which was lying on an internal rim, all unused ports were sealed with rubber stoppers and the plants were fixed within a foam disk in their ports to minimise evaporation.*

**Line 126: I am wondering how the additional 200 mg/L sodium in the Si groups might influence both evaporation and transpiration. Since sodium was not added to the controls then you do really require some other form of control.**

Our nutrient solution did not contain 200 mg/L sodium. It is initially approximately 25 µg/g (or 25 mg/L). The sodium concentration of the nutrient solution throughout the experiment is given in

Figure S1. We have not performed any control experiments, neither without sodium nor silicon. As stated above control experiments do not make sense in isotope fractionation studies.

**Line 145: What about the high sodium content? Also problematic.**

The employed chromatography procedure can handle those large amounts of Na. In fact, in the fusion process an additional 400 mg of NaOH is introduced to dissolve silicates. The Na concentration of the purified solution is verified to be below the detection limits, prior to the stable isotope ratio measurements.

**Line 155: an exploratory study? Perhaps this should be included herein? If this preliminary work was integral to method development as seems to be the case then it also requires to be reviewed.**

The exploratory study is not an independent study from the here presented results. Rather a test with a subset of the samples to determining the amount of sample needed to obtain. We have adapted the sentence to clarify this: *"50-800 mg of plant material, depending on the Si concentration determined in an exploratory subset of the samples, was weighed into Ag crucibles and combusted overnight (2h at 200 °C, 4h at 600 °C, then cooled to room temperature) in a furnace (LVT 5/11/P330, Nabertherm)."*

**Line 158: Why is this added at this point?**

This is the chemical reagent needed to perform the dissolution of silicates, see also our explanation in the previous response letter.

**Line 161: So, it is a dry digestion? What about the nutrient solutions?**

Yes, the fusion is performed dry. The nutrient solution is dried down (see chapter 2.5.1, line 149), prior to the fusion.

**Line 162: I don't understand how this is done and how you would know that you have separated 60 micrograms of Si. I do not want to read these papers to know this so a brief explanation would be helpful. What happens to silicon-rich solutions when their pH is adjusted to 1.5? Are these solutions undersaturated wrt silicic acid or will they be solutions of Si4+?**

From our previous response: *"The dissolution procedure of silicates, silicon and bio silica is state of the art in geosciences. Si is present in $SiO_2$ as $Si^{4+}$, counterbalanced by 2 $O^{2-}$. Therefore, we do not need to convert Si into a cation. The NaOH accelerates the dissolution of the oxide, and after the addition of water silicon is present as silicic acid ($H_4SiO_4$ and depending on the pH also in the form of $H_3SiO_4^-$ (see e.g. (Stamm et al., 2019), their Fig. 1 for an aqueous Si species in equilibrium diagram)."* There is no change in the silicon-rich solution when the pH is adjusted to 1.5, this is however needed for the procedure with the anion exchange resin. The solutions (before and after purification) are undersaturated with regard to the solubility of silicic acid to avoid polymerisation and precipitation.

The dissolution procedure of silicates, silicon and bio silica is standard in geosciences and in particular in Si isotope geochemistry. We do not think that they have to be repeated in detail in this paper that should not be burdened with analytical background that geochemists are fully aware of, and others can read in the cited literature. We have clarified some aspects of chapter 2.5.3: *"The crucibles containing the sample (nutrient solution or plant material) and NaOH were placed in a furnace at 750 °C for 15 min to perform the fusion. The fusion cake was dissolved in ultrapure water (for 24h, followed by 30 min ultrasonic bath), the solution was decanted into precleaned PP flask. The remains of the fusion cake were fully dissolved in 0.03 M HCl (for 3h), both solutions were combined and the pH was adjusted to 1.5. The Si concentration was determined by ICP-OES and approximately 60 µg Si (present in the form of silicic acid) was chromatographically separated using cation exchange resin (following a*

*procedure outlined by Georg et al., 2006; Zambardi & Poitrasson, 2011; Schuessler & von Blanckenburg, 2014). The Si yield of the fusion procedure and the column chemistry was determined in a 1:10-fold dilution by ICP-OES. Si blanks of the fusion and column separation procedure were in general below 1 μg Si, equivalent to less than 1 % of the total Si processed. See Methods S3 for more details."*

**Line 163: When I asked you previously about whether or not you were using pH to convert Si to its cation you said no. However you cannot use a cation exchange resin to collect Si unless it is a cation. I have 40 years experience of Si chemistry, you need to explain this procedure clearly since it forms an integral part of your results.**

Again, this technique is standard, well-tested, and very well-known in geochemistry. The reviewer may not be aware of this substantial body of analytic-chemical literature. Si is present as a silicic acid in our solution, thus we can purify Si using a cation exchange resin retains unwanted cations (e.g. Ca, Na, Mg etc…) but let neutral silicic acid pass through. No further changes made, see comment before.

**Line 185: Where is the Si in stem tissue?**

Shoots include the stem and the leaves, we have clarified this on the first occurrence of the term shoots, see also the glossary of Esau's Plant Anatomy for a definition (Evert, 2006).

**Line 187: stem?**

See the comment before.

**Line 189: No mention is made about the health of plants across the different treatments. Did plants grow equally well in each treatment. Did any plants die or show any signs of disease. Were root masses different between treatments as this would influence how much root-associated water was being counted as transpired water. Details, including pictures if possible is needed on plant health throughout the study.**

All the requested information is presented in the results chapter:

- Chapter 3.1 reports on the root and shoot dry mass, the individual results ('raw data') are provided in Table S4.

Competition for light (especially for mustard) and nutrients probably led to the different biomass of the plants within the containers, whereas the different temperature requirements of the three plant species may have been responsible for their different biomass formation (see also line 58ff).

- No plants have died during the experiment.
- Regarding the health of the plants we have stated in chapter 3.2 that mustard showed first signs of nutrient deficiency in the form of chlorosis in young and old leaves. This is (likely) caused by the full consumption of the available nutrient elements (Ca and Mg). Other than that, the plants were healthy.

As all this information was already present no changes were made.

[Figure]

*Figure 2: Image to support the plant's health between week 4 and week 5.*

**Line 190: Check written English here, this sentence does not make sense to me.**

*Thanks for bringing this up, we have improved the section: "We define plant transpiration as the amount of water taken up by the plants via the roots. Transpiration was measured weekly by weighing the remaining growth solution with the lids and plants removed. The difference in mass from the previous week is considered to be the mass of water transpired by the plants. The gravimetrically determined transpiration does not account for the amount of water present in the plants at harvest nor any possible guttation (Joachimsmeier et al., 2012)."*

**Line 191: How do you account for water associated with the roots when you lift the plants from their pots?**

The roots are lifted such that the roots drip their excess water into the nutrient container. Water which is adhered or associated with the roots will be counted as a transpiration loss. When the roots are immersed back into the nutrient solution the adhered / associated water will be mixed again with the nutrient solution. No changes made.

[Figure]

*Figure 3: Exemplary tomato plants (container 3) lifted from the nutrient solution to measure the transpired water mass during the previous week. Roots were at the time of taking the picture still dripping excess water back into the container, the reading of the balance was made after the roots have stopped dripping and the balance was stable.*

**Line 193: Guttation is not negligible and it varies considerably between plants. For example, anyone who has grown cucumber plants will see guttation visibly as the tips of leaves. Find data on guttation for all your plants and include it in your calculations. The reference you cite is not an authoritative example.**

We have grown tomato, mustard, and spring wheat, and not cucumber and can thus only comment regarding our experiments: we did not observe guttation. This is not to a surprise, since under the conditions of the climate chamber 65% humidity (day and night) and permanent air flow, any possible guttation liquid would evaporate, which means that escaping Si would remain on the shoot surface and counted towards the shoot Si content at the end of the experiment. According to Singh, *"high atmospheric humidity is an essential prerequisite for guttation fluid to appear. As stated earlier, guttation is ample and continues for a relatively longer period when the soil and atmosphere are saturated with water"* (Singh, 2016).

We have clarified that guttation was not observed, but if it were present that the Si would not have been lost but counted towards the amounts of Si present in the shoots. This can also be seen in the high retrieval rate (Si pool present at the start of the experiment vs the Si pool present at the end of the experiment) of up to 100% (see Table 3). In Chapter 4.1 we added: *"Guttation (Joachimsmeier et al., 2012; Yamaji et al., 2008) and litter fall were not observed during the experiment. Even if guttation were present no Si would be lost since under the experimental conditions the fluid would evaporate leaving amorphous silica on the shoots. Thus, silicic acid excreted by guttation is counted towards the Si amounts in the shoots."*

**Line 196: Do you mean the weight of whole plants prior to sampling?**

No, this is the dry weight of the biomass after harvest and drying to constant weight. No change made.

**Line 199: Do you mean the sum of Si in roots, stem and shoot?**

Yes, the total Si mass in the roots, shoots (which include the steam and leaves). No change made.

**Line 200: By transfer you mean from stem to shoot?**

Shoots include stems, see also the comment before. No change made.

**Line 204: You mean the amount of silicic acid entering plants with transpired water? I think some example calculations are required here since the culture medium silicic acid content fell continuously over the growth period both due to transpired water being replaced with pure water but also sampled water being likewise replaced. I do wonder why you needed to replace water lost as this greatly complicates things. Just use a high volume of culture medium.**

Yes, this is the amount of silicic acid entering the plants with the transpired water. We have extended the paragraph and provided the formula to calculate the expected concentration of Si in the plants, assuming a purely passive uptake: *"We also calculated an "expected Si uptake" defined to represent exactly the mass of Si contained in the water utilised. This value was calculated from on the amount of transpired water and the nutrient solution Si concentration determined in the week prior:*

$$Expected\ Si\ Uptake = \sum_{Week=1}^{Week=6} [Si]_{week\ i-1} \cdot m_{transpired\ water, week\ i}$$

*where $[Si]_{week\ i-1}$ is the Si concentration in the nutrient solution the week prior, and $m_{transpired\ water,\ week\ i}$ the mass of water transpired during past week."*

Not replenishing the water transpired by the plants would eventually expose the roots to air instead of the nutrient solution, thereby changing the *effective* root biomass. This would complicate the study even more.

**Line 206: I am confused. If you use the change in silicic acid concentration of nutrient media combined with water loss due to transpiration and sampling over the culture period then you will have a figure for how much silicic acid WAS taken up over this period. Si loss from nutrient media = Si movement into plant. Si deposited in plant cannot be higher than the amount taken up. It can be lower due to silicic acid taken up not being deposited and being lost by guttation. This why you need to show your calculations for this 'expected' value. It is not an expected value it is an actual value based upon loss of Si in nutrient media.**

The measured and the expected Si uptake are two independent measures. The measured Si uptake is the determined by the biomass and the Si concentration in the roots and shoots. The expected Si uptake is determined by the mass of transpired water and the Si concentration of the nutrient solution. These two measures can be equal in the case of a purely passive uptake. A rejective Si incorporation is when the measured Si uptake is smaller than the expected Si uptake by transpiration and vice versa for an active incorporation. We have clarified the section: *"The plant Si uptake characteristics can be classified based on the ratio between the measured (based on the biomass and the Si concentration measured therein) and the expected Si uptake. A ratio of greater than 1 indicates an active uptake mechanism, a ratio much smaller than 1 a rejective strategy, and a ratio of 1 indicates passive uptake."*

Si deposited in the plants is not higher than the amount lost from the nutrient solution, this information can be found in chapter 3.3 and the accompanying Table 3.

**Line 208: These need to take account of all water loss being replaced by pure water?**

As described in ch. 2.4 we have taken account for the water loss, both in the element and the isotopic budgets. The weekly sampling has taken place after the replenishing the nutrient solution with ultra-pure water (which contained no significant amounts of the investigated nutrients) and arrogating for a prolonged time. No changes made.

**Line 214: The isotopes of Si in nutrient media are not changing over time?**

Of course, the composition of Si in the nutrient solution changes over time (see Figure 2 in our manuscript) if there is isotope fractionation during uptake! This is the point of the entire experiment, and basic stable isotope chemical mass balance. In detail mass balance requires that the overall pool of Si in the experiment (Si in the nutrient solution, plants) shall be invariant. We calculate the mass balance at the start and at the end of the experiment to assure that this requirement is fulfilled. The results (in ch. 4.1) show this: *"There is no significant difference between the isotopic composition of the starting solution and the weighted average of the isotopic compositions of the different compartments at the end (see Table 4)."*

**Line 223: A complicated way to collect silica from plant tissue. Just follow the digestion method of Law and Exley 2011, much simpler.**

Thanks for providing information on an alternative method for the extraction for plant silica.

**Line 224: revealed?**

Changed.

**Line 224: This is a Result of this study? Put in Results not here.**

We have removed this sentence from the materials and method section.

**Line 233: Where are the transpiration data for all plants including the controls? These are critical data and should be presented.**

The data is shown in Figure 1, panel A. We realised that the raw data were not provided, and have added the weekly measured mass of transpiration in the supplement tables (Table S6).

**Line 238: What about differences between control and treatment plants?**

As remarked in the introduction to this response letter, we have performed a growth experiment in which three crop species with known differences in their Si uptake capabilities were grown under the same environmental conditions. In such a study, there are no control experiments in a *classical sense*. We have performed all experiments in triplicate, with four plants in each pot and have discussed possible systematic differences between the triplicates and within the pots and cause thereof in our manuscript (see e.g. ch. 3.1, and the individual results ('raw data') are provided in Table S4).

**Line 240: There are no data for the control plants?**

There were no control experiments performed, see the explanation in the introduction of this response letter. No change made.

**Line 245: Larger plants had higher transpiration, not unexpected!**

No change made

**Line 245: What is water use efficiency? How is it calculated and why?**

This is explained in ch. 2.6.1. We added a link to the chapter.

**Line 248: This is where much clearer information is needed as to what exactly was measured. You have Si concentrations for each nutrient solution for each weekly time point. You can use these data to measure exactly how much Si has gone from each of them over the culture period. You also have bulk Si measurements for each plant and you can then compare Si in a plant with Si that was lost from the culture solution in which the plant was growing. Clearly the amount in the plant cannot exceed the amount lost from the nutrient solution. What do these data look like?**

These results are already presented in Table 3. $m_{start}$ is the mass of the element in the nutrient solution before the start of the experiment, $m_{End}$ is the remaining mass of the element in the nutrient solution after the harvest of the plants. $m_{Plants}$ is the mass of the element found in the plants. The amount of Si found in the plants has not exceeded the amount lost from the nutrient solution. The retrieval rate for Si is between 79 – 100 % and corroborate the reliability of our results. No change made.

**Line 249: Do you mean based upon how much Si disappeared from culture media? Where are these data in Table 1?**

Yes, the data is shown in Figure 1, panel B and in the supplement Figure S1 for the other nutrient elements. The results were however not provided in tabulated form. We have supplied these data and added to the supplement the concentration of the nutrient Ca, Fe, K, Mg, P, S, and Si and silicon isotope composition in tabulated form (see our new Supplement Table S5).

**Line 250: So the question is (i) what was the Si content of tissues at the start of the experiment or in the control plants at the end of the culture period. (ii) What is the difference in Si tissue content in the Si groups at the end of the culture period and (iii) how does this 'difference' compare with how much Si was actually lost from nutrient media over the culture period. These are the critical data at this point. Where are they? Once these data are established one might ask whether there is any relationship with transpired water.**

We are somewhat perplexed by these questions: these questions are answered in our manuscript and the results are presented (see Table S4 and ch. 3.3 and 4.1)

   (i)     The plants have been germinated on half strength nutrient media containing no Si, thus the amount Si at the start of the experiment is equal to that of the seeds.
           There were no control experiments performed, see also the introduction to this response.
   (ii)    The Si content of the shoots (which includes the stems!) and the roots are presented in Table S4. The only source of Si which could be present at the start of the experiment originates from the seed. In an approximation, the difference between start and the end of the experiments is equal to the amounts at the end, neglecting the initial Si amounts in the seeds.
   (iii)   This is the retrieval rate, see results in 3.3 and discussion in 4.1.

**Line 251: Where are the data for stems? They should be included. They are not included anywhere in figures or tables.**

See our comment before. The definition of shoots include stems and leaves.

**Response to Prof. Hodson**

**Line 46 regulates**
**Line 212 revealed**
**Line 219 3 Results should be a subheading and on the next line**
**Line 237 wheat as an Si**
**Line 416 researchers**
**Line 438 experiments**

We thank Prof. Hodson and made the suggested corrections.

**Additional changes made to the manuscript**

We added in ch. 3.2 a short discussion of the expected Si uptake and the ratio of measured and expected Si uptake results: "*The expected Si uptake (see ch. 2.6.1 and Eq. 3 for definition) traces the passive uptake of Si contained in the water utilised by the plants. The dynamics throughout the experiment is shown in Figure 1c (closed symbols) together with the ratio of measured and expected Si uptake (open symbols) at the end of the experiment. The measured and expected Si uptake ratios for all three species deviate significantly from 1 (see Table 2). The means of the measured and expected Si uptake for mustard ($57.2^a \pm 1.3$ vs $457.9^b \pm 16.4$), wheat ($337.0^b \pm 67.9$ vs $177.3^a \pm 40.7$) and tomato ($15.5^a \pm 4.9$ vs $141.1^b \pm 27.0$) are significantly different (based on t-test at 5% significance level, denoted). This indicates that Si uptake and/or transport in the three plant species investigated under the given environmental conditions differ from unspecific passive uptake and/or unspecific passive transport within the plants.*"

[revised manuscript text omitted]

The Table S5 is in the following pages.

Table S5: Composition (major element concentration (in µg g$^{-1}$) and silicon isotopic composition) of the weekly sampled nutrient solutions for the individual pots. See Table S3 for the starting composition (week 0).

| Difference between |  | Week 1 | Week 2 | Week 3 | Week 4 | Week 5 | Week 6 |
|---|---|---|---|---|---|---|---|
|  |  | Week 0 | Week 1 | Week 2 | Week 3 | Week 4 | Week 5 |
|  |  | [g] | [g] | [g] | [g] | [g] | [g] |
| Mustard | Pot 1 | 407.5 | 671.2 | 1842.2 | 2707.5 | 2863.3 | 2277.8 |
|  | Pot 4 | 239 | 929.9 | 2469.8 | 3397.6 | 2627.5 | 1673.4 |
|  | Pot 7 | 137.8 | 449.4 | 1500.7 | 3119.5 | 3413.4 | 2152.7 |
| Spring Wheat | Pot 2 | 231.1 | 374.7 | 791.1 | 1416 | 2083.7 | 2471.3 |
|  | Pot 5 | 147.6 | 331.3 | 914.9 | 1816.3 | 2306.7 | 2422.1 |
|  | Pot 8 | 133.5 | 189 | 474.7 | 996.6 | 1576.9 | 1756.8 |
| Tomato | Pot 3 | 206.7 | 316.2 | 526.6 | 772.6 | 998.9 | 951.6 |
|  | Pot 6 | 224.1 | 233.2 | 352.6 | 625.2 | 913 | 1005.8 |
|  | Pot 9 | 127.1 | 164.6 | 292.4 | 444.6 | 672.3 | 800.2 |

Table S 6: Weekly transpiration (in g), determined by weighing the pots without the plants. The transpired water was replenished weekly with ultrapure water.

[revised manuscript text omitted]

**Methods**

**Method S1 Preparation of the nutrient solution**

The nutrient solution was prepared from technical graded salts and dissolved in 10 L of ultrapure water. Macro nutrients 1.23 g MgSO$_4$·7H$_2$O, 3.54 g Ca(NO$_3$)$_2$·4H$_2$O, 0.33 g Ferric sodium EDTA, 3.6 g KNO$_3$, 1.1 g KCl and 0.82 g KH$_2$PO$_4$. Micro nutrients: 0.55 mg Al$_2$(SO$_4$)$_3$, 0.28 mg KJ, 0.28 mg KBr, 0.55 mg TiO$_2$, 0.28 mg SnCl$_2$ 2H$_2$O, 0.28 mg LiCl, 0.39 mg MnCl$_2$ 4H$_2$O, 6.1 mg H$_3$BO$_3$, 0.55 mg ZnSO$_4$, 0.55 mg CuSO$_4$ 5H$_2$O, 0.55 mg NiSO$_4$ 6H$_2$O, 0.55 mg Co(NO$_3$)$_2$ 6H$_2$O, 0.05 mg As$_2$O$_3$, 0.28 mg BaCl$_2$, 0.05 mg Bi(NO$_3$)$_3$, 0.05 mg Rb$_2$SO$_4$, 0.28 mg K$_2$CrO$_4$, 0.05 mg KF, 0.05 mg PbCl$_2$, 0.05 mg HgCl$_2$, 0.28 mg MoO$_3$, 0.05 mg H$_2$SeO$_4$, 0.28 mg SrSO$_4$, 0.05 mg H$_2$WO$_4$, 0.05 mg VCl$_2$). Silicon: 2.03 g Na$_2$O$_7$Si$_3$·3H$_2$O. pH was adjusted to 6.0 using HNO$_3$ (PA grade).

**Method S2 Plant germination and growth conditions**

Plant seeds were germinated in in Petri dishes containing a nutrient solution of half the concentration than the solution used for growth experiments (Methods S1) and in the absence of NaSiO$_4$. After cotyledons germinated, seeds and roots were clamped in a foam disk (3 cm high with a diameter of 2.5 cm) and each seedling (foam disk) transferred to a PP vial (50 mL centrifuge tube) filled with half-concentrated nutrient solution without NaSiO$_4$. Two weeks later, the foam disks including young plants were transferred to the experimental containers, four plants per container, 3 replicated container per species. These containers were opaque plastic containers 25.5 cm high, 20.5 cm deep and 20.5 cm wide (with a wall thickness of 0.5 cm). In order to reduce evaporation and to prevent algae growth in the nutrient solution, the containers were closed with opaque lids which had holes for the plants (foam disks). Germination and plant cultivation were performed in a growth chamber under controlled conditions. The temperature in the growth chamber during the day and night was maintained at 18 °C for 14 h and at 15 °C for 10 h, respectively, and the daylight intensity at the top of the container was adjusted to 350 µE m$^{-2}$ s$^{-1}$) at the start of the experiment. The relative humidity was maintained at approximately 65 %. For comparability, the cultivation conditions for the three species were the same, knowingly they are not equally suited for all species. The relatively low temperatures may have inhibited the growth of the more thermophilic tomato, while the conditions for mustard and summer wheat were close to their optimum. To supply the roots with oxygen, perforated PVC tubes were used to inject (approx. 6 L) room air into the nutrient solution twice a day for two hours each. The transpired water was replenished weekly with ultrapure water.

**Method S3 Dried plant and nutrient residue digestion and chromatographic purification of Si**

70    The crucibles containing the sample (dried down nutrient solution or charred plant material with approximately 400 mg NaOH) were placed in a high temperature furnace at 750 °C for 15 min. After cooling down the crucibles were cleaned externally with ultrapure water and placed in precleaned 50mL PP centrifuge tubes and covered with ultrapure water for 24 h. Thereafter, the crucibles were placed in an ultra-sonic bath for 30 min to facilitate the dissolution of the fusion cake. This solution #1 was decanted and collected in precleaned PP flask. The silver crucibles were then stored for ~3 h in a 0.03 M HCl solution and this

75    solution #2 was combined with solution #1 in the PP flask. Using concentrated HCl the pH was adjusted to 1.5. If the concentration was expected to be above 60 µg g$^{-1}$ additional 0.03 M HCl solution was added. 1:10-fold dilution was analysed by ICP-OES to determining the Si content. Approximately 60 µg Si from are loaded onto precleaned and preconditioned columns using a cation exchange resin (1.5 mL, DOWEX 50WX8, Sigma-Aldrich) and eluted using 5 mL ultrapure water. The cation exchange resin is then regenerated using HCl and HNO$_3$. The Si yield of the fusion procedure and the column

80    chemistry was determined in a 1:10-fold dilution by ICP-OES.

---

## Author Response (AR3)

Dear Prof. Bahn

Thank-you very much for the attentive handling of our manuscript. Following the last round of reviews, we have focussed the revision on improving accessibility and placing this study into the context of broader perspective (namely, the impact of Si-rich crop residues removal in food production).

In detail we have revised the abstract, introduction and the conclusion to improve the accessibility for a broader audience and have placed our conclusions and its broader implications more prominently in the abstract.

Furthermore, we have re-read the manuscript and supplement to carefully check its scientific content. The chemical formula of sodium silicate trihydrate ($Na_2O_7Si_3 \cdot 3H_2O$) which was stated false in the supplement has been corrected.

We hope that this revision makes this MS suitable for publication.

Best regards,

Daniel A. Frick on behalf of all co-authors.

[revised manuscript text omitted]
$. Here we have shown that Si in mustard roots is precipitated as biogenic silica in the roots, a process shown observed previously for 
[revised manuscript text omitted]

| | | | Ca | Fe | K | Mg | P | S | Si | δ30Si | |
|---|---|---|---|---|---|---|---|---|---|---|---|
| | | | µg/g | µg/g | µg/g | µg/g | µg/g | µg/g | µg/g | ‰ NBS28 | 2 s |
| Week 1 | Mustard | Pot 1 | 63.3 | 4.4 | 210.5 | 14.0 | 20.2 | 21.4 | 47.3 | -0.27 | 0.09 |
| | | Pot 4 | 62.0 | 4.4 | 206.2 | 13.9 | 19.7 | 21.3 | 47.3 | -0.17 | 0.02 |
| | | Pot 7 | 61.7 | 4.4 | 205.2 | 13.8 | 19.4 | 20.6 | 46.6 | -0.25 | 0.02 |
| | Spring Wheat | Pot 2 | 63.7 | 4.4 | 210.3 | 14.0 | 20.0 | 21.3 | 47.0 | -0.26 | 0.09 |
| | | Pot 5 | 63.9 | 4.5 | 209.1 | 14.2 | 19.6 | 21.5 | 47.1 | -0.29 | 0.06 |
| | | Pot 8 | 64.6 | 4.5 | 214.1 | 14.3 | 20.4 | 21.9 | 46.8 | -0.27 | 0.05 |
| | Tomato | Pot 3 | 63.8 | 4.4 | 211.5 | 14.2 | 20.2 | 21.8 | 47.2 | -0.33 | 0.03 |
| | | Pot 6 | 62.8 | 4.4 | 209.5 | 14.1 | 20.4 | 21.8 | 47.0 | -0.29 | 0.10 |
| | | Pot 9 | 63.8 | 4.5 | 213.6 | 14.0 | 20.6 | 21.8 | 47.0 | -0.22 | 0.05 |
| Week 2 | Mustard | Pot 1 | 54.2 | 4.2 | 188.8 | 12.8 | 16.6 | 18.3 | 47.4 | -0.30 | 0.11 |
| | | Pot 4 | 44.8 | 4.2 | 171.5 | 11.6 | 13.9 | 16.2 | 47.3 | -0.23 | 0.09 |
| | | Pot 7 | 54.1 | 4.3 | 192.0 | 12.8 | 16.6 | 18.4 | 47.9 | -0.19 | 0.03 |
| | Spring Wheat | Pot 2 | 62.8 | 4.3 | 200.6 | 13.8 | 18.3 | 20.7 | 45.2 | -0.34 | 0.03 |
| | | Pot 5 | 62.8 | 4.4 | 198.2 | 13.6 | 17.7 | 20.9 | 44.3 | -0.22 | 0.08 |
| | | Pot 8 | 62.3 | 4.4 | 203.3 | 13.8 | 18.7 | 21.0 | 44.8 | -0.21 | 0.03 |
| | Tomato | Pot 3 | 60.2 | 4.2 | 202.7 | 13.3 | 18.8 | 21.3 | 46.8 | -0.32 | 0.10 |
| | | Pot 6 | 62.1 | 4.3 | 208.6 | 13.5 | 19.5 | 21.5 | 47.0 | -0.25 | 0.04 |
| | | Pot 9 | 62.1 | 4.3 | 208.0 | 13.7 | 19.8 | 21.3 | 46.8 | -0.26 | 0.08 |
| Week 3 | Mustard | Pot 1 | 32.7 | 3.7 | 151.5 | 9.3 | 9.9 | 11.1 | 43.3 | -0.16 | 0.09 |
| | | Pot 4 | 15.8 | 3.7 | 119.4 | 6.8 | 5.8 | 5.4 | 42.2 | -0.15 | 0.08 |
| | | Pot 7 | 24.5 | 3.7 | 130.4 | 8.1 | 6.5 | 7.6 | 44.1 | -0.11 | 0.08 |
| | Spring Wheat | Pot 2 | 59.0 | 3.9 | 178.1 | 12.4 | 13.2 | 18.5 | 38.1 | -0.20 | 0.12 |
| | | Pot 5 | 56.8 | 3.8 | 158.3 | 11.5 | 10.0 | 17.2 | 29.6 | -0.01 | 0.22 |
| | | Pot 8 | 60.4 | 4.0 | 187.5 | 12.8 | 14.5 | 19.8 | 38.7 | -0.17 | 0.11 |
| | Tomato | Pot 3 | 51.4 | 3.6 | 185.8 | 11.2 | 13.9 | 18.7 | 43.5 | -0.17 | 0.04 |
| | | Pot 6 | 55.0 | 3.6 | 194.1 | 12.0 | 15.5 | 19.6 | 43.8 | -0.17 | 0.02 |
| | | Pot 9 | 57.2 | 3.9 | 199.1 | 12.4 | 16.3 | 20.1 | 44.5 | -0.17 | 0.17 |
| Week 4 | Mustard | Pot 1 | 15.7 | 3.4 | 116.0 | 6.2 | 6.2 | 5.2 | 39.4 | -0.13 | 0.05 |
| | | Pot 4 | 1.7 | 3.4 | 83.9 | 3.1 | 1.1 | 4.0 | 38.1 | -0.07 | 0.15 |
| | | Pot 7 | 2.3 | 3.4 | 70.3 | 3.4 | 0.4 | 0.4 | 39.8 | -0.10 | 0.12 |
| | Spring Wheat | Pot 2 | 54.2 | 3.8 | 138.6 | 11.2 | 8.3 | 17.1 | 30.0 | 0.00 | 0.10 |
| | | Pot 5 | 51.2 | 3.6 | 108.6 | 9.9 | 3.8 | 15.8 | 13.1 | 0.33 | 0.15 |
| | | Pot 8 | 55.7 | 3.9 | 151.9 | 11.5 | 9.6 | 18.0 | 28.5 | -0.06 | 0.11 |
| | Tomato | Pot 3 | 39.7 | 3.4 | 157.3 | 9.1 | 9.6 | 16.9 | 42.2 | -0.14 | 0.13 |
| | | Pot 6 | 46.3 | 3.4 | 175.5 | 10.3 | 12.0 | 18.1 | 43.0 | -0.16 | 0.05 |
| | | Pot 9 | 50.6 | 3.8 | 183.2 | 10.8 | 13.2 | 18.7 | 43.9 | -0.14 | 0.05 |
| Week 5 | Mustard | Pot 1 | 5.4 | 3.3 | 92.2 | 3.6 | 2.7 | 0.3 | 35.7 | -0.04 | 0.07 |
| | | Pot 4 | 0.1 | 3.5 | 61.3 | 0.3 | 0.2 | 0.3 | 36.4 | -0.11 | 0.11 |
| | | Pot 7 | 0.0 | 3.1 | 35.0 | 0.1 | 0.0 | 0.5 | 34.2 | -0.02 | 0.12 |
| | Spring Wheat | Pot 2 | 49.3 | 3.6 | 93.2 | 9.6 | 3.0 | 15.3 | 17.4 | 0.22 | 0.03 |
| | | Pot 5 | 47.5 | 3.4 | 76.1 | 8.8 | 0.2 | 14.5 | 2.6 | 1.36 | 0.14 |
| | | Pot 8 | 51.7 | 3.7 | 115.1 | 10.3 | 5.2 | 16.4 | 19.0 | 0.17 | 0.10 |
| | Tomato | Pot 3 | 27.5 | 3.1 | 130.2 | 6.8 | 4.9 | 14.3 | 40.9 | -0.10 | 0.03 |
| | | Pot 6 | 34.1 | 3.2 | 147.1 | 7.9 | 7.4 | 15.7 | 41.7 | -0.10 | 0.08 |
| | | Pot 9 | 39.8 | 3.6 | 155.6 | 9.0 | 9.3 | 16.6 | 43.2 | -0.20 | 0.08 |
| Week 6 | Mustard | Pot 1 | 0.4 | 3.1 | 77.6 | 0.8 | 0.5 | 0.4 | 33.4 | -0.08 | 0.14 |
| | | Pot 4 | 0.0 | 3.4 | 50.3 | 0.1 | 0.2 | 0.4 | 35.4 | 0.06 | 0.28 |
| | | Pot 7 | 0.0 | 3.5 | 21.6 | 0.0 | 0.1 | 0.8 | 34.7 | 0.01 | 0.10 |
| | Spring Wheat | Pot 2 | 44.9 | 3.2 | 63.2 | 7.4 | 0.0 | 11.4 | 4.3 | 0.75 | 0.03 |
| | | Pot 5 | 44.7 | 3.0 | 60.0 | 7.0 | 0.0 | 12.1 | 0.2 | 0.00 | 0.00 |
| | | Pot 8 | 50.6 | 3.4 | 94.2 | 8.1 | 1.3 | 14.2 | 9.6 | 0.51 | 0.13 |
| | Tomato | Pot 3 | 16.5 | 2.9 | 111.3 | 4.1 | 0.6 | 9.5 | 38.9 | -0.06 | 0.04 |
| | | Pot 6 | 21.7 | 2.9 | 124.3 | 4.9 | 2.4 | 10.6 | 39.2 | -0.09 | 0.04 |
| | | Pot 9 | 31.5 | 3.4 | 144.6 | 6.6 | 6.2 | 13.5 | 41.6 | -0.13 | 0.06 |

**Table S5: Composition (major element concentration (in µg g$^{-1}$) and silicon isotopic composition) of the weekly sampled nutrient solutions for the individual pots. See Table S3 for the starting composition (week 0).**

| Difference between | | Week 1 | Week 2 | Week 3 | Week 4 | Week 5 | Week 6 |
|---|---|---|---|---|---|---|---|
| | and | Week 0 | Week 1 | Week 2 | Week 3 | Week 4 | Week 5 |
| | | [g] | [g] | [g] | [g] | [g] | [g] |
| | Pot 1 | 407.5 | 671.2 | 1842.2 | 2707.5 | 2863.3 | 2277.8 |
| Mustard | Pot 4 | 239 | 929.9 | 2469.8 | 3397.6 | 2627.5 | 1673.4 |
| | Pot 7 | 137.8 | 449.4 | 1500.7 | 3119.5 | 3413.4 | 2152.7 |
| | Pot 2 | 231.1 | 374.7 | 791.1 | 1416 | 2083.7 | 2471.3 |
| Spring Wheat | Pot 5 | 147.6 | 331.3 | 914.9 | 1816.3 | 2306.7 | 2422.1 |
| | Pot 8 | 133.5 | 189 | 474.7 | 996.6 | 1576.9 | 1756.8 |
| | Pot 3 | 206.7 | 316.2 | 526.6 | 772.6 | 998.9 | 951.6 |
| Tomato | Pot 6 | 224.1 | 233.2 | 352.6 | 625.2 | 913 | 1005.8 |
| | Pot 9 | 127.1 | 164.6 | 292.4 | 444.6 | 672.3 | 800.2 |

**Table S 6: Weekly transpiration (in g), determined by weighing the pots without the plants. The transpired water was replenished weekly with ultrapure water.**

**Methods**

**Method S1 Preparation of the nutrient solution**

[revised manuscript text omitted]

---

## Author Response (AR4)

Dear Prof. Bahn

Based on the recommendation of the "Anonymous Referee 4" we have made the following changes to the manuscript:

**L. 63. Lsi1 is not located/expressed in the root epidermis. It is located in the exodermis and endodermis.**

Thanks for bringing this to our attention, we have corrected this.

Line 60: *In rice, a cooperative system of Si-permeable channels at both the root exodermis and endodermis …*

Additionally, we changed the term "root epidermis" in the conclusion to "root cortex" to cover the different species:

Line 474: *The incorporation and fractionation of stable Si isotopes at the root cortex is likely governed by the preferential diffusion of the lighter homologue of silicic acid.*

**The concept of rejective, passive and active absorption was developed pre-discovery of transporters. Recent papers suggest that plants should be classified on the basis of whether they have functional Si transporters (Lsi1 and Lsi2) or not. Thereafter, quantitative absorption/presence can be influenced by many factors but functional transporters do not make for a passive or active uptake. The authors should make that distinction.**

We have expanded the traditional classification approach from Takahashi et al., 1990 to include the recommendation by Coskun et al., 2019b to classify based on the presence of Si transporters:

Line 49: *Traditionally, higher plants were grouped into three categories: active, passive and rejective, according to the amount of Si taken up (Marschner and Marschner, 2012).*

and

Line 63: *The research on the identification of molecular pathways and mechanisms supplements and extends the phenomenological classification of the Si uptake, in particular where genomic data is available that disclose functional Si transporters (Coskun et al., 2019b).*

We have rephrased the second and third paragraph of the introduction to clarify the distinction:

Line 52: *Rejective species (e.g. tomato, mustard, and soybean) strongly discriminate against Si during uptake (Epstein, 1999; Hodson et al., 2005; Ma et al., 2001; Takahashi et al., 1990). However, whether the terminology "active" or "passive" is justified is subject to an intense debate that revolves around the evidence for involvement of an active, metabolically controlled process in some plant species (Coskun et al., 2019a; Exley, 2015; Exley et al., 2020).*

Line 66: *Recent empirical studies demonstrated the simultaneous operation of passive uptake mechanisms and actively facilitated Si uptake through Si uptake transporter (Sun et al., 2016b; YAN et al., 2018). Yet other researchers have suggested that the low permeability of Lsi1 does not permit the transfer of silicic acid at all (Exley et al., 2020). Thus, it remains unclear what contribution active and passive Si transporters make during Si uptake by the different plant species.*

We hope that this revision makes our manuscript suitable for publication.

Best regards,

Daniel A. Frick on behalf of all co-authors.

[revised manuscript text omitted]